

# Long-time divergences in the nonlinear response of gapped one-dimensional many-particle systems

**Michele Fava[1], Sarang Gopalakrishnan[2], Romain Vasseur[3], Siddharth A. Parameswaran[4] and Fabian H. L. Essler[4]**

**1** Philippe Meyer Institute, Physics Department, École Normale Supérieure (ENS), Université PSL, 24 rue Lhomond, F-75231 Paris, France
**2** Department of Electrical and Computer Engineering, Princeton University, Princeton, New Jersey 08544, USA
**3** Department of Physics, University of Massachusetts, Amherst, MA 01003, USA
**4** Rudolf Peierls Centre for Theoretical Physics, Clarendon Laboratory, Oxford OX1 3PU, UK

## Abstract

We consider one dimensional many-particle systems that exhibit kinematically protected single-particle excitations over their ground states. We show that momentum and time-resolved 4-point functions of operators that create such excitations diverge linearly in particular time differences. This behaviour can be understood by means of a simple semiclassical analysis based on the kinematics and scattering of wave packets of quasiparticles. We verify that our wave packet analysis correctly predicts the long-time limit of the four-point function in the transverse field Ising model through a form factor expansion. We present evidence in favour of the same behaviour in integrable quantum field theories. In addition, we extend our discussion to experimental protocols where two times of the four-point function coincide, e.g. 2D coherent spectroscopy and pump-probe experiments. Finally, focusing on the Ising model, we discuss subleading corrections that grow as the square root of time differences. We show that the subleading corrections can be correctly accounted for by the same semiclassical analysis, but also taking into account wave packet spreading.


doi:10.21468/SciPostPhys.19.4.086

# 1 Introduction

The linear response to weak time-dependent perturbations [1,2] has historically been the main method of extracting information on dynamical properties of many-particle quantum systems. Especially in condensed matter setups, typical perturbations are weak enough to produce a response that is well described by linear response theory. This is, for example, the case in most conventional transport, light absorption, and neutron scattering experiments. From a conceptual standpoint, the fact that the outcome of these measurements can be modeled within linear response theory is convenient, as it directly relates the measurement outcome to two-point functions of local operators, which in turn provides detailed information about excitations in many-particle systems.

More recently, the advent of pump-probe experiments [3,4] and 2D coherent spectroscopy (2DCS) techniques [5–10] has made it possible to probe complex materials outside the linear response regime. These experimental techniques are well established as important tools in chemistry and few-body physics [11], but are relatively novel probes of many-body physics. Therefore, their potential applicability to furthering our understanding of strongly-correlated and exotic materials still needs to be explored. This has motivated theoretical works aimed at developing a comprehensive understanding of nonlinear response [1] in strongly correlated many-particle quantum systems [12–25,25–28]. Most analytical results available to date stem from calculations on systems that decouple into ensembles of few-body systems [29–31], free theories [25,29,32–36], or interacting integrable systems in one dimension [37–42]. However, until relatively recently it has been unclear how and to what extent these results generalize to systems that are not in one of these particular classes. There is currently an ongoing experimental and theoretical effort [43–48] to extend nonlinear response beyond the existing two-level system and integrable model paradigms. In this manuscript, expanding on a previous work of ours [49], we develop an understanding of third-order response functions predicting a long-time divergence in a broad class of models, namely gapped, one-dimensional systems featuring stable quasiparticle excitations at low energy. We show that when the system is perturbed by operators with a definite momentum, long-time divergences arise. These can be understood from a semiclassical perspective, accounting for ballistic propagation of quasiparticles and their scattering events; a picture that can be made precise using a wavepacket basis for the quasiparticles.

Before proceeding we note that many works over the years focused on wavepacket dynamics or a similar semiclassical description of the low-energy dynamics. In a seminal paper [50] Sachdev and Young showed how semiclassical arguments can account for the low-temperature linear response functions in the Ising model. More recently, Refs. [24,36] explained the presence of persistent oscillations in certain nonlinear response functions through a semiclassical picture accounting for ballistic motion of quasiparticles. In a spirit similar to this work, Refs. [51,52] found similar long-time divergences in a system with anyonic QPs, although with different exponents stemming from the fact that the QPs have nontrivial mutual statistics. Finally, a recent work [53] showed that wavepacket dynamics can be used as a way to derive generalized hydrodynamics in integrable models.

In the next two sections, we describe the setup (Sec. 2) to which our results apply and provide a summary of our results (Sec. 3). The rest of the manuscript is devoted to detailed derivations of our results.

## 2 Nonlinear response in pump-probe and two-dimensional coherent spectroscopy setups

The general setup of interest to us is as follows: we consider a one-dimensional many-particle system with Hamiltonian $H_0$ that is initially prepared in its ground state, which we assume to be translationally invariant. We then add a very small time-dependent perturbation

$$H(t) = H_0 + \sum_q h(t,q)A(q), \tag{1}$$

where we take $A(q)$ to be the Fourier transform of a local Hermitian operator $\mathcal{A}(x)$

$$A(q) = \int dx \, e^{iqx} \mathcal{A}(x). \tag{2}$$

Note that for concreteness we will assume that the system in consideration is defined in continuous space, but all the formulas equally apply to lattice models upon proper redefinition of integrals, Fourier transform, etc. The expectation value of $A(q)$ at time $t$ is then[1]

$$\langle A(q,t) \rangle = \langle A(q) \rangle_0 + V \sum_{n \geq 1} \int_{-\infty}^{\infty} dt_1 \dots dt_n \sum_{q_1,\dots,q_n} \chi^{(n)}(\boldsymbol{q}^{(n)}; \boldsymbol{t}^{(n)}) h(t_1,q_1)\dots h(t_n,q_n), \tag{3}$$

where $\langle A(q) \rangle_0$ denotes the expectation value in absence of the perturbation and where we have defined $\boldsymbol{q}^{(n)} = (q_1,\dots q_n,q)$, $\boldsymbol{t}^{(n)} = (t_1,\dots t_n,t)$. The nonlinear susceptibilities are given in terms of expectation values of nested commutators

$$\chi^{(n)}(\boldsymbol{q}; \boldsymbol{t}) = \frac{(-i)^n}{V} \theta(t_{n+1}-t_n)\dots\theta(t_2-t_1)\langle [\dots[A(q_{n+1},t_{n+1}),A(q_n,t_n)],\dots,A(q_1,t_1)]\rangle_{\mathrm{GS}}, \tag{4}$$

where $V$ is the volume of the system, and $\langle . \rangle_{\mathrm{GS}}$ denotes the expectation value in the ground state $|\Omega\rangle$ (Note that, as is standard, the $n^{\mathrm{th}}$-order response involves $n+1$ powers of the perturbing fields.). As we discuss below, this choice of normalization guarantees that, at finite times, $\chi^{(n)}(\boldsymbol{q}; \boldsymbol{t})$ is finite in the thermodynamic limit. As a consequence of the spatial homogeneity of the ground state $|\Omega\rangle$, the nonlinear susceptibility vanishes unless the momenta sum up to zero (mod $2\pi$ on the lattice), i.e.

$$\chi^{(n)}(\boldsymbol{q}; \boldsymbol{t}) \propto \delta_{Q,0}, \qquad Q = \sum_{j=1}^{n+1} q_j. \tag{5}$$

In the following we focus on $\chi^{(3)}(\boldsymbol{q},\boldsymbol{t})$ due to its relevance for pump-probe [3,4] and two-dimensional coherent spectroscopy (2DCS) [5–10] experiments. As detailed below, such experiments give access to $\chi^{(3)}(\boldsymbol{q},\boldsymbol{t})$ with two of the times $t_j$ (at which the perturbation acts) being close to one another.

The third-order nonlinear susceptibility can be expressed in terms of connected 4-point correlation functions

$$C_4(\boldsymbol{q}; \boldsymbol{t}) = \frac{1}{V} \langle A(q_4,t_4)A(q_3,t_3)A(q_2,t_2)A(q_1,t_1) \rangle_C. \tag{6}$$

Introducing a shorthand notation $\langle A_4 A_3 A_2 A_1 \rangle_C = C_4(q_4,q_3,q_2,q_1;t_4,t_3,t_2,t_1)$ we have[2]

$$\chi^{(3)}(\boldsymbol{q},\boldsymbol{t}) = i\langle A_4 A_3 A_2 A_1 \rangle_C - i\langle A_1 A_4 A_3 A_2 \rangle_C - i\langle A_2 A_4 A_3 A_1 \rangle_C + i\langle A_1 A_2 A_4 A_3 \rangle_C + \text{c.c.} \tag{7}$$

---

[1]For simplicity we restrict our attention to cases where the operator being measured at time $t$ is the perturbing operator $A$ itself.

[2]As required by causality only connected four-point functions contribute to $\chi^{(3)}$.

## 2.1 Pump-probe response

The pump-probe setup is described by a time-dependent Hamiltonian of the form

$$H(t) = H_0 + V_{\text{pump}}(t) + V_{\text{probe}}(t).\tag{8}$$

For simplicity we consider pump and probe potentials of the form

$$V_{\text{pump}}(t) = \mu\, \delta(t) \sum_q g(q) M(q), \qquad V_{\text{probe}}(t) = \mu' \sum_p f(t,p)\, M(p),\tag{9}$$

where $f(t,p) = f^*(t,-p)$ and $g(q) = g^*(-q)$ are smooth functions, $M(q) = M^\dagger(-q)$ are Fourier modes of an observable such as the magnetization, and $\mu$ and $\mu'$ are small parameters. Then the action of the pump potential can be viewed as a quantum quench

$$\begin{aligned}
|\psi(t = 0^+)\rangle &= e^{-i\mu \sum_q g(q) M(q)} |\psi(t = 0^-)\rangle \equiv |\psi_\mu\rangle,\\
|\psi_{\mu,\mu'}(t)\rangle &= T e^{-i \int_{0^+}^t dt' [H_0 + \mu' \sum_p f(t',p) M(p)]} |\psi_\mu\rangle,
\end{aligned}\tag{10}$$

with $T$ indicating that the exponential is time-ordered. We now introduce the following notations for the expectation value in the time evolved state (10)

$$\langle \mathcal{O} \rangle_{t,\mu,\mu'} \equiv \langle \psi_{\mu,\mu'}(t) | \mathcal{O} | \psi_{\mu,\mu'}(t)\rangle.\tag{11}$$

Assuming that the probe parameter $\mu'$ is small then gives

$$\begin{aligned}
&\frac{\langle M(Q)\rangle_{t,\mu,\mu'} - \langle M(Q)\rangle_{t,\mu,0}}{L}\\
&= \mu' \int_{0^+}^t dt' \sum_p f(t',p)\, \chi_\mu^{(1)}(Q,p;t,t')\\
&\quad + (\mu')^2 \int_{0^+}^t dt' \int_{0^+}^{t'} dt'' \sum_{p,p'} f(t',p) f(t'',p')\, \chi_\mu^{(2)}(Q,p,p';t,t',t'') + \mathcal{O}(\mu'^3),
\end{aligned}\tag{12}$$

where

$$\begin{aligned}
\chi_\mu^{(1)}(Q,p;t,t') &= -\frac{i}{L} \langle \psi_\mu | [M(Q,t), M(p,t')] | \psi_\mu\rangle, \qquad M(Q,t) = e^{iH_0 t} M(Q) e^{-iH_0 t},\\
\chi_\mu^{(2)}(Q,p,p';t,t',t'') &= -\frac{1}{L} \langle \psi_\mu | [[M(Q,t), M(p,t')], M(p',t'')] | \psi_\mu\rangle.
\end{aligned}\tag{13}$$

Finally, we imagine repeating the linear response measurement with $\mu = 0$ and subtracting the two results to obtain the pump-probe response

$$\Xi_{\text{PP}}(Q,p;t,t') = \chi_\mu^{(1)}(Q,p;t,t') - \chi_0^{(1)}(Q,p;t,t').\tag{14}$$

Choosing our initial state before the pump process to be the ground state $|\psi(t = 0^-)\rangle = |\Omega\rangle$ and then expanding $\Xi_{\text{PP}}(t,t')$ to second order in $\mu$ we obtain

$$\begin{aligned}
\Xi_{\text{PP}}(Q,p;t_1 + t_2, t_1) ={}& -\frac{\mu}{L} \sum_q g(q) \langle \Omega | [[M(Q,t_1 + t_2), M(p,t_1)], M(q)] | \Omega\rangle\\
&+ i\frac{\mu^2}{2L} \sum_{q,q'} g(q) g(q') \langle \Omega | [[[M(Q,t_1 + t_2), M(p,t_1)], M(q')], M(q)] | \Omega\rangle\\
&+ \mathcal{O}(\mu^3)\\
={}& \mu \sum_q g(q) \chi^{(2)}(q,p,Q;0,t_1,t_1 + t_2)\\
&+ \frac{\mu^2}{2} \sum_{q,q'} g(q) g(q') \chi^{(3)}(q,q',p,Q;0,0,t_1,t_1 + t_2) + \mathcal{O}(\mu^3).
\end{aligned}\tag{15}$$

In the cases of interest to us here the Hamiltonian and the ground state have a $\mathbb{Z}_2$ symmetry (e.g. spin rotational invariance by an angle $\pi$ around the x-axis in the case of the transverse field Ising chain and the spin-1 Heisenberg model), and the operators $M(q)$ are odd under this symmetry. In this case the term linear in $\mu$ in (15) vanishes and the leading contributions are proportional to $\chi^{(3)}$. For later convenience we now define

$$\chi_{\text{PP}}^{(3)}(q_1, q_2; \tau_1, \tau_2) = \chi^{(3)}(q_1, -q_1, q_2, -q_2; 0, 0, \tau_1, \tau_1 + \tau_2), \tag{16}$$

where $\tau_{1,2} > 0$. As we will see later (16) gives the dominant contribution to the pump-probe signal in the limit of interest.

## 2.2 Two-dimensional coherent spectroscopy (2DCS)

A second experimental setup of interest to us is that of 2DCS, in which the system is perturbed by a pair of field pulses at varying times. More precisely, in Eq. (9) for 2DCS we have $f(t, p) = \delta(t - t_1)F(p)$ and $\mu'$ is comparable to $\mu$. We can then proceed as for the pump-probe response. Assuming again that the Hamiltonian and the ground state have a $\mathbb{Z}_2$ symmetry and the operators $M(q)$ are odd under this symmetry, the leading terms in the 2DCS response are then

$$\frac{\langle M(Q)\rangle_{t,\mu,\mu'} - \langle M(Q)\rangle_{t,\mu,0} - M(Q)\rangle_{t,0,\mu'} + M(Q)\rangle_{t,0,0}}{L}$$
$$= \frac{\mu'\mu^2}{2} \sum_{q,q',p} g(q)g(q')F(p)\chi^{(3)}(q, q', p, Q; 0, 0, t_1, t_1 + t_2)$$
$$+ \frac{(\mu')^2\mu}{2} \sum_{q,p,p'} g(q)F(p)F(p')\chi^{(3)}(q, p, p', Q; 0, t_1, t_1, t_1 + t_2) + \dots \tag{17}$$

The first part is the same as the pump-probe response, while the second has a different temporal structure. As we will show, the dominant contribution is given in terms of the *non-rephasing* susceptibility

$$\chi_{\text{NR}}^{(3)}(q_1, q_2; \tau_1, \tau_2) = \chi^{(3)}(q_1, q_2, -q_2, -q_1; 0, \tau_1, \tau_1, \tau_1 + \tau_2). \tag{18}$$

The non-rephasing and pump-probe susceptibilities can be expressed in terms of connected correlators as follows

$$\chi_{\text{NR}}^{(3)}(q_1, q_2; \tau_1, \tau_2) = -\text{Im}\Big[ C_4(q_1, q_2, -q_2, -q_1; 0, \tau_1, \tau_1, \tau_1 + \tau_2) - C_{\text{NR}}(q_1, q_2; \tau_1, \tau_2)$$
$$- C_{\text{NR}}(q_1, -q_2; \tau_1, \tau_2) + C_4(q_1, -q_1, q_2, -q_2; 0, \tau_1 + \tau_2, \tau_1, \tau_1) \Big], \tag{19}$$

$$\chi_{\text{PP}}^{(3)}(q_1, q_2; \tau_1, \tau_2) = -\text{Im}\Big[ C_4(q_1, -q_1, q_2, -q_2; 0, 0, \tau_1, \tau_1 + \tau_2) - C_{\text{PP}}(q_1, q_2; \tau_1, \tau_2)$$
$$- C_{\text{PP}}(-q_1, q_2; \tau_1, \tau_2) + C_4(q_2, -q_2, -q_1, q_1; \tau_1, \tau_1 + \tau_2, 0, 0) \Big], \tag{20}$$

where $C_{\text{PP}}$ and $C_{\text{NR}}$ denote particular connected 4-point functions

$$C_{\text{NR}}(q_1, q_2; \tau_1, \tau_2) = \frac{1}{V}\langle A(-q_2, \tau_1)A(-q_1, \tau_1 + \tau_2)A(q_2, \tau_1)A(q_1, 0)\rangle_C, \tag{21}$$

$$C_{\text{PP}}(q_1, q_2; \tau_1, \tau_2) = \frac{1}{V}\langle A(-q_1, 0)A(-q_2, \tau_1 + \tau_2)A(q_2, \tau_1)A(q_1, 0)\rangle_C. \tag{22}$$

## 2.3 Nonlinear response in systems with stable, gapped quasiparticle excitations

The purpose of our work is to analyze the behaviour of connected 4-point functions as well as the dynamical nonlinear susceptibilities $\chi_{\mathrm{NR}}^{(3)}$ and $\chi_{\mathrm{PP}}^{(3)}$ in quantum many-particle systems that feature single-particle excitations with a finite gap $\Delta$ above the ground state. We further require these "one-QP excitations" to be stable, at least in some momentum range. At low particle density, and in particular at zero temperature, we expect a wave packet treatment of the QPs, accounting for their ballistic propagation and their scattering processes, to be applicable and provide reliable results. There is a wide variety of models and materials that display this kind of behaviour. Arguably the experimentally most important class are quasi-one-dimensional quantum magnets. One paradigm are Haldane-gap quantum spin chains [54–63]. Other examples are spin-1/2 field-induced gap quantum spin chain materials described by the sine-Gordon model [64–70] and Ising-like quantum magnets displaying confinement [71–76], where each stable meson mode can be identified with a distinct QP species.

Furthermore, stable quasiparticles are a hallmark of integrable many-particle systems. Among these, our results pertain in particular to cases where the ground state coincides with a "reference state" of the Bethe Ansatz. One of the simplest examples is provided by the transverse-field Ising model [77], which can be exactly diagonalized in terms of free fermions, in such a way that the ground state corresponds to the fermion vacuum (see e.g. Appendix A of Ref. [78]). Richer, but more complicated, examples are provided by integrable relativistic Quantum Field Theories (QFTs) [70, 79, 80] like the sine-Gordon model [81–83], the O(N) nonlinear sigma-model [84–88] and the Ising field theory in a magnetic field [71, 89–94]. We will use the fact that integrable QFTs provide non-trivial examples for our setup in order to benchmark some of our results.

## 3 Summary of results

Our main result is to show that if the action of $\mathcal{A}(x)$ on the ground state $|\Omega\rangle$ creates a stable 1-QP excitation, then the related connected 4-point function grows linearly in the time difference $|t_{32}| \equiv |t_3 - t_2|$

$$
\begin{aligned}
C_4(\boldsymbol{q};\boldsymbol{t}) &= \frac{1}{V}\langle\Omega|A(q_4,t_4)A(q_3,t_3)A(q_2,t_2)A(q_1,t_1)|\Omega\rangle_C \\
&\sim |t_3 - t_2| + o(|t_3 - t_2|) \quad \text{if } v|t_3 - t_2| \gg a_0.
\end{aligned}
\tag{23}
$$

Here $v$ is the typical group velocity of the stable QP excitations involved in the process, and $a_0$ a short-distance cutoff scale like the lattice spacing. In the following we develop a physically intuitive picture for the mechanism that gives rise to the behaviour (23) based on the kinematics and scattering of wave packets made from the stable QP excitation. We show the leading processes in the long-time limit, i.e. the one that give rise to contributions proportional to $|t_{32}|$, are processes like the ones in Fig. 1(a). The latter is meant to be interpreted in terms of probability amplitudes as follows. The system is initially in its ground state. At a time $t_1$ the local operator $\mathcal{A}(x_1)$ acts and creates, with a certain probability amplitude, a 1-QP wave packet with (average) momentum $q_1$. At time $t_2$ the operator $\mathcal{A}(x_2)$ acts and creates, with a certain probability amplitude, a 1-QP wave packet with (average) momentum $q_2$. Both these wave packets then propagate until they meet at a time $t_S$ and scatter elastically into two 1-QP wave packets.

For simplicity we assume that the only processes allowed such that the outgoing momenta remain $q_1$ and $q_2$.[3] At time $t_3$ ($t_4$) the first (second) of these wave packets is annihilated by

---

[3]The most general case could be treated in a similar fashion.

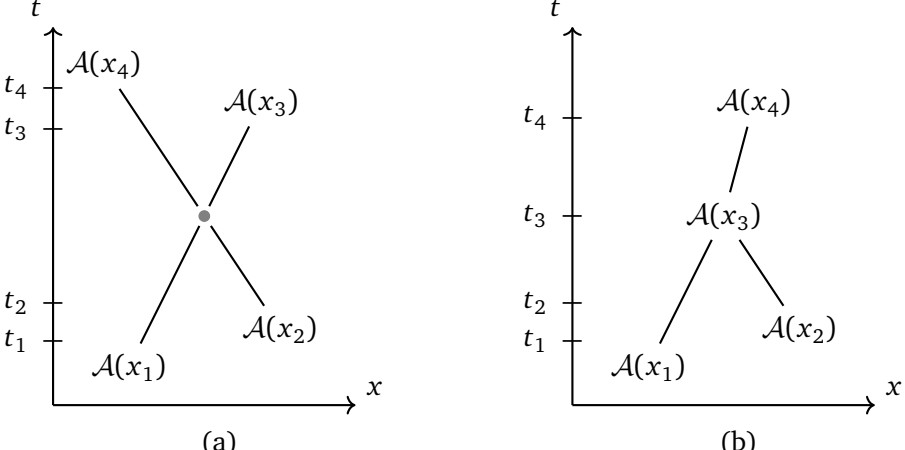

Figure 1: Two examples of scattering-connected processes (a) and fully-connected process (b). Black lines denote the trajectories of the center of QP wave packets, which, in the long-time limit can be approximated as ballistic. Grey dots denote scattering events taking place where two wave packets collide. QPs are created and annihilated by $\mathcal{A}$ operators. Scattering-connected processes are those where the removal of the scattering events would reduce the diagram to a disconnected one, i.e. a process whose amplitude can be expressed as the product of two amplitudes related to two-point functions.

the operator $\mathcal{A}(x_3)$ ($\mathcal{A}(x_4)$). We term these *scattering-connected* processes, as they are almost separable into products of two 2-point functions, except for the presence of scattering events which alter the free propagation of QPs. These should be contrasted with more generic processes that are "connected" by creation and annihilation processes involving the $\mathcal{A}$ operators themselves (see e.g. Fig. 1(b)), which we will refer to as *fully connected*. As we have already stated, our focus is on the late time regime. We differentiate between two cases:

1. All time differences $t_{ij} = t_i - t_j$ are large for $i \neq j$. Here the processes giving the leading contributions are just the ones where every $\mathcal{A}$ operator creates or annihilates a single QP, like in Fig. 1(a).

2. Two of the times $t_j$ coincide, but the remaining time differences are large. This different limit is important to understand the long-$\tau$ behavior in $C_{\mathrm{NR}}$ (21) and $C_{\mathrm{PP}}$ (22), of relevance to 2DCS and pump-probe experiments. In this case, there are more *scattering-connected* processes giving divergent contributions in the long-$\tau$ limit, e.g. those depicted in Figs. 2 and 3.

We now discuss these two cases in turn.

### 3.1 All time differences are large ($|t_{ij}| \to \infty$)

Here the leading processes are the ones where $\mathcal{A}(x_1, t_1)$ and $\mathcal{A}(x_2, t_2)$ create two independent QP wave packets over the ground state $|\Omega\rangle$. Under time-evolution, these wave packets scatter and the products are then annihilated by $\mathcal{A}(x_3, t_3)$ and $\mathcal{A}(x_4, t_4)$. The corresponding contribution to $C_4(\boldsymbol{q}; \boldsymbol{t})$ is given by

$$
\begin{aligned}
C_4(\boldsymbol{q}; \boldsymbol{t}) \sim \sum_a & F_{a_1}^*(q_1) F_{a_2}^*(q_2) F_{a_3}(q_3) F_{a_4}(q_4) e^{-i\epsilon_{a_1}(q_1)t_{21}} e^{-i(\epsilon_{a_1}(q_1)+\epsilon_{a_2}(q_2))t_{32}} e^{-i\epsilon_{a_4}(-q_4)t_{43}} \\
& \times \left[ \mathcal{S}_{a_1 a_2}^{a_3 a_4}(q_1, q_2)\delta_{q_3, -q_1} + \mathcal{S}_{a_1 a_2}^{a_4 a_3}(q_1, q_2)\delta_{q_3, -q_2} \right] |v_{a_1}(q_1) - v_{a_2}(q_2)||t_{32}|.
\end{aligned}
\tag{24}
$$

The meaning of the various factors is as follows.

- The indices $a_n$ label the distinct species of QPs and $\boldsymbol{a} = (a_1, a_2, a_3, a_4)$. In the simplest case there would only be a single such species.

- The $F_a(k)$ denote the matrix elements of the operators $\mathcal{A}(0)$ between the ground state and a excited state $|k\rangle_a$ involving a single QP of species $a$ with momentum $k$

$$F_a(k) = \langle \Omega | \mathcal{A}(0) | k \rangle_a \,. \tag{25}$$

  The state $|k\rangle_a$ will have an energy $\epsilon_a(k)$, and consequently, at the wave packet level, the QP will propagate with the group velocity $v_a(k) = \partial_k \epsilon_a(k)$.

- The time-dependent phase factors in Eq. (24) account for the fact that states with spatially well-separated QP wave packets are (approximate) energy eigenstates.

- The quantity $\mathcal{S}^{a'b'}_{ab}(q_1, q_2)$ encodes the amplitude for the processes where two QPs with momenta $(q_1, q_2)$ and species $(a, b)$ scatter into two QPs with identical momenta $(q_1, q_2)$ and species $(a', b')$. Note that for simplicity we are restricting ourselves to the case where the energy dispersion of the post-scattering QP species is the same of the energy dispersion of the pre-scattering species, viz. $\epsilon_a(k) = \epsilon_{a'}(k)$ and $\epsilon_b(k) = \epsilon_{b'}(k)$ for all values of momenta $k$, and hence, the momenta after the scattering are $q_1$ and $q_2$. $\mathcal{S}^{a'b'}_{ab}(q_1, q_2)$ is directly related to the scattering matrix of the system (see Eq. (123) and subsequent discussion). Defining $\tilde{\mathcal{S}}^{a'b'}_{ab}(q_1, q_2)$ as the amplitude for the on-shell scattering process under consideration —with the convention that $\tilde{\mathcal{S}} \equiv 0$ if the scattering process is not allowed— then

$$\mathcal{S}^{a'b'}_{ab}(q_1, q_2) = \tilde{\mathcal{S}}^{a'b'}_{ab}(q_1, q_2) - \delta^{a'}_a \delta^{b'}_b \,. \tag{26}$$

  The first factor in the second line of (24) involves the sum of two $\mathcal{S}$-terms because there are two different ways of annihilating the products of the scattering process: either $A(q_3, t_3)$ annihilates the QP with momentum $q_1$ and $A(q_4, t_4)$ annihilates the QP with momentum $q_2$ (first term in the bracket) or the opposite happens (second term). The factors of Kronecker delta, $\delta_{q_3, -q_1}$ and $\delta_{q_3, -q_2}$, express kinematic "resonance conditions" for the momentum $q_3$ in order to annihilate the relevant QP. We recall that the momentum $q_4 = q_1 + q_2 - q_3$ is always fixed by translational invariance.

- The final factor in the second line of (24) has a kinematic origin and gives rise to the long-time divergence of the 4-point function. The derivation of how this divergence arises is the main focus of this work.

Some comments are in order. First, the divergence in $t_{32}$ is a strong feature and should be easy to observe and identify in an experiment. Second, the fact that the divergent signal involves the scattering matrix raises the interesting possibility that $S$-matrices could be directly measured in non-linear response setups. Third, one may worry that the resonance conditions for the momentum $q_3$ requires fine-tuning in order to observe the linear in time growth. This is not the case and the result above is robust against small momenta mismatches in the following sense. For example, if $q_3$ differs slightly from $-q_1$ the factor $|t_{32}| \, \delta_{q_3, -q_1}$ in (24) is replaced replaced by

$$|t_{32}| \, \delta_{q_3, -q_1} \longrightarrow \frac{\sin(\alpha |t_{32}|)}{\alpha}, \qquad \alpha = |v_{a_1}(q_1) - v_{a_2}(q_2)| \frac{q_3 + q_1}{2} \ll 1 \,. \tag{27}$$

The resulting expression displays a linear increase in the time difference $|t_{32}|$ up to a large time scale $T \sim \alpha^{-1}$.

The results above are derived by means of a simple wave packet analysis presented in Sec. 6. We further benchmark Eq. (24) by carrying out exact calculations for certain integrable models. This is done by employing a Lehmann representation and analyzing its divergences. In the case of the transverse-field Ising model we can analyze all terms in the form factor expansion and verify that the result in Eq. (24) is the leading term in the long-time limit. This analysis is reported in Sec. 4. Finally we consider the case of interacting IQFTs with diagonal scattering matrix (i.e. $\tilde{S}_{ab}^{cd} \propto \delta_{a,c}\delta_{b,d}$) in Ref. 5. In this case, we identify certain terms in the form factor expansion that give rise to Eq. (24) and assume that other terms, which we are currently unable to analyze explicitly, only give subleading corrections in the long-time limit.

The late-time divergences in $C_4(\boldsymbol{q};\boldsymbol{t})$ directly translate into corresponding divergences in $\chi^{(3)}(\boldsymbol{q};\boldsymbol{t})$, *cf.* Eq. (7).

Finally, we note that this leading contribution vanishes when $v_{a_1}(q_1) = v_{a_2}(q_2)$ for all QP species $a_1$, $a_2$. A prominent instance of this is when $q_1 = q_2 = 0$ in a system symmetric under spatial reflection. In this case, in some integrable QFT it is known that the leading term grows like $\sqrt{|t_{32}|}$ instead [95,96]. Focusing on the simple case of the Ising model, we show in Sec. 4.3 that this result also has a simple origin: scattering and wavepacket spreading.

## 3.2 Two equal times

We now turn to the case where two times in the nonlinear susceptibility coincide. The leading contributions to $\chi_{\mathrm{NR}}^{(3)}(q_1, q_2; \tau_1, \tau_2)$ and $\chi_{\mathrm{PP}}^{(3)}(q_1, q_2; \tau_1, \tau_2)$ in the limit $\tau_{1,2} \to \infty$ arise, respectively, from $C_{\mathrm{NR}}$ and $C_{\mathrm{PP}}$ defined in (21). We note that both $C_{\mathrm{NR}}$ and $C_{\mathrm{PP}}$ are not time-ordered correlators, but should be thought of as path-ordered on the Keldysh contour, i.e. they involve processes where time flows forward from 0 to the measurement time $\tau_1 + \tau_2$, and then flows backward to time 0 again (see leftmost diagrams in Figs 2 and 3). This becomes clear if one rewrites the correlators in the Schrödinger picture, e.g.

$$C_{\mathrm{NR}}(q_1, q_2; \tau_1, \tau_2) = \frac{1}{V} \underbrace{\langle\Omega| U^\dagger(\tau_1) A(-q_2) U^\dagger(\tau_2)}_{\text{bra}} \underbrace{A(-q_1) U(\tau_2) A(q_2) U(\tau_1) A(q_1) |\Omega\rangle}_{\text{ket}}_C . \qquad (28)$$

Then the forward and backward branch of the Keldysh contour correspond to the ket and bra side, respectively.

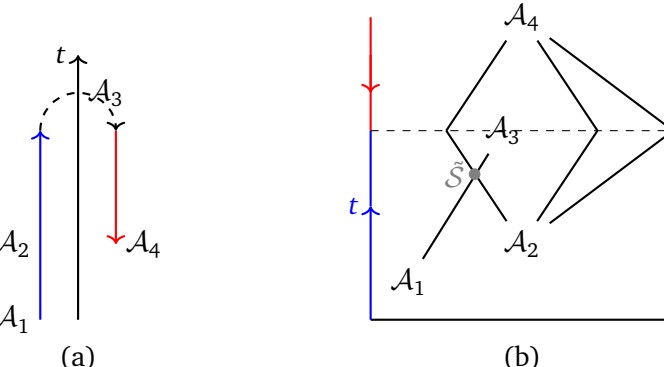

Figure 2: (a) $C_{\mathrm{NR}}$ involves amplitudes where the state is evolved forward in time from $t_1 = 0$ to $t_3 = \tau_1 + \tau_2$, and then evolves backwards to $t_4 = \tau_1$. (b) Processes that give rise to the leading contribution in the limit $\tau_{1,2} \to \infty$ limit. Here $\mathcal{A}_2 := \mathcal{A}(x_2, \tau_1)$ creates a shower of $n$ QPs ($n = 3$ in the figure), which spread ballistically and scatter with a QP exchanged between $\mathcal{A}_1$ and $\mathcal{A}_3$. When evolved backwards in time, the $n$ QPs refocus at the same position and are annihilated by $\mathcal{A}_4$.

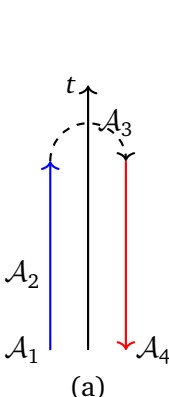
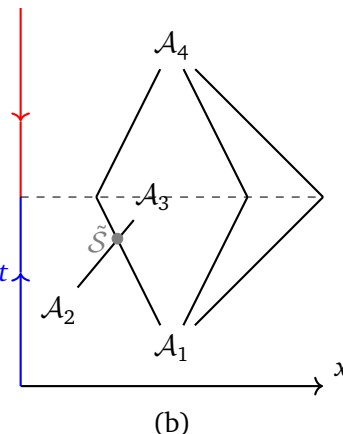

(a)                                        (b)

Figure 3: Left: $C_{PP}$ is a correlator where the state is evolved forward in time from $t_1 = 0$ to $t_3 = \tau_1 + \tau_2$, to finally be evolved backward to time $t_4 = 0$ again. (Right cartoon) Types of processes giving rise to the leading contribution in the $\tau_1, \tau_2 \to \infty$ limit. Here the first operator $\mathcal{A}_1 := \mathcal{A}(x_1, 0)$ can create a shower of $n$ QPs ($n = 3$ in the figure), which spread ballistically and can scatter against the QP exchanged between $\mathcal{A}_2$ and $\mathcal{A}_3$. When evolved backward in time, the $n$ QPs refocus at the same position and can be annihilated by $\mathcal{A}_4$.

The processes giving the leading contribution to $C_{NR}$ for $\tau_{1,2} \to \infty$. are such that the operators at time $\tau_1$ create/annihilate a shower of $n > 1$ QPs. As depicted in Fig. 2, these QPs propagate forward in time up to the "measurement time" $\tau_1 + \tau_2$, and then retrace their previous trajectory backwards until they are annihilated at $\tau_1$ in the backward branch in a neighborhood of the point at which they were created. During the propagation process they can scatter with a QP being exchanged between $A(-q_2, \tau_1 + \tau_2)$ and $A(q_2, 0)$, thus giving rise to a contribution again proportional to $\tau_2$.

The situation for $C_{PP}$ is quite similar, the only difference being that the shower of QPs is exchanged between operators at time $\tau_1$ in the forward and backward branch, *cf.* Fig. 3.

The long-time limits of $C_{NR}$ and $C_{PP}$ are obtained in sections 7.1 and 7.2 respectively and take the forms

$$C_{NR}(q_1, q_2; \tau_1, \tau_2) \sim \sum_a e^{-i\epsilon_a(q_1)(\tau_1 + \tau_2)} \alpha_{NR}^{(a)}(q_1, q_2) \tau_2, \tag{29}$$

$$C_{PP}(q_1, q_2; \tau_1, \tau_2) \sim \sum_a (\tau_1 + \tau_2) e^{-i\epsilon_a(q_2)\tau_2} \mathcal{C}_{PP}^{(a)}\left(\frac{\tau_2}{\tau_1 + \tau_2}; q_1, q_2\right), \tag{30}$$

where

$$\alpha_{NR}^{(a)}(q_1, q_2) = \mathcal{C}_{PP}^{(a)}(1; q_1, q_2). \tag{31}$$

The functions $\mathcal{C}_{PP}$ and $\alpha_{NR}$ depend on the details of the model considered. We have benchmarked the results of our wave packet analysis against exact calculations for the transverse field Ising model and in particular determined $\mathcal{C}_{PP}$ and $\alpha_{NR}$ in Secs. 7.1 and 7.2 respectively.

We stress that (29) and (30) are non-vanishing even if all momenta $q_j$ are taken to zero. This is in contrast to the leading result (24) analyzed in the previous subsection, which vanishes in this limit. Hence (29) and (30) are relevant for optical experiments that can be currently performed [4, 9, 10].

# 4 Benchmark I: Transverse-field Ising model

We start by considering the nonlinear response in the transverse-field Ising model. We utilize a form factor expansion in terms of exact energy and momentum eigenstates. A common thread of these calculations will be a careful analysis of individually divergent contributions that stem from "annihilation poles" in the form factor expansion; we will demonstrate that these divergences are eliminated in an appropriately defined connected correlator whose leading behaviour in the long-time limit coincides with the results of a semiclassical scattering analysis presented in sections 6 and 7.

The Hamiltonian of the transverse field Ising model is

$$H = -J\left(\sum_{j=0}^{L-1} \sigma_j^z \sigma_{j+1}^z + h\sum_{j=0}^{L-1} \sigma_j^x\right), \tag{32}$$

where we have imposed periodic boundary condition $\sigma_L^\alpha \equiv \sigma_0^\alpha$. The model can be diagonalized through a Jordan-Wigner transformation, which maps the spin Hamiltonian to a quadratic fermionic Hamiltonian. The model host only one QP species and the scattering matrix $S \equiv S(k_1, k_2; k_1', k_2') = -1$ just encodes for the exchange statistics of the fermionic QPs.

For the semiclassical picture developed below to apply we need to perturb the system by coupling to some operator connecting the ground state with the 1-QP sector. For this purpose we consider the paramagnetic phase of the chain ($h > 1$) and four-point correlators of the order parameter

$$\mathcal{A}(j) = \sigma_j^z. \tag{33}$$

Note that, under the Jordan-Wigner transformation $\mathcal{A}$ is mapped to a non-local fermionic operator, thus computation of four-point functions is non-trivial [97] and is best tackled through a form factor expansion.

The eigenstates of the system can be labelled by the number of fermionic QPs and their momenta. Furthermore, we need to distinguish between two sectors of the theory depending on the parity of the number of QPs. For odd number of QPs, the state lies in the Ramond sector

$$|k_1, \ldots, k_{2m+1}\rangle_\mathrm{R}, \; k_j = \frac{2\pi}{L} n_j, \quad n_j \in \left\{-\frac{L}{2}, \ldots \frac{L}{2} - 1\right\}, \tag{34}$$

where, for simplicity, we assumed that $L$ is even. Conversely, for even number of QPs, the state lies in the Neveu-Schwartz sector

$$|p_1, \ldots, p_{2n}\rangle_\mathrm{NS}, \; p_j = \frac{2\pi}{L}\left(n_j + \frac{1}{2}\right), \quad n_j \in \left\{-\frac{L}{2}, \ldots \frac{L}{2} - 1\right\}. \tag{35}$$

It is useful to introduce some shorthand notations for the above eigenstates and their energies and momenta

$$|\boldsymbol{k}\rangle_a, \quad a = \mathrm{R}, \mathrm{NS}, \quad E_{\boldsymbol{k}} = \sum_j \epsilon(k_j), \quad P_{\boldsymbol{k}} = \sum_j k_j, \quad \epsilon(k) = 2J\sqrt{1 + h^2 - 2h\cos(k)}. \tag{36}$$

We further introduce the sets of momenta for $m$-particle states

$$\begin{aligned} M_{2n} &= \{p_1, \ldots p_{2n}\}, \\ M_{2n-1} &= \{k_1, \ldots k_{2n-1}\}. \end{aligned} \tag{37}$$

### 4.1 Connected 4-point function

We now expand the four-point function $\tilde{C}_4$ (including disconnected parts) in a Lehmann representation in terms of exact energy eigenstates

$$\tilde{C}_4(\boldsymbol{q},t) = \frac{1}{L}\,_{\mathrm{NS}}\langle\Omega|\prod_{j=4}^{1}\sigma^z(q_j,t_j)|\Omega\rangle_{\mathrm{NS}} = \sum_{n_1,n_2,n_3\geq 0}\tilde{C}_4^{(2n_1+1,2n_2,2n_3+1)}(\boldsymbol{q},t), \qquad (38)$$

where

$$\tilde{C}_4^{(n)}(\boldsymbol{q},t) = L^3\sum_{\substack{k\in M_{n_1}\\ p\in M_{n_2}\\ K\in M_{n_3}}}\frac{\mathcal{F}(\boldsymbol{K},\boldsymbol{p},\boldsymbol{k})}{n_1!n_2!n_3!}e^{-it_{21}E_k-it_{32}E_p-it_{43}E_K}\Delta(\boldsymbol{q}|\boldsymbol{K},\boldsymbol{p},\boldsymbol{k}). \qquad (39)$$

Here we have defined

$$\mathcal{F}(\boldsymbol{K},\boldsymbol{p},\boldsymbol{k}) = \,_{\mathrm{NS}}\langle\Omega|\sigma_0^z|\boldsymbol{K}\rangle_{\mathrm{R}}\,_{\mathrm{R}}\langle\boldsymbol{K}|\sigma_0^z|\boldsymbol{p}\rangle_{\mathrm{NS}}\,_{\mathrm{NS}}\langle\boldsymbol{p}|\sigma_0^z|\boldsymbol{k}\rangle_{\mathrm{R}}\,_{\mathrm{R}}\langle\boldsymbol{k}|\sigma_0^z|\Omega\rangle_{\mathrm{NS}},$$
$$\Delta(\boldsymbol{q}|\boldsymbol{K},\boldsymbol{p},\boldsymbol{k}) = \delta_{q_1,P_k}\delta_{q_1+q_2,P_p}\delta_{-q_4,P_K}. \qquad (40)$$

The form factors of $\sigma_0^z$ are given by [98–101] and up to exponentially small corrections in the number of sites $L$ take the form

$$\,_{\mathrm{NS}}\langle p_1,\ldots,p_{2n}|\sigma_0^z|k_1,\ldots,k_{2m+1}\rangle_{\mathrm{R}} = i^{n+m}(4J^2h)^{\frac{(2m+1-2n)^2}{4}}|1-h^2|^{\frac{1}{8}}\prod_{j=1}^{2n}\Big(\frac{1}{L\epsilon(p_j)}\Big)^{\frac{1}{2}}$$

$$\times\prod_{l=1}^{2m+1}\Big(\frac{1}{L\epsilon(k_l)}\Big)^{\frac{1}{2}}\prod_{j<j'}^{2n}\frac{2\sin\frac{p_j-p_{j'}}{2}}{\epsilon(p_j)+\epsilon(p_{j'})} \qquad (41)$$

$$\times\prod_{l<l'}^{2m+1}\frac{2\sin\frac{k_l-k_{l'}}{2}}{\epsilon(k_l)+\epsilon(k_{l'})}\prod_{j=1}^{2n}\prod_{l=1}^{2m+1}\frac{\epsilon(p_j)+\epsilon(k_l)}{2\sin\frac{p_j-k_l}{2}}.$$

Of crucial importance to the following discussion will be the annihilation pole structure of the form factors: for $k_i - p_j \to 0$

$$\,_{NS}\langle p_1,\ldots,p_{2n}|\sigma_0^z|k_1,\ldots,k_{2m+1}\rangle_R \propto \frac{1}{k_i-p_j}. \qquad (42)$$

#### 4.1.1 Identifying divergences through power counting

Following Refs. [78, 102–109] we want to identify possible divergences in the long-time behaviour of $\tilde{C}_4^{(n)}(\boldsymbol{q},t)$. For the analysis in this paragraph it is convenient to work in terms of frequencies rather than times directly. The idea is that long-time divergences transform to singularities in frequency space, which are easier to identify. We thus define

$$\hat{C}_4^{(n)}(\boldsymbol{q},\boldsymbol{\omega}) = \int_{-\infty}^{+\infty}\Big(\prod_{j=1}^{3}dt_{j+1,j}\,e^{i\omega_j t_{j+1,j}}\Big)\tilde{C}_4^{(n)}(\boldsymbol{q},t), \qquad (43)$$

with $\boldsymbol{\omega} = (\omega_1,\omega_2,\omega_3)$. The form factor expansion for $\hat{C}_4^{(n)}$ is obtained from the one for $\tilde{C}_4^{(n)}$ by the replacement

$$e^{-it_{21}E_k-it_{32}E_p-it_{43}E_K} \longrightarrow \delta(\boldsymbol{\omega}|\boldsymbol{K},\boldsymbol{p},\boldsymbol{k}) \equiv (2\pi)^3\delta(\omega_1-E_k)\delta(\omega_2-E_p)\delta(\omega_3-E_K). \qquad (44)$$

We observe that singularities at a given point $\omega$ can arise through a combination of two mechanisms. (i) When $\Delta(\boldsymbol{q}|\boldsymbol{K},\boldsymbol{p},\boldsymbol{k})$ is enforced, the sum of $\delta(\omega|\boldsymbol{K},\boldsymbol{p},\boldsymbol{k})$ is divergent, viz. the density of states is singular at $\omega$, once the total momenta $q_{1,2,3}$ are fixed. (ii) There is a momentum region contributing to the neighbourhood of $\omega$, such that the sum over momenta of $\mathcal{F}(\boldsymbol{K},\boldsymbol{p},\boldsymbol{k})\Delta(\boldsymbol{q}|\boldsymbol{K},\boldsymbol{p},\boldsymbol{k})$ diverges as $L \to \infty$. As we will see, such a divergence can be produced by annihilation poles.

Regarding (i), in 1D the density of states is divergent at the onset of 2QP continuum only. Here (i) can produce divergences only for $\boldsymbol{n} = (1,2,1)$ when $q_1 = q_2 = -q_3 = -q_4$; this case has been already studied in Ref. [95,96] and, as we will further discuss in Sec. 4.3 produces a contribution proportional to $\sqrt{|t_{32}|}$. Since we are interested in terms proportional to $|t_{32}|$, in the following we focus our attention on divergences of $\mathcal{F}(\boldsymbol{K},\boldsymbol{p},\boldsymbol{k})\Delta(\boldsymbol{q}|\boldsymbol{K},\boldsymbol{p},\boldsymbol{k})$ produced by annihilation poles.

We start by considering a generic set of momenta, far away from annihilation poles. In the limit of large $L$, then $\mathcal{F}(\boldsymbol{K},\boldsymbol{p},\boldsymbol{k})\Delta(\boldsymbol{q}|\boldsymbol{K},\boldsymbol{p},\boldsymbol{k}) = O(L^{-n_1-n_2-n_3+3})$. Focusing on a region in momentum space of $O(1)$ volume, this will contain $O(L^{n_1+n_2+n_3-3})$ points that contribute to the sum. Thus, as expected, the powers of $L$ cancel and a generic momentum region gives an $O(1)$ contribution to $\tilde{C}_4$.

Next we want to understand how this is modified if the momentum region encompasses one annihilation pole. E.g. suppose that $k_1 - p_1 = O(1/L)$, while other momenta are otherwise generic and far from annihilation poles. In this case $\mathcal{F}(\boldsymbol{K},\boldsymbol{p},\boldsymbol{k})$ will increase by a factor $L$, however the condition $k_1 - p_1 = O(1/L)$ reduces the number of points in the momentum region under consideration by a factor $L$. Therefore, also this momentum region would give an overall $O(1)$ condition. Generalizing the argument above, we see that imposing proximity to $m$ annihilation poles will increase the magnitude of $\mathcal{F}(\boldsymbol{K},\boldsymbol{p},\boldsymbol{k})$ by $O(L^m)$, but the number of points that satisfy the proximity condition to the $m$ annihilation poles are reduced by a factor $O(L^{-m})$ w.r.t. a generic region. In summary, if we are considering a momentum region proximate to any number of poles, this region will not produce divergences in $\tilde{C}_4$ if the proximity to every annihilation pole must be independently enforced.

This argument can be evaded only if, after imposing the proximity to $m$ annihilation poles — thus reducing the number of points to $O(L^{-m})$— the momentum-conservation conditions in $\Delta(\boldsymbol{q}|\boldsymbol{K},\boldsymbol{p},\boldsymbol{k})$ force the proximity to other $m_e$ annihilation poles. In this case the overall contribution to $\tilde{C}_4$ would be $O(L^{m_e})$.

It is then immediate to see that for $m_e$ to be greater than zero, all momenta $k_j$ and $K_j$ must be approximately fixed by the proximity to a pole. In particular, there are only two possible cases in which $m_e > 0$. (i) $\boldsymbol{n} = (m, n+m, n)$, $q_3 \simeq -q_1$, and, up to permutations, $\boldsymbol{p}$ can be split into two set of momenta $\boldsymbol{p} = (\boldsymbol{p}^3, \boldsymbol{p}^1)$ with $\boldsymbol{p}^1 \simeq \boldsymbol{k}$ and $\boldsymbol{p}^3 \simeq \boldsymbol{K}$. (ii) $\boldsymbol{n} = (n, n+m, n)$, $q_3 \simeq -q_2$, and, up to permutations, $\boldsymbol{p}$ can be split into two set of momenta $\boldsymbol{p} = (\boldsymbol{p}^1, \boldsymbol{p}^2)$ with $\boldsymbol{K} \simeq \boldsymbol{p}^1 \simeq \boldsymbol{k}$. In both cases $m_e = 1$, thus the overall contribution would be a singularity $O(L)$.

Cases (i) and (ii) should be supplemented with the extra case (iii): $n_2 = 0$ and $q_2 = -q_1$, which immediately yields an $O(L)$ contribution due to the automatic enforcement of $P_{\boldsymbol{p}} = 0$ in $\Delta(\boldsymbol{q}|\boldsymbol{K},\boldsymbol{p},\boldsymbol{k})$.

### 4.1.2 Disconnected components

The possible divergences identified in the previous section arise in proximity of disconnected contributions to the 4-point function. We will now show that the latter give rise to divergent terms of $\mathcal{O}(L)$, which ensure that the connected four-point function

$$
\begin{aligned}
C_4 = \tilde{C}_4 &- L^{-1} \langle\Omega|A(q_4,t_4)A(q_3,t_3)|\Omega\rangle \langle\Omega|A(q_2,t_2)A(q_1,t_1)|\Omega\rangle \\
&- L^{-1} \langle\Omega|A(q_4,t_4)A(q_2,t_2)|\Omega\rangle \langle\Omega|A(q_3,t_3)A(q_1,t_1)|\Omega\rangle \\
&- L^{-1} \langle\Omega|A(q_3,t_3)A(q_2,t_2)|\Omega\rangle \langle\Omega|A(q_4,t_4)A(q_1,t_1)|\Omega\rangle ,
\end{aligned}
\tag{45}
$$

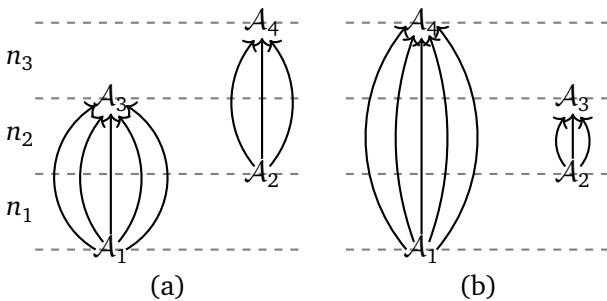

Figure 4: The form factor expansion of a disconnected contribution would give rise to terms where $\boldsymbol{n}$ is either of the form (i) $(m, n+m, n)$ (as in panel (a) where $n = 5$, $m = 3$), (ii) $(n, n+m, n)$ (as in panel (b) where $n = 5$, $m = 3$), or (iii) $(m, 0, n)$.

remains finite in the thermodynamic limit (we have assumed that the one-point function vanishes). Above we have expressed the four point function as

$$\tilde{C}_4(\boldsymbol{q}, t) = \sum_{n_1, n_2, n_3 \geq 0} \tilde{C}_4^{(n)}(\boldsymbol{q}, t). \tag{46}$$

The divergences of $\tilde{C}_4^{(n)}(\boldsymbol{q}, t)$ with system size are eliminated by focusing only on the connected contributions in (45). To see this we consider

$$C_4^{(m,n+m,n)}(\boldsymbol{q}, t) = \tilde{C}_4^{(m,n+m,n)}(\boldsymbol{q}, t) - L\delta_{q_3,-q_1} C_2^{(n)}(q_1, t_{31}) C_2^{(m)}(q_2, t_{42}), \tag{47}$$

$$C_4^{(n,n+m,n)}(\boldsymbol{q}, t) = \tilde{C}_4^{(m,n+m,n)}(\boldsymbol{q}, t) - L\delta_{q_4,-q_1} C_2^{(n)}(q_1, t_{41}) C_2^{(m)}(q_2, t_{32}), \tag{48}$$

$$C_4^{(1,2,1)}(\boldsymbol{q}, t) = \tilde{C}_4^{(1,2,1)}(\boldsymbol{q}, t) - L\delta_{q_3,-q_1} C_2^{(1)}(q_1, t_{31}) C_2^{(1)}(q_2, t_{42})$$
$$- L\delta_{q_4,-q_1} C_2^{(1)}(q_1, t_{41}) C_2^{(1)}(q_2, t_{32}), \tag{49}$$

$$C_4^{(m,0,n)}(\boldsymbol{q}, t) = 0, \tag{50}$$

where $C_2^{(n)}(q, t)$ denotes the $n-$QP contribution to the two-point function in a form factor expansion.

In the following we demonstrate that (47)-(49) indeed eliminates all divergences in $L$ and determine the leading term in the long-time limit, which coincides with the semiclassical scattering-connected contributions identified in Sec. 6.

### 4.1.3  Contribution $C_4^{(1,2,1)}(q, t)$

Following the previous discussion, we consider $\tilde{C}_4^{(1,2,1)}$. For concreteness in the following we focus on the case $q_3 \simeq -q_1$ and $q_4 \simeq -q_2$. Similar results would apply in the case $q_3 \simeq -q_2$ and $q_4 \simeq -q_1$. We expand the contribution due to kinematic poles, and, after imposing the momentum constraints, we are left with only one unconstrained momentum $p := p_1$

$$\tilde{C}_4^{(1,2,1)}(\boldsymbol{q}, t) \simeq C_2^{(1)}(q_1, t_{31}) C_2^{(1)}(-q_4, t_{42}) e^{i(\epsilon(q_1) + \epsilon(-q_4))t_{32}}$$
$$\times \frac{1}{L} \sum_{p \in \text{NS}} \frac{1}{2} \left[ \frac{e^{-it_{32}(\epsilon(p) + \epsilon(q_1 + q_2 - p))}}{\sin \frac{p-q_1}{2} \sin \frac{p+q_3}{2}} + \frac{e^{-it_{32}(\epsilon(p) + \epsilon(q_1 + q_2 - p))}}{\sin \frac{p-q_2}{2} \sin \frac{p+q_4}{2}} \right], \tag{51}$$

where $C_2^{(1)}(q, t) = (4J^2 h)^{1/2} |1 - h^2|^{1/4} (\epsilon(q))^{-1} e^{-it\epsilon(q)}$ and, in the long-time limit, $C_2^{(1)}(q, t) \simeq C_2(q, t)$, as discussed in Sec 6.

If $q_1 = -q_3$ exactly, we can use Eq. (C.1) to extract the leading behaviour in the $L \to \infty$ and large times limit. After subtracting the disconnected component we then have

$$C_4^{(1,2,1)}(\boldsymbol{q}, t) = -2C_2^{(1)}(q_1, t_{31})C_2^{(1)}(q_2, t_{42})|t_{32}||v(q_1) - v(q_2)| + O(t_{32}^0). \tag{52}$$

If $q_1 \neq -q_3$, we can instead use Eq. (C.7) to extract the leading behavior in the $L \to \infty$ and $q_3 + q_1 \to 0$ limit (at fixed, large $|t_{32}|$), thus obtaining

$$
\begin{aligned}
C_4^{(1,2,1)}(\boldsymbol{q}, t) = {} & -2C_2^{(1)}(q_1, t_{31})C_2^{(1)}(q_2, t_{42}) \ \mathrm{sign}(t_{32}(v(q_2) - v(q_1))) \\
& \times \frac{1 - e^{-it_{32}[\epsilon(-q_3) + \epsilon(-q_4) - \epsilon(q_1) - \epsilon(q_2)]}}{i(q_1 + q_3)} + \mathcal{O}((q_1 + q_3)^0).
\end{aligned} \tag{53}
$$

To leading order in $q_1 + q_3 \to 0$ the exact expressions we just derived agree with the semiclassical picture in Eq. (147), with $\mathcal{S} = -2$.

### 4.1.4 Contributions involving larger numbers of QPs

Next, we consider divergent contributions from more-QP states, i.e. $\boldsymbol{n} = (m, n+m, n)$ and $\boldsymbol{n} = (n, n+m, n)$. We show in Sec 4.2 that, once the disconnected part is cancelled, they can be bounded by the product of two-point functions times a term proportional to the overall timescale. More specifically

$$C_4^{(m,n+m,n)} = C_2^{(n)}(q_1, t_{31})C_2^{(m)}(q_2, t_{42})O(|t_{32}|), \tag{54}$$

$$C_4^{(n,n+m,n)} = C_2^{(n)}(q_1, t_{41})C_2^{(m)}(q_2, t_{32})O(|t_{32}| + |t_{21}|). \tag{55}$$

The form factor expansion of $C_2^{(n)}(q, t)$ contains a sum over $n-1$ momenta. Since the dependence over these momenta is smooth, the sum can be converted into an integral in $n-1$ dimension. In the limit of large $t$, the integral can thus be bounded through a stationary phase approximation, that yields $C_2^{(n)}(q, t) = O(t^{-(n-1)/2})$. We can therefore bound the overall contributions as

$$C_4^{(m,n+m,n)} = O\left(\frac{|t_{32}|}{|t_{31}|^{(n-1)/2}|t_{42}|^{(m-1)/2}}\right), \tag{56}$$

$$C_4^{(n,n+m,n)} = O\left(\frac{|t_{32}| + |t_{21}|}{|t_{41}|^{(n-1)/2}|t_{32}|^{(m-1)/2}}\right). \tag{57}$$

Thus, when all times are well separated, the leading contribution is given by $C_4^{1,2,1}$, which, as discussed in the previous subsections, agrees with the results derived through the WP analysis in Sec. 6.

## 4.2 Non-rephasing and pump-probe response functions

Our analysis of the structure of the form factor expansion in the Ising model revealed that proximity to any disconnected contribution can produce divergences in the long-time limit. In the case where all $|t_{ij}| \to \infty$ we observed that the leading behavior was given by $C_4^{(1,2,1)}(\boldsymbol{q}, t)$. In principle, there are two additional families of terms that could give rise to further divergences in the long-time limit: $C_4^{(m,n+m,n)}(\boldsymbol{q}, t)$ for $q_3 \simeq -q_1$, and $C_4^{(n,n+m,n)}(\boldsymbol{q}, t)$ for $q_3 \simeq -q_2$.

We now consider these two families of contributions and extract their leading long-time behaviour. The main conclusion we will obtain from this analysis are the following.

1. In the limit $|t_{ij}| \to \infty \ \forall i \neq j$, we will show that the bounds anticipated in Eqs. (55) and (55) hold.

2. The sum of terms of the form $C_4^{(m,1+m,1)}(\boldsymbol{q},\boldsymbol{t})$ give the leading contribution to $C_{\mathrm{NR}}$, and will confirm the result obtained in section 7.1 by studying the WP dynamics.

3. The sum of terms of the form $C_4^{(n,1+n,n)}(\boldsymbol{q},\boldsymbol{t})$ give the leading contribution to $C_{\mathrm{PP}}$ and will allow us to recover the WP results presented in section 7.2.

We set up the form factor calculation for generic times and only later restrict ourselves to $C_{\mathrm{NR}}$ and $C_{\mathrm{PP}}$. Additionally, it will be useful to split the two limits of $C_4$: near $q_3 = -q_1$ or near $q_3 = -q_2$. We denote the limit of $C_4$ in these two cases as $C_{\mathrm{NR}'}$ and $C_{\mathrm{PP}'}$ respectively — they have the same form of $C_{\mathrm{NR}}$ and $C_{\mathrm{PP}}$, but for arbitrary times. Finally, for technical reasons, it will be convenient to add an extra regulator and define

$$C_{\mathrm{NR}',\gamma} = \sum_{q_1' \in \mathrm{R}} \frac{1}{L^2} \mathcal{L}(q_1 - q_1') \; \langle\Omega|A(-q_2',t_4)A(-q_1',t_3)A(q_2,t_2)A(q_1,t_1)|\Omega\rangle \,, \tag{58}$$

$$C_{\mathrm{PP}',\gamma} = \sum_{q_1' \in \mathrm{R}} \frac{1}{L^2} \mathcal{L}(q_1 - q_1') \; \langle\Omega|A(-q_1',t_4)A(-q_2',t_3)A(q_2,t_2)A(q_1,t_1)|\Omega\rangle \,, \tag{59}$$

where $q_2'$ is fixed by momentum conservation to $q_2' = q_1 + q_2 - q_1'$ and

$$\mathcal{L}(q) = \frac{\gamma\sqrt{1+\gamma^2}}{\gamma^2 + \sin\left(\frac{q}{2}\right)^2} \,. \tag{60}$$

The idea is that instead of constraining the momenta to a precise value, we average over the values of $q_1'$ within a window of length $\gamma$ inside the region giving rise to divergent contributions. We will perform the form factor calculation in a regime where $L \gg \gamma^{-1} \gg t_{ij}$ and finally take the limit $\lim_{\gamma\to 0} \lim_{L\to\infty}$. Note that the normalization is chosen such that

$$\lim_{L\to\infty} \frac{1}{L} \sum_{q\in\mathrm{R}} \mathcal{L}(q - q') = 1 \,. \tag{61}$$

This implies that

$$\lim_{\gamma\to 0} \lim_{L\to\infty} \frac{1}{L} \sum_{q\in\mathrm{R}} \mathcal{L}(q - q')A(q') = A(q) \,, \tag{62}$$

and consequently

$$C_{\mathrm{NR}'} = \lim_{\gamma\to 0} \lim_{L\to\infty} C_{\mathrm{NR}',\gamma} \,, \qquad C_{\mathrm{PP}'} = \lim_{\gamma\to 0} \lim_{L\to\infty} C_{\mathrm{PP}',\gamma} \,. \tag{63}$$

Finally, while the times of $C_{\mathrm{NR}',\gamma}$ and $C_{\mathrm{PP}',\gamma}$ are arbitrary, we can study the non-rephasing and pump-probe correlator through the substitution

$$C_{\mathrm{NR}}(q_1,q_2;\tau_1,\tau_2) = \lim_{\gamma\to 0} \lim_{L\to\infty} C_{\mathrm{NR}',\gamma}(q_1,q_2,\boldsymbol{t})\Big|_{\substack{t_{42}=0 \\ t_{32}=\tau_2 \\ t_{21}=\tau_1}} \,, \tag{64}$$

$$C_{\mathrm{PP}}(q_1,q_2;\tau_1,\tau_2) = \lim_{\gamma\to 0} \lim_{L\to\infty} C_{\mathrm{PP}',\gamma}(q_1,q_2,\boldsymbol{t})\Big|_{\substack{t_{41}=0 \\ t_{32}=\tau_2 \\ t_{21}=\tau_1}} \,. \tag{65}$$

We note that since all the timescales are kept finite while taking the limits $L \to \infty$ and $\gamma \to 0$, the time substitution can be made at any point. In particular, we will take the limits first and then replace the times $t_j$ in terms of $\tau_{1,2}$.

### 4.2.1 Non-rephasing susceptibility and $C^{(m,n+m,n)}(q_1, q_2, t)$

As emerged from the discussion in section 4.1.1, long-time divergences arise when the momenta of the various QP in the form factor expansion are proximate to the momenta of a disconnected process. For $C_{NR'}$ this can happen in terms of the form $C_4^{(m,n+m,n)}$. The disconnected component of the four point-function takes contributions from the momenta where, up to internal permutations of $p$, $p \approx (k, K) \equiv \kappa$. Expanding $\mathcal{F}(K, p, k)$ around $p \approx \kappa$ using the explicit form (41) of the form factors gives

$$\mathcal{F}(K, p, k) \approx \left| {}_{NS}\langle\Omega|\sigma_0^z|K\rangle_R \right|^2 \left| {}_{NS}\langle\Omega|\sigma_0^z|k\rangle_R \right|^2 \frac{i^{n+m}}{L^{n+m}} \prod_{\ell=1}^{n+m} \frac{1}{\sin\left(\frac{p_\ell - \kappa_\ell}{2}\right)}. \tag{66}$$

The singular contributions to the four-point function $C_{NR',\gamma}$ can then be expressed as

$$C_{NR',\gamma}(q_1, q_2, t) = \sum_{n,m} C_{NR',\gamma}^{(m,n+m,n)}(q_1, q_2, t),$$

$$C_{NR',\gamma}^{(m,n+m,n)}(q_1, q_2, t)\Big|_{sing} = \sum_{k \in M_n^R} \frac{1}{n!} L\delta_{q_1,P_k} \left| {}_{NS}\langle\Omega|\sigma_0^z|k\rangle_R \right|^2 e^{-iE_k t_{31}} \tag{67}$$

$$\times \sum_{K \in M_m^R} \frac{1}{m!} L\delta_{q_2,P_K} \left| {}_{NS}\langle\Omega|\sigma_0^z|K\rangle_R \right|^2 e^{-iE_K t_{42}} H_{NR}(t_{32}; \kappa),$$

where $M_n^a$ denotes the n-dimensional momentum grid in the a=R/NS-sector and

$$H_{NR}(t; \kappa) = \widetilde{H}_{NR}(t; \kappa) - \mathcal{L}(0),$$

$$\widetilde{H}_{NR}(t; \kappa) = \frac{i^{n+m}}{L^{n+m}} \sum_{p \in M_{n+m}^{NS}} \prod_{j=1}^{n+m} \frac{1}{\sin\left(\frac{p_j - \kappa_j}{2}\right)} \mathcal{L}(P_p - P_\kappa) e^{-it(E_p - E_\kappa)}. \tag{68}$$

Eqn (68) follows from the subtraction scheme in Eq. (47), where the disconnected contribution is now weighted by $\mathcal{L}(0)$, as a consequence of the $\gamma$-regularization. Finally, we anticipate that in the first equation the dependence on $k$ and $K$ is regular, therefore the sums over momenta can be approximated by integrals in the $L \to \infty$ limit.

Next, we need to compute $H_{NR}$. As $H_{NR}(t; \kappa)$ is invariant under the permutation of the elements in $\kappa$ we may reorder them such that $v(\kappa_{S_1}) < \cdots < v(\kappa_{S_{n+m}})$. It is shown in Appendix D that

$$\lim_{\gamma \to 0} \lim_{L \to \infty} \sum_\kappa H_{NR}(t, \kappa) f(\kappa) = \lim_{L \to \infty} \sum_\kappa \left[ -2t \sum_{j=1}^{n+m} (-1)^j v(\kappa_{S_j}) \right] f(\kappa) + \dots \tag{69}$$

From this result, we can easily obtain everything we set out to achieve. (i) Since the velocities are bounded we have

$$\lim_{\gamma \to 0} \lim_{L \to \infty} H_{NR} = O(|t_{32}|). \tag{70}$$

Plugging this into the expression for $C_{NR',\gamma}^{(m,n+m,n)}$ above we immediately obtain the bound in Eq. (54). (ii) Furthermore, when considering $C_{NR}$, from the same bound we find that

$$C_{NR',\gamma \to 0}^{(m,n+m,n)}\Big|_{\substack{t_{21}=\tau_1 \\ t_{32}=\tau_2 \\ t_{42}=\tau_0}} = C_2^{(m)}(q_2, 0) C_2^{(n)}(q_1, \tau_1 + \tau_2) O(\tau_2). \tag{71}$$

From this it follows that the contributions with $n > 1$ are subleading w.r.t. the ones with $n = 1$. To show this we can evaluate $C_2^{(n)}(q_1, \tau_1 + \tau_2)$ within the stationary-phase approximation to obtain that $C_2^{(n)}(q_1, \tau_1 + \tau_2) = O((\tau_1 + \tau_2)^{(n-1/2)})$, thus reaching the conclusion that

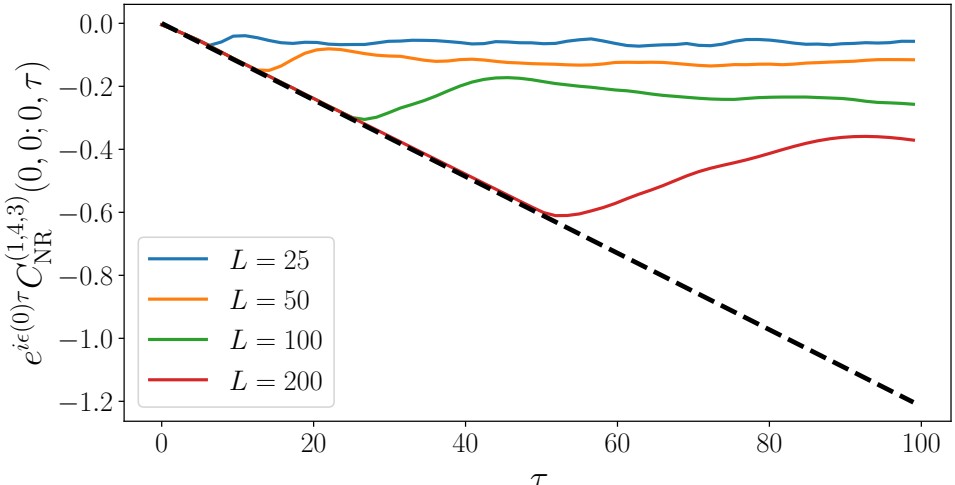

Figure 5: $C_{\text{NR}}^{(1,4,3)}$ as obtained by numerically summing over momenta at finite size $L$. A black dashed line reports the WP prediction, to which the finite size data converges in the infinite volume limit.

contributions terms with $n > 1$ are subleading in the large-$\tau_1, \tau_2$ limit. Furthermore, we can immediately note that, when $n = 1$

$$\lim_{\gamma \to 0} \lim_{L \to \infty} H_{\text{NR}} \Big|_{\substack{t_{21}=\tau_1 \\ t_{32}=\tau_2 \\ t_{42}=\tau_0}} = \int dr_1 \, \mathfrak{S} \,, \tag{72}$$

with the RHS explicitly reported in Eq. (163). From this it follows that, by summing the $n = 1$ terms in the form factor expansion, we can recover the semiclassical results in Sec. 7.1.

As a further check, we have explicitly performed the sum over momenta numerically in some simple cases. In Fig. 5 we report the numerical behaviour of $C_{\text{NR}}^{(1,4,3)}$ for $J = 1$ and $h = 2$ at various system sizes $L$, clearly indicating a convergence to the WP result (black dashed line) in the thermodynamic limit.

### 4.2.2 Pump-probe susceptibility and $C^{(n,n+m,n)}(q_1, q_2, t)$

The discussion for the pump-probe signal follows closely the one for the non-rephasing one. From the power-counting argument in Subsec. 4.1.1, we expect long-time divergences to arise in proximity of disconnected contributions. For terms of the form $C^{(n,n+m,n)}(q_1, q_2, t)$ this takes place if the momenta of the first and third state approximately coincide, i.e. up to permutations $K \simeq k$, and a subset of the momenta of the second state approximately coincides with $k$, i.e. $p \approx (k, p')$. In this case, we can expand the product of form factors as

$$\mathcal{F}(K, p, k) \approx \Big|_{\text{NS}} \langle \Omega | \sigma_0^z | k \rangle_{\text{R}} \Big|^2 \Big|_{\text{NS}} \langle \Omega | \sigma_0^z | p' \rangle_{\text{R}} \Big|^2 \frac{1}{L^{2n}} \prod_{\ell=1}^{n} \frac{1}{\sin\left(\frac{p_\ell - k_\ell}{2}\right) \sin\left(\frac{p_\ell - K_\ell}{2}\right)} \,. \tag{73}$$

We express the singular contributions to the four-point function $C_{\text{PP},\gamma}$ as

$$C_{\text{PP},\gamma}(q_1, q_2, t) = \sum_{n,m} C_{\text{PP}',\gamma}^{(n,n+m,n)}(q_1, q_2, t) \,, \tag{74}$$



$$C_{\mathrm{PP},\gamma}^{(n,n+m,n)}(q_1,q_2,\boldsymbol{t})\Big|_{\mathrm{sing}} = \sum_{\boldsymbol{k}\in M_n^R} \frac{L\delta_{q_1,P_{\boldsymbol{k}}}}{n!}\Big|_{\mathrm{NS}}\langle\Omega|\sigma_0^z|\boldsymbol{k}\rangle_{\mathrm{R}}\Big|^2 e^{-iE_{\boldsymbol{k}}t_{41}}$$

$$\times \sum_{\boldsymbol{p}'\in M_m^R}\frac{1}{n!}\Big|_{\mathrm{NS}}\langle\Omega|\sigma_0^z|\boldsymbol{p}'\rangle_{\mathrm{R}}\Big|^2 e^{-iE_{\boldsymbol{p}'}t_{32}}H_{\mathrm{PP}}(\boldsymbol{t};\boldsymbol{k},\boldsymbol{p}'), \tag{75}$$

where

$$H_{\mathrm{PP}}(\boldsymbol{t};\boldsymbol{k},\boldsymbol{p}') = \frac{1}{L^{2n-1}}\sum_{\substack{\boldsymbol{p}\in M_n^{\mathrm{NS}}\\ \boldsymbol{K}\in M_n^R}}\prod_{j=1}^n \frac{e^{-it_{21}\epsilon(K_j)-it_{32}\epsilon(p_j)+it_{31}\epsilon(k_j)}}{\sin\left(\frac{p_j-k_j}{2}\right)\sin\left(\frac{p_j-K_j}{2}\right)}\mathcal{L}_{\gamma_1}(P_{\boldsymbol{k}}-P_{\boldsymbol{K}})\delta_{q_2+q_1,P_{\boldsymbol{p}}+P_{\boldsymbol{p}'}}$$

$$-\mathcal{L}_\gamma(0)L\delta_{P_{\boldsymbol{p}'},q_2}. \tag{76}$$

We now simplify the first contribution in $H_{\mathrm{PP}}(\boldsymbol{t};\boldsymbol{k},\boldsymbol{p}')$ using the case $n=m=1$ as guidance. The sum over $p_1'$ can be carried out using the momentum conservation Kronecker delta. This fixes

$$p_1' = q_1 + q_2 - P_{\boldsymbol{p}} - \sum_{j=2}^m p_j'. \tag{77}$$

As the leading contribution we are interested in arises from $p_j \approx k_j$ we then replace (77) by $p_1' = q_2 - \sum_{j=2}^m p_j'$ in the (non-singular) form factor ${}_{\mathrm{NS}}\langle\Omega|\sigma_0^z|\boldsymbol{p}'\rangle_{\mathrm{R}}$, but retain (77) in the oscillating factors. This results in the replacement

$$H_{\mathrm{PP}}(\boldsymbol{t};\boldsymbol{k},\boldsymbol{p}') \to H_{\mathrm{PP}}(\boldsymbol{t};\boldsymbol{k},p_1')L\delta_{q_2,P_{\boldsymbol{p}'}}, \tag{78}$$

where

$$H_{\mathrm{PP}}(\boldsymbol{t};\boldsymbol{k},p_1') = \widetilde{H}_{\mathrm{PP}}(\boldsymbol{t};\boldsymbol{k},p_1') - \mathcal{L}_\gamma(0),$$

$$\widetilde{H}_{\mathrm{PP}}(\boldsymbol{t};\boldsymbol{k},p_1') = \frac{1}{L^{2n}}\sum_{\substack{\boldsymbol{p}\in M_n^{\mathrm{NS}}\\ \boldsymbol{K}\in M_n^R}}\prod_{j=1}^n \frac{e^{-it_{21}\epsilon(K_j)-it_{32}\epsilon(p_j)+it_{31}\epsilon(k_j)}}{\sin\left(\frac{p_j-k_j}{2}\right)\sin\left(\frac{p_j-K_j}{2}\right)}\mathcal{L}_{\gamma_1}(P_{\boldsymbol{k}}-P_{\boldsymbol{K}})$$

$$\times e^{-it_{32}\left(\epsilon(p_1'+q_1-P_{\boldsymbol{p}})-\epsilon(p_1')\right)}. \tag{79}$$

To confirm Eq. (55) it suffices to show that for any fixed $m$ and $n$

$$\lim_{\gamma\to 0}\lim_{L\to\infty}H_{\mathrm{PP}}(\boldsymbol{t};\boldsymbol{k},p_1') = \mathcal{O}(|t_{32}|+|t_{21}|). \tag{80}$$

This is done in Appendix E. The bound (80) further implies that

$$C_{\mathrm{PP},\gamma\to 0}^{(n,n+m,n)}(q_1,q_2,0,\tau_2+\tau_1,\tau_1,0) = C_2^{(m)}(q_2,\tau_2)C_2^{(n)}(q_1,0)\mathcal{O}(\tau_2+\tau_1), \tag{81}$$

implying that contributions with $m>1$ are subleading in $C_{\mathrm{PP}}$.

Finally, to confirm the validity of the WP calculation of Sec. 7.2 we would need to show that for $m=1$,

$$\lim_{\gamma\to 0}\lim_{L\to\infty}H_{\mathrm{PP}} = \int dr_2\mathfrak{S}, \tag{82}$$

with the RHS given in Eq. (174). This equation is more complex to derive explicitly for any $n$. In Appendix E we write recursion relations that allow to compute $\lim_{\gamma\to 0}\lim_{L\to\infty}H_{\mathrm{PP}}$ for arbitrary $n$. Using these relations we checked explicitly that property Eq. (82) holds for $n=1,2,3$. Although we have not been able to explicitly prove it for arbitrary $n$, we conjecture that Eq. (82) is nonetheless valid, as suggested by the WP calculation of Sec. 7.2.

## 4.3 Subleading $\sqrt{t}$ growth

In the previous sections we extracted the leading behavior at long times and argued that in generic cases the signals asymptotically grows linearly in time. In this section we try to better understand the next-to-leading corrections in the long-time limit. For simplicity we will focus on the Ising model and we restrict ourselves to corrections arising due to the simplest contribution: $C_4^{(1,2,1)}$. In other words, we consider only contributions where all operators create or annihilate a single QP.

In App. C we analyze the subleading corrections to Eq. (24) and find that, when $q_1 \simeq q_2$ there is an intermediate-time regime characterized by a $\sqrt{|t_{32}|}$ growth of the four-point function

$$C_4^{(1,2,1)} \propto \sqrt{(\epsilon''(q_1) + \epsilon''(q_2)) |t_{32}|}, \qquad \text{when} \quad |t_{32}| \lesssim \frac{\epsilon''(q)}{|v(q_1) - v(q_2)|^2}, \qquad (83)$$

whereas for longer $t_2$ any additional contribution on top of Eq. (24) is exponentially suppressed.

Eq. (83) is particularly important when $v(q_1) = v(q_2)$ exactly. This includes the prominent case of optical probes, for which $q_j = 0 \; \forall j$ on which we will focus on until the end of the section. In this case, $C_4^{(1,2,1)}$ asymptotically grows like $\sqrt{|t_{32}|}$ at all times. This can be observed by computing the Fourier transform of the frequency-space singularity found in Refs. [95,96]. For completeness, we will provide an alternative way to extract this asymptotic behavior from the form factor expansion for $C_4^{(1,2,1)}$. Finally, we will move to the main part of this section, showing how the $\sqrt{|t_{32}|}$ growth can be *quantitatively* understood using a simple path-integral approach, which can be thought of as a generalization of the wavepacket approach, where the subleading wavepacket spreading is treated exactly.

Before proceeding, we note that the role played by the $\sqrt{|t_{32}|}$-divergences is different in the case of well-separated times (see Sec. 6) or two equal times (i.e. $C_{\text{NR}}$ in Sec. 7.1 and $C_{\text{PP}}$ in Sec. 7.2). For well-separated times, the linear-in-time divergence vanishes when $v(q_1) = v(q_2)$, and therefore Eq. (83) become the leading contribution. For two equal times, when $v(q_1) = v(q_2)$ the 1-QP contributions (i.e. the $n = 1$ terms in Eqs. (161) and (167)) vanish. Here, in the long-time limit, the term (83) will be subleading w.r.t. to the linear-in-time divergences discussed throughout this manuscript. However, given that, depending on the microscopic details of the model, the amplitude to create a single QP could be much larger than the amplitude to create a shower of $n > 1$ QPs, the contribution (83) can be non-negligible at intermediate times.

We now begin by extracting the asymptotic long time behavior through the form factor expansion of $C_4^{(1,2,1)}$ in the Ising model. For concreteness, we use the notation of well-separated times; the analysis for $C_{\text{NR}}$ and $C_{\text{PP}}$ would be identical and the results can be obtained by replacing $t_{32} \mapsto \tau_2$. Given that we are interested in the long-time behavior, we can expand $\tilde{C}_4^{(1,2,1)}$ near the annihilation pole to obtain

$$\tilde{C}_4^{(1,2,1)} \simeq |F(q=0)|^4 e^{-i(t_{31}+t_{42})\epsilon(q=0)} \frac{1}{L} \sum_{p \in \text{NS}} \frac{8}{p^2} e^{-it_{32} 2[\epsilon(p) - \epsilon(q=0)]} . \qquad (84)$$

Note that if we were to convert the sum over $p$ into an integral, this would diverge due to the $1/p^2$ singularity. This is expected, given that we have not yet subtracted the disconnected contributions proportional to $L$ (see Eq. (49)). To extract the finite contribution we exploit that the part proportional to $L$ has no time dependence — except for the trivial oscillations already encoded in the phase $e^{-i(t_{31}+t_{42})\epsilon(q=0)}$. Therefore, we can take a derivative w.r.t. $t_{32}$

to eliminate the divergence. Approximating $\epsilon(p) \simeq \epsilon''(0)p^2/2$, we then obtain

$$C_4^{(1,2,1)} \simeq -8|F(q=0)|^4 \sqrt{\frac{it_{32}\epsilon''(q=0)}{\pi}} e^{-i\epsilon(0)(t_{42}+t_{31})}. \tag{85}$$

Note that the same calculation could be done on any integrable QFT with fermionic QPs (i.e. $S(\theta = 0) = -1$) and would yield the same result, since it only relies on the annihilation pole structure at zero momentum.

We now explain how this divergence can be interpreted as arising from quasiparticle propagation and scattering. At time $t_2$, the state of the system can be written as

$$U(t_{21})A(q=0)\,|\Omega\rangle = F^*(0)e^{-i\epsilon(0)t_{21}} \int dx\,|x\rangle\,. \tag{86}$$

When the operator $A(q = 0, t_2)$ acts on this state —neglecting the scenarios in which it acts within a length $\xi$ of $x$, which we can argue to give non-divergent contributions only— we obtain

$$A(0)U(t_{21})A(0)\,|\Omega\rangle = (F^*(0))^2\,e^{-i\epsilon(0)t_{21}} \int dx \int dy\,|x,y\rangle\,. \tag{87}$$

Similarly we can expand the state starting from the bra side to obtain

$$\langle\Omega|A(0)U^\dagger(t_{43})A(0)\,|\Omega\rangle = (F(0))^2\,e^{-i\epsilon(0)t_{43}} \int dx' \int dy'\,\langle x',y'|\,. \tag{88}$$

Finally, we need to compute

$$L^{-1} \int dx \int dy \int dx' \int dy'\,\langle x',y'|U(t_{32})|x,y\rangle\,. \tag{89}$$

We assume these two particles propagate independently, except for the fermionic statistics. More explicitly, if we consider the case where the QP in $x$ propagates to $x'$ and the one in $y$ to $y'$ the amplitude will be $G(x'-x, t_{32})G(y'-y, t_{32})$ times a factor $S = -1$ if the ordering along the line has changed from $(x, y)$ to $(x', y')$. Here $G(r, t)$ is the Feynman propagator, that, since we are interested in processes taking place near $q = 0$, we can approximate as

$$G(r,t) = \sqrt{\frac{1}{2\pi i\epsilon''(0)t}} \exp\left(-\frac{r^2}{2it\epsilon''(0)}\right) e^{-i\epsilon(0)t}\,. \tag{90}$$

Note that, once the disconnected contributions are subtracted only the processes where the QPs have exchanged are left, and these will be weighed by a factor $S - 1 = -2$. Therefore the connected contribution can be written as

$$2L^{-1}(S-1) \int_{-\infty}^{+\infty} dx \int_{-\infty}^{+\infty} dy \int_{-\infty}^{+\infty} dx' \int_{-\infty}^{+\infty} dy'$$
$$\times G(x'-x, t_{32})G(y'-y, t_{32})\theta\left((x'-y')(y-x)\right), \tag{91}$$

with the factor of 2 accounting for the different contractions scheme that would yield the Green functions $G(y'-x, t_{32})G(x'-y, t_{32})$, which would be equivalent upon relabelling the integration variables. We can fix $x = 0$ and eliminate the $L^{-1}$ normalization, and using symmetry arguments we finally obtain

$$C_4 \simeq -8|F(q=0)|^4 \int_0^{+\infty} dy \int_{-\infty}^{+\infty} dx' \int_{-\infty}^{x'} dy'\,G(x'-x, t_{32})G(y'-y, t_{32})$$
$$= -8\sqrt{\frac{it_{32}\epsilon''(0)}{\pi}} e^{-i\epsilon(0)(t_{42}+t_{31})}, \tag{92}$$

in agreement with the form factor result. Intuitively the $\sqrt{\epsilon''(0)|t_{32}|}$ scaling can be understood as being the typical distance travelled, e.g. by the QP starting in $x$. Therefore when $y$ is within a distance $\sqrt{\epsilon''(0)|t_{32}|}$ of $x$ there is a significant amplitude for a scattering process (or, more properly, an exchange process) to take place.

We conjecture that also in more general interacting theories, the same $\sqrt{|t_{32}|}$ terms will appear due to similar reasons. In these cases factor $S-1$ will be replaced with the appropriate scattering matrix between two QPs with identical momentum.

## 5 Benchmark II: IQFT with diagonal scattering

The calculation of the previous section can be partially repeated in a more generic setting. Specifically, we consider a IQFT where $\mathcal{A}(x)$ is a local bosonic operator (viz. $[\mathcal{A}(x), \mathcal{A}(x')] = 0$ for $x \neq x'$). For simplicity we further restrict ourselves to the case where the scattering matrix is diagonal. An example would be the Ising model in a field [110].

The different particle species in the IQFT are labelled by an index $a$ and have masses $m_a$. Given that the theory is Lorentz invariant, the momentum $k$ and energy $\epsilon$ of particles are conveniently labelled by its rapidity $\theta$: $k_a(\theta) = \frac{m_a}{v} \sinh(\theta)$ and $\epsilon_a(\theta) = m_a \cosh(\theta)$. In the following we set $v = 1$ to ease notations. The eigenstates of the theory can be obtained from the vacuum $|\Omega\rangle$ through the action of the Faddeev-Zamolodchikov creation and annihilation operators [111, 112] $Z_a^\dagger(\theta), Z_a(\theta)$

$$|\boldsymbol{\theta}\rangle_{\boldsymbol{a}} = Z_{a_1}^\dagger(\theta_1) \cdots Z_{a_n}^\dagger(\theta_n) |\Omega\rangle . \tag{93}$$

The algebra of the $Z$ operators encodes the scattering matrix[4] as

$$Z_a^\dagger(\theta_1) Z_b^\dagger(\theta_2) = S_{a,b}(\theta_1 - \theta_2) Z_b^\dagger(\theta_2) Z_a^\dagger(\theta_1), \tag{94}$$

form factors of operators

$$F_{\boldsymbol{a}}(\boldsymbol{\theta}) = \langle \Omega | \mathcal{A}(0) | \boldsymbol{\theta} \rangle_{\boldsymbol{a}} , \tag{95}$$

can be determined through the form factor bootstrap (see e.g. [79, 87, 110, 113, 114]). Form factors between eigenstates with different rapidities can be obtained by crossing symmetry

$$_b\langle \boldsymbol{\theta}' | \mathcal{A}(0) | \boldsymbol{\theta} \rangle_{\boldsymbol{a}} = F_{\boldsymbol{a}, \bar{b}}(\boldsymbol{\theta}, \boldsymbol{\theta}' + i\pi), \tag{96}$$

where $\bar{b}$ labels the charge-conjugated particle of $b$. The annihilation poles we already encountered in the Ising model are a general feature of IQFT and matrix elements like (96) exhibit singularities of the form $\delta_{a_j, b_i}(\theta_j - \theta_i')^{-1}$ in the limit $\theta_j - \theta_i' \to 0$. This is expressed by the so-called annihilation-pole axiom, which states that when viewed as a function of $\theta_n$ $F_{a_1, \dots, a_j, \dots, a_n}(\theta_1, \dots, \theta_j, \dots, \theta_n)$ exhibits simple poles at positions $\theta_n = \theta_j + i\pi$ with residues given by

$$
\begin{aligned}
F_{a_1, \dots, a_j, \dots, a_n}(\theta_1, \dots, \theta_j, \dots, \theta_n) \sim & -i\delta_{a_n, \bar{a}_j} \frac{F_{a_1, \dots, \hat{a}_j, \dots, a_{n-1}}(\theta_1, \dots, \hat{\theta}_j, \dots, \theta_{n-1})}{\theta_n - \theta_j - i\pi} \\
& \times \Big[ \prod_{m=j+1}^{n-1} S_{a_m, a_j}(\theta_m - \theta_j) - l_{a_j}(\mathcal{A}) \prod_{m=1}^{j-1} S_{a_j, a_m}(\theta_j - \theta_m) \Big],
\end{aligned}
\tag{97}
$$

---

[4]In terms of the previously introduced notations we have

$$S_{a,b}^{a',b'}(k_a(\theta_1), k_b(\theta_2); k_a(\theta_1), k_b(\theta_2)) = S_{a,b}(\theta_1 - \theta_2) \delta_a^{a'} \delta_b^{b'} .$$

where $\hat{\theta}_j$ and $\hat{a}_j$ denote that the $j$-th argument of $F$ have been dropped and $l_a(\mathcal{A})$ is the so-called mutual non-locality index [87, 113], which expresses the locality property of the operator $\mathcal{A}$ relative to the elementary excitation created by $Z_a^\dagger$. In the following we only consider operators with $l_a(\mathcal{A}) = 1$.

Matrix elements of the form (96) for general sets of rapidities require a regularisation procedure [87] and involve generalized functions. As the form factor expansion of $C_4$ involves products of form factors, additional regularisations are required (see e.g. [102, 103]). We therefore resort to the finite-size regularization scheme of Ref. [115] and consider the IQFT on a ring of length $R$. In this case, periodic boundary conditions constrain the rapidities of particles to a discrete set determined by the solution of the Bethe equations and generally this constrains the rapidity in the bra and ket to be different, thus producing a well-defined expression. More precisely, energy eigenstates $|\theta_1, \ldots, \theta_n\rangle^R_{a_1, \ldots, a_n}$ on a ring of circumference $R$ are parametrized by rapidities that are solutions of the Bethe Ansatz equations

$$e^{ik_{a_j}(\theta_j)R} \prod_{l \neq j} S_{a_j, a_l}(\theta_j - \theta_l) = 1, \qquad j = 1, \ldots, n. \tag{98}$$

Under these constraints the rapidities in the bra and ket sites will generally differ at least by $O(R^{-1})$ as they obey different quantization conditions. Eq. (96) is then well defined in a finite-length system and suffices to define all form factors of interests. The logarithmic form of the Bethe Ansatz equations reads

$$Y_j(\theta_1, \ldots, \theta_n) = R k_{a_j}(\theta_j) + \sum_{l \neq j} \delta_{a_j, a_l}(\theta_j - \theta_l) = 2\pi I_j, \tag{99}$$

where $\delta_{a,b}(\theta) = -i \ln\left(-S_{ab}(\theta)\right)$ and $I_j$ are half-odd integers parametrizing the different solutions. The density of Bethe states is defined as

$$\rho_{a_1, \ldots, a_n}(\theta_1, \ldots, \theta_n) = \det\left(\frac{\partial Y_j(\theta_1, \ldots, \theta_n)}{\partial \theta_k}\right). \tag{100}$$

To leading order in $R$ the density of states is simply

$$\rho_a(\boldsymbol{\theta}) = \prod_j \frac{1}{R\epsilon_{a_j}(\theta_j)} + O(R^{-n-1}). \tag{101}$$

This approximation is sufficient for our purposes. Finally, the form factors in a finite volume $R$ are related to the thermodynamic-limit form factors by [115]

$$^R_b\langle\boldsymbol{\theta}'|\mathcal{A}(0)|\boldsymbol{\theta}\rangle^R_a = \frac{_b\langle\boldsymbol{\theta}'|\mathcal{A}(0)|\boldsymbol{\theta}\rangle_a}{\sqrt{\rho_a(\boldsymbol{\theta})}\sqrt{\rho_b(\boldsymbol{\theta}')}}, \tag{102}$$

up to exponentially small correction in the system size.

From these premises we expect the power counting argument presented in subsection 4.1.1 to apply also in the more general IQFT context upon renaming the number of sites $L$ with with the length $R$. Consequently, we also expect the subtraction scheme in Eq. (47)-(49) to be physically motivated also in the IQFT case and yield finite results in the thermodynamic limit.

We now proceed to verify this claim, as well as the semiclassical picture in Section 6 for the simplest case of $C_4^{(1,2,1)}$ in the two limits $q_3 \simeq -q_1$ and $q_4 \simeq -q_1$. We have

$$\tilde{C}_4^{(1,2,1)}(\boldsymbol{q}, t) = \sum_{a,b_{1,2},c,\beta_1} \mathcal{F}_{c;b_1,b_2;a}(\gamma; \beta_1, \beta_2; \alpha)\, e^{-it_{32}(\epsilon_{b_1}(\beta_1) + \epsilon_{b_2}(\beta_2)) - it_{43}\epsilon_c(\gamma) - it_{21}\epsilon_a(\alpha)}, \tag{103}$$

where we have defined

$$\mathcal{F}_{c;b_1,b_2;a}(\gamma;\beta_1,\beta_2;\alpha) = \frac{R^3}{2} \langle \Omega|\mathcal{A}(0)|\gamma\rangle^{R\,R}_{c\,c} \langle\gamma|\mathcal{A}(0)|\beta_1,\beta_2\rangle^R_{b_1,b_2}$$
$$\times^R_{b_1,b_2} \langle\beta_1,\beta_2|\mathcal{A}(0)|\alpha\rangle^{R\,R}_{a\,a} \langle\alpha|\mathcal{A}(0)|\Omega\rangle . \tag{104}$$

In the above momentum conservation fixes $\alpha$, $\beta_2$ and $\gamma$ as

$$\alpha = \operatorname{arcsinh}(q_1/m_a), \qquad \beta_2 = \operatorname{arcsinh}\frac{q_1+q_2-k_{b_1}(\beta_1)}{m_{b_2}}, \qquad \gamma = \operatorname{arcsinh}(-q_4/m_c). \tag{105}$$

## 5.1 Contribution $C_4^{(1,2,1)}(q,t)$

In the following we focus on the case where $q_3 \approx -q_1$, $q_2 \approx -q_4$. The study of the case $q_4 \simeq -q_1$ would proceed along the same lines. Isolating the most singular contributions due to annihilation poles gives

$$\mathcal{F}_{c;b_1,b_2;a}(\gamma;\beta_1,\beta_2;\alpha) \approx \frac{1}{2R} \frac{\left|1 - S_{b_2,b_1}(\beta_2-\beta_1)\right|^2}{\epsilon_{b_1}(\beta_1)\epsilon_{b_2}(\beta_2)} \left[ \frac{|F_{b_1}(\alpha)|^2|F_{b_2}(\gamma)|^2\delta_{a,b_1}\delta_{c,b_2}}{(k_{b_1}(\beta_1)-q_1)(k_{b_1}(\beta_1)+q_3)} \right.$$
$$\left. + \frac{|F_{b_2}(\alpha)|^2|F_{b_1}(\gamma)|^2\delta_{a,b_2}\delta_{c,b_1}}{(k_{b_1}(\beta_1)-q_2)(k_{b_1}(\beta_1)+q_4)} \right].$$

If $q_1 = -q_3$ we can use (C.13) to carry out the sum over $\beta_1$

$$\tilde{C}_4^{(1,2,1)}(q,t) \approx \sum_{b_1,b_2} C_{2,b_1}^{(1)}(q_1,t_{21})C_{2,b_2}^{(1)}(q_2,t_{43})|t_{32}||v_{b_1}(q_1)-v_{b_2}(q_2)|\mathcal{S}_{b_1,b_2}^{b_1,b_2}(q_1,q_2;q_1,q_2)$$
$$+ R\sum_{b_1,b_2} C_{2,b_1}^{(1)}(q_1,t_{31})C_{2,b_2}^{(1)}(q_2,t_{42}), \tag{106}$$

where $v_a(q) = \sqrt{m_a^2 + q^2}$ and we have defined

$$C_{2,a}^{(1)}(q,t) = \frac{|F_a(\alpha)|^2}{\epsilon_a(\alpha)}e^{-it\epsilon_a(\alpha)}, \qquad k_a(\alpha) = q. \tag{107}$$

We see that $C_{2,a}^{(1)}(q,t)$ is precisely the contribution of particles of species $a$ to the two-point function, i.e.

$$C_2(q,t) \simeq \sum_a C_{2,a}^{(1)}(q,t) + \dots \tag{108}$$

We conclude that the connected contribution $C_4^{(1,2,1)}(q,t)$ defined in accordance with (47) is indeed finite in the $R \to \infty$ limit and at large $t_{32}$ given by

$$C_4^{(1,2,1)}(q,t)$$
$$\approx \sum_{b_1,b_2} C_{2,b_1}^{(1)}(q_1,t_{21})C_{2,b_2}^{(1)}(q_2,t_{43})|t_{32}||v_{b_1}(q_1)-v_{b_2}(q_2)|\mathcal{S}_{b_1,b_2}^{b_1,b_2}(q_1,q_2;q_1,q_2). \tag{109}$$

When $0 < |q_1+q_3| \ll 1$ we can use (C.17) to obtain

$$C_4^{(1,2,1)}(q,t) \approx \sum_{b_1,b_2} C_{2,b_1}^{(1)}(q_1,t_{31})C_{2,b_2}^{(1)}(q_2,t_{42})\frac{1-e^{-it_{32}\left[\epsilon_a(-q_3)+\epsilon_{b_2}(-q_4)-\epsilon_a(q_1)-\epsilon_{b_2}(q_2)\right]}}{i(q_1+q_3)}$$
$$\times \operatorname{sign}\left(t(v_{b_2}(q_2)-v_{b_1}(q_1))\right)\mathcal{S}_{b_1,b_2}^{b_1,b_2}(q_1,q_2;q_1,q_2) + \dots \tag{110}$$

Note that (109) and (110) match the results of the semiclassical analysis in Sec. 6.

# 6 Wave packet analysis for well-separated times

In this section we consider the late time regime of $C_4(\boldsymbol{q};\boldsymbol{t})$ for well-separated times, i.e. the limit where all $|t_{ij}|$ are large if $i \neq j$. We will now show that in this limit $C_4(\boldsymbol{q};\boldsymbol{t})$ is dominated by processes like the ones shown in Fig. 1(a). These processes give rise to a contribution proportional to $|t_{32}|$. Our analysis is based on the kinematics and scattering of wave packet states, which we now introduce.

## 6.1 Wave packet states and their evolution

We start by considering the 1-QP sector of the theory. At length scales $\ell$ much larger than the correlation length $\xi$, we can build a basis of states from approximate eigenstates of both position $r$ and momentum $k$

$$|r,k\rangle_a = \int dx\, W(x;r,k)|x\rangle_a\,, \qquad W(x;r,k) = \frac{\exp\left(-\frac{(x-r)^2}{2\ell^2}\right)}{\sqrt{2\pi}\ell}e^{ikx}\,. \tag{111}$$

Here $|x\rangle_a$ denotes a state with one QP of species $a$ at position $x$. Equivalently, we can define wave packet states in the momentum basis

$$|r,k\rangle_a = \int \frac{dp}{2\pi}\tilde{W}(p;r,k)|p\rangle_a\,, \qquad \tilde{W}(p;r,k) = \exp\left(-\ell^2\frac{(p-k)^2}{2}\right)e^{i(k-p)r}\,. \tag{112}$$

These wave packet states provide a resolution of the identity in the 1-QP sector

$$\mathbb{1}_1 = \sum_a 2\sqrt{\pi}\ell \int \frac{dr\,dk}{2\pi}\,|r,k\rangle_{a\ a}\langle r,k|\,. \tag{113}$$

The advantage of the wave packet basis is that even though 1-QP excitations are essentially spatially localized, their time evolution is simple: the wave packet propagates approximately ballistically with a velocity set by its group velocity. More formally, the time evolution of the wave packet $|r,k\rangle_a$ can then be computed by inserting a resolution of the identity in terms of momentum eigenstates. Approximating the energy dependence as $\epsilon_a(p) \simeq \epsilon_a(k) + v_a(k)(p-k) + \frac{1}{2}\partial_k^2\epsilon_a(k)(p-k)^2$, we recover the well-known result that

$$U(t)|r,k\rangle_a \approx e^{-i\epsilon_a(k)t}|r+v(k)t,k\rangle_a\,. \tag{114}$$

In our discussion we have ignored the diffusive broadening of the wave packet proportional to $\sqrt{\partial_k^2\epsilon_a(k)t}$ as it is sub-leading with respect to the ballistic motion.

We now turn to wave packet states $|r_1,r_2;k_1,k_2\rangle_{a_1,a_2}$ involving two QPs, which are well defined as long as $|r_1-r_2| \gg \ell$. We start by defining "momentum-space wave packets" ("MWPs") of asymptotic scattering states $|p_1,p_2\rangle_{a_1,a_2}$

$$|r_1,r_2;k_1,k_2\rangle_{a_1,a_2}^{(p)} = \int \frac{dp_1}{2\pi} \int \frac{dp_2}{2\pi}\tilde{W}(p_1;r_1,k_1)\tilde{W}(p_2;r_2,k_2)|p_1,p_2\rangle_{a_1,a_2}\,. \tag{115}$$

The states $|p_1,p_2\rangle_{a_1,a_2}$ have non-trivial intertwining relations

$$|k_1,k_2\rangle_{a_1,a_2}^{(p)} = S_{a_1,a_2}^{a_1',a_2'}(k_1,k_2)|k_2,k_1\rangle_{a_2',a_1'}^{(p)}\,, \tag{116}$$

where $S(k_1,k_2)$ is the two-particle scattering matrix. MWP states inherit this intertwining relation if we assume that $\ell^{-1}$ is much smaller than the momentum scale over which $S(k_1,k_2)$ varies, i.e.

$$|r_1,r_2;k_1,k_2\rangle_{a_1,a_2}^{(p)} \approx S_{a_1,a_2}^{a_1',a_2'}(k_1,k_2)|r_2,r_1;k_2,k_1\rangle_{a_2',a_1'}^{(p)}\,. \tag{117}$$

For our purposes it is convenient to define a different class of WP states by

$$|r_1, r_2; k_1, k_2\rangle_{a_1,a_2} \equiv \hat{R} |r_1, r_2; k_1, k_2\rangle^{(p)}_{a_1,a_2} , \tag{118}$$

where

$$\hat{R} |r_1, r_2; k_1, k_2\rangle^{(p)}_{a_1,a_2} = \theta_H(r_{21}) |r_1, r_2; k_1, k_2\rangle^{(p)}_{a_1,a_2} + \theta_H(r_{12}) |r_2, r_1; k_2, k_1\rangle^{(p)}_{a_2,a_1} . \tag{119}$$

Here $r_{ij} = r_i - r_j$ and $\theta_H$ denotes Heaviside $\theta$ function. The rationale underlying the definition of the $\hat{R}$-ordering is that it ensures the property

$$|r_1, r_2; k_1, k_2\rangle_{a_1,a_2} = |r_2, r_1; k_2, k_1\rangle_{a_2,a_1} . \tag{120}$$

The time evolution of 2WP states is derived in Appendix A. First, provided that the two QPs never come close to each other, i.e. as long as $\forall t'$ s.t. $|t'| < |t|$, $|r_1 + v_{a_1}(k_1)t' - r_2 - v_{a_2}(k_2)t'| \gg \ell$,

$$U(t) |r_1, r_2; k_1, k_2\rangle_{a_1,a_2} \Big|_{\text{ns}} \simeq e^{-i(\epsilon_{a_1}(k_1)+\epsilon_{a_2}(k_2))t} |r_1 + v_{a_1}(k_1)t, r_2 + v_{a_2}(k_2)t; k_1, k_2\rangle_{a_1,a_2} . \tag{121}$$

Here "ns" indicates that during the time evolution no scattering event takes place. If instead at some intermediate time $t_S$ the two wave packets are in a neighbourhood of the same position $r_S$, a scattering process will take place. In this case there will be a time window around $t_S$ when the two-QP state does not admit a simple description in terms of wave packet states. Nonetheless, as we discuss in Appendix A, after some microscopic time the wave packet description will become accurate again and the evolution beyond the time $t_S$ can be approximated by[5]

$$U(t) |r_1, r_2; k_1, k_2\rangle_{a_1,a_2} \Big|_{\text{s}} \simeq e^{-i(\epsilon_{a_1}(k_1)+\epsilon_{a_2}(k_2))t} \\ \times \sum_{a_3,a_4} \tilde{\mathcal{S}}^{a_3 a_4}_{a_1 a_2}(k_1, k_2) |r_1 + v_{a_1}(k_1)t, r_2 + v_{a_2}(k_2)t; k_1, k_2\rangle_{a_3,a_4} . \tag{122}$$

In Eq. (122), $\tilde{\mathcal{S}}^{a'b'}_{ab}(k_1, k_2)$ is related to the on-shell scattering matrix $S^{a'b'}_{ab}(k_1, k_2)$ by

$$\tilde{\mathcal{S}}^{a'b'}_{ab}(k_1, k_2; k'_1, k'_2) = \begin{cases} S^{a'b'}_{ab}(k_1, k_2), & \text{if } [v_a(k_1) - v_b(k_2)]t > 0, \\ S^{b'a'}_{ba}(k_2, k_1), & \text{if } [v_a(k_1) - v_b(k_2)]t < 0. \end{cases} \tag{123}$$

[In practice, the sign of $t$ will depend on whether the scattering event takes place in the forward or backward branch of the Keldysh contour.] The matrix $\mathcal{S}^{a'b'}_{ab}(k_1, k_2)$ in Eq. (24) is then given by

$$\mathcal{S}^{a'b'}_{ab}(k_1, k_2) = \tilde{\mathcal{S}}^{a'b'}_{ab}(k_1, k_2) - \delta^{a'}_a \delta^{b'}_b , \tag{124}$$

if the scattering process is on shell and zero otherwise.

In the description of the scattering process above we did not consider the amplitude for the two QPs to scatter into an asymptotic states with more than two QPs, since such processes would only give rise to subleading corrections in the long-time limit. Similarly, we did not consider the case where the two QPs fuse into a single new QP; in fact, the hypothetical product of this fusion would not be stable and thus should not be accounted among the stable QPs of the theory. Rather, the whole process of fusion into the meta-stable excitation and its subsequent decay into two or more QPs should be considered as the scattering process.

---

[5]Here we are neglecting displacement of the center of the wave packets due to the scattering phase shift (see e.g. Ref. [116]). The rationale is that the displacement due to scattering is a microscopic length-scale, much smaller than the width of the wave packet $\ell$. Therefore, the displacement due to scattering is negligible when computing overlaps between wave packet states.

## 6.2  Wave packet states with more QPs

The discussion above can be generalized to the sector with $n > 2$ QPs. The $n$-QP sector can be most easily defined in terms of asymptotic momentum states $|\boldsymbol{p}\rangle_a$, e.g. in or out states, as commonly done in the context of IQFT or asymptotic Bethe ansatz [117]. In terms of these we can immediately generalize Eqs. (115) and (118)

$$|\boldsymbol{r};\boldsymbol{k}\rangle_a^{(p)} = \int \frac{d^n \boldsymbol{p}}{(2\pi)^n} |\boldsymbol{p}\rangle_a \prod_{j=1}^n \tilde{W}(p_j;r_j,k_j), \qquad |\boldsymbol{r};\boldsymbol{k}\rangle_a = \hat{R}|\boldsymbol{r};\boldsymbol{k}\rangle_a^{(p)}, \qquad (125)$$

where, as before, $\hat{R}$ permutes the order of the WPs in such a way that the order in the ket directly reflects the order in real space.

In order for these states to have a simple time evolution it is necessary that the wave packets are in the asymptotic region, i.e. $|r_i - r_j| \gtrsim \ell \; \forall i, j$. However, even if this condition is satisfied initially, eventually the wave packets might collide. To understand what happens in these cases we need to distinguish between an integrable system and a non-integrable system which only admits a QP description at zero energy density.

For integrable systems, assuming that at a final time $t$ the wave packets are in the asymptotic region, the final state will be obtained by combining ballistic evolution of single QPs with 2-QP scattering processes as in Eq. (122). Due to integrability, this simple picture of the evolution will yield correct results even if three or more QPs were to meet in the same region at the some intermediate time.

For non-integrable systems, the same picture is approximate, i.e. it can be applied *only if* three or more QPs never meet in the same region at the same time. Nonetheless, in all scenarios considered in this manuscript, the wave packets are always created in such a way that this condition applies in the long-time limit. Therefore our results should apply to integrable and non-integrable models alike.

## 6.3  Operators and their action on the vacuum

For our purposes, the discussion above gives a complete picture of the time evolution of wave packet states. In order to be able to compute correlation functions we need to characterize how wave packets are created and annihilated by the operators $A$ and $\mathcal{A}$. In other words, we need to determine the matrix elements of these operators between two wave packet states. We first consider the simplest case where one of the two states, e.g. the ket, is the QP vacuum and postpone the discussion for the case where both states contain QPs to subsection 6.5.

The simplest contribution to $\mathcal{A}(x)|\Omega\rangle$ corresponds to the creation of a single QP. We define its matrix element with plane wave states as

$$_a\langle p|\mathcal{A}(x)|\Omega\rangle = F_a^*(p)e^{-ipx}. \qquad (126)$$

This implies

$$_a\langle r,k|\mathcal{A}(x)|\Omega\rangle \approx F_a^*(k)W^*(x;r,k), \qquad (127)$$

where we have assumed that $F_a(p)$ is approximately constant over a scale $\ell^{-1}$. In terms of one-wave packet states, the operator with the most straightforward action is then

$$\mathfrak{A}(r,q) = \int dx\, W(x;r,q)\mathcal{A}(x). \qquad (128)$$

This is related to $A(q)$ by

$$A(q) = \int dr\, \mathfrak{A}(r,q). \qquad (129)$$

Given (127) the action of $\mathfrak{A}(r,q)$ on the vacuum can be approximated as

$$\mathfrak{A}(r,q)\,|\Omega\rangle \simeq \sum_a F_a^*(q)\,|r,q\rangle_a + (n-\text{QP states}, n \geq 2) + \cdots \tag{130}$$

Here $\cdots$ denotes possible component of the state that go beyond the QP description in non-integrable systems. Throughout our work we assume these to give subleading corrections in the long-time limit.

In Eq. (130), we can also enquire about the component of the state with two or more QPs. While we do not expect this component to admit a simple wave packet description immediately after the action of $\mathfrak{A}$, upon time-evolution the various QPs will drift away from each other and eventually a wave packet description of the state will be possible. We start by considering the state $\mathfrak{A}(0,q)\,|\Omega\rangle$. After time evolution this will take the approximate form

$$U(t)\mathfrak{A}(0,q)\,|\Omega\rangle \simeq \sum_n \sum_{\boldsymbol{a}} \int \frac{d^n\boldsymbol{k}}{(2\pi)^n n!}\tilde{F}_{\boldsymbol{a}}^*(\boldsymbol{k};q)e^{-it\sum_{j=1}^n \epsilon_{a_j}(k_j)}|r,\boldsymbol{k}\rangle_{\boldsymbol{a}}\,, \tag{131}$$

for some functions $\tilde{F}_{\boldsymbol{a}}(\boldsymbol{k};q)$ that play the role of wave packet form factors. Given that the QPs are created around $x=0$, we must have that in the long-$t$ limit $\boldsymbol{r}$ are of the form $r_j \simeq v_{a_j}(k_j)t$ up to a $t$-independent correction that we neglect. Furthermore, given that $\mathfrak{A}(r,q)$ approximately increases the momentum of the state it acts upon by $q$, it must be that $\tilde{F}_{\boldsymbol{a}}(\boldsymbol{k};q) \simeq 0$ unless $q - \sum_{j=1}^n k_j \sim \ell^{-1}$.

From the above equation, we can derive an approximate expression of the state for any $r \neq 0$, i.e.

$$U(t)\mathfrak{A}(r,q)\,|\Omega\rangle \simeq \sum_n \sum_{\boldsymbol{a}} \int \frac{d^n\boldsymbol{k}}{(2\pi)^n n!}\tilde{F}_{\boldsymbol{a}}^*(\boldsymbol{k};q)e^{ir\left(q-\sum_{j=1}^n k_j\right)}e^{-it\sum_{j=1}^n \epsilon_{a_j}(k_j)}|r,\boldsymbol{k}\rangle_{\boldsymbol{a}}\,, \tag{132}$$

with $r_j \simeq r + v_{a_j}(k_j)t$. The presence of the factor $e^{ir\left(q-\sum_{j=1}^n k_j\right)}$ for $r \neq 0$ can be inferred by studying how both sides of the equation transform under translations.

Note that, the discussion above of the $n$-QP component for $n > 1$ is approximate, unlike Eq. (130) which is asymptotically exact in the $\ell \to \infty$ limit. Nonetheless, since we will only use this description to argue that certain processes are subleading, we expect it to be a sufficiently good approximation. Instead, in Sec. 7.1 and 7.2, when processes where more QPs are created by a single operator are important, a more precise form of the wave packet states can be instead obtained for the action of $A(q)$ directly, as shown in Appendix B.

## 6.4 Two-point functions

With the technology introduced so far, we can tackle the computation of two-point functions, which will serve as a warm-up exercise for the determination of four-point functions. We can show that their long-time limit can be recovered from a semiclassical description of 1-QP states. While this is a well-known fact, it will allow us to clarify aspects that will also enter the 4-point function argument. We thus consider the following 2-point function

$$C_2(q,t) = \frac{1}{V}\,\langle\Omega|A(-q;t)A(q;0)|\Omega\rangle_C\,. \tag{133}$$

We express $A(q;0)$ on the right in terms of its semiclassical counterpart $\mathfrak{A}$, and express $A(-q;t)$ on the left in terms of the local object $\mathcal{A}(x,t)$. Using translational invariance we then fix $x=0$ and remove the $V^{-1}$ factor. This gives

$$C_2(q,t) = \int dr\,\langle\Omega|\mathcal{A}(0;t)\mathfrak{A}(r,q;0)|\Omega\rangle_C\,. \tag{134}$$

To evaluate this, we consider

$$U(t)\mathfrak{A}(r,q;0)|\Omega\rangle \simeq \sum_a F_a^*(q)e^{-i\epsilon_a(q)t}|r+v_a(q)t,q\rangle_a$$
$$+\sum_{a,b}\int \frac{dk_1\,dk_2}{(2\pi)^2}\tilde{F}_{a,b}^*(k_1,k_2;q)e^{ir(q-k_1-k_2)}e^{-it(\epsilon_a(k_1)+\epsilon_b(k_2))} \quad (135)$$
$$\times |r+v_a(k_1)t,r+v_b(k_2)t;k_1,k_2\rangle_{a,b}$$
$$+\ldots$$

To obtain a non-zero contribution to $C_2$, all QPs need to be annihilated by $\mathcal{A}(0;t)$.

We start by arguing that the contribution from states with two QPs is suppressed in the long-time limit — a similar discussion would apply to three or more QPs. In the large $t$ limit the two wave packets will be separated by a distance $|v_a(k_1)-v_b(k_2)|t \gg \ell$. In this case, it will be impossible to annihilate both QPs with a single $\mathcal{A}(0)$ operator, which can create or annihilate QPs only in the proximity of $x=0$. We can thus conclude that these contributions are suppressed in the limit of large $t$ and we must look at the 1-QP state to compute the leading contribution.[6] In this case we have

$$C_2(q,t) = \int dr \sum_a F_a^*(q)e^{-i\epsilon_a(q)t}\langle\Omega|\mathcal{A}(0;0)|r+v(q)t,q\rangle_a + o(1)$$
$$= \sum_a |F_a(q)|^2 e^{-i\epsilon_a(q)t} + o(t^0), \quad (136)$$

as expected.

## 6.5 Action of the operators on few-QPs states

In the analysis of four-point functions we will further need to understand the action of the operator $\mathcal{A}$ and $\mathfrak{A}$ on states $|r,k\rangle_a$ containing several QPs. Here the key property is the locality of $\mathcal{A}(x)$, i.e. $\mathcal{A}(x)$ can affect previously existing QPs or create new QPs only within a neighbourhood of $x$. Instead if a state $|\psi\rangle$ contains QPs further away from $x$, these are unaffected by the action of $\mathcal{A}(x)$. The same property holds for the wave packet operator $\mathfrak{A}(r,t)$ which can create QPs and destroy QPs only in the neighborhood of $r$.

We can summarize this property in the following equation:

$$_b\langle r',k'|\mathfrak{A}(x,q)|r,k\rangle_a \approx {}_{b_{\mathrm{F}}}\langle r_{\mathrm{F}}',k_{\mathrm{F}}'|r_{\mathrm{F}},k_{\mathrm{F}}\rangle_{a_{\mathrm{F}}}$$
$$\times {}_{b_{\mathrm{N}}}\langle r_{\mathrm{N}}',k_{\mathrm{N}}'|\mathfrak{A}(x,q)|r_{\mathrm{N}},k_{\mathrm{N}}\rangle_{a_{\mathrm{N}}}. \quad (137)$$

Here the subscript F and N respectively denotes the subset of wave packets that are far ($|r_{\mathrm{F},j}-x| \gg \ell$) and near ($|r_{\mathrm{N},j}-x| \lesssim \ell$) from the point $x$ where $\mathfrak{A}$ acts.

We further expect that the amplitude $_{b_{\mathrm{N}}}\langle r_{\mathrm{N}}',k_{\mathrm{N}}'|\mathfrak{A}(x,q)|r_{\mathrm{N}},k_{\mathrm{N}}\rangle_{a_{\mathrm{N}}}$ in the second line has a regular dependence w.r.t. the positions and momenta of the wave packets. This should be contrasted with the action of operators on extended plane waves, which, as a consequence of locality has annihilation poles (sometimes also called kinematic poles). The existence of this

---

[6]To be more precise, it can be shown that the contribution from two-QPs states is $O(t^{-1/2})$. One way to see this is by expanding the two-point function in terms of momentum eigenstates and evaluating the integral over momenta within the stationary phase approximation. Alternatively, within the wavepacket analysis, one can reason as follows. At time $t$, the center of the two wave packets in Eq. (135) will be separated by a distance $(v_a(k_1)-v_b(k_2))t$. The width of the two wave packets instead only grows diffusely, i.e. like $\sqrt{Dt}$ for some $D$. Therefore, the values of $k_1$ and $k_2$ which will yield a non-negligible contribution to $C_2$ are the ones for which $(v_a(k_1)-v_b(k_2))t \sim \sqrt{Dt}$. Consequently, the integrand in $k_1$ and $k_2$ is non-negligible only in a window where $k_1-k_2 = O(1/\sqrt{t})$, which controls the $1/\sqrt{t}$ scaling of the two-QPs contribution to $C_2$.

pole structure is well-known in integrable models (either continuum or on the lattice) [79, 87, 110, 113, 114].

While at first glance it might seem that the annihilation pole structure of the form factors in terms of plane waves and the regularity of the amplitudes above are in tension with one another, we show here that the structure above can be derived by re-expressing the plane wave form factors in terms of wave packets. We do this by considering a simple case where $\mathfrak{A}(r_2, q_2)$ acts on a single-particle state $|r_1, q_1\rangle_a$. For simplicity we further restrict ourselves to the case where after the action of $\mathcal{A}$, a two-QPs state is obtained, viz. we compute $\mathbb{1}_2 \mathfrak{A}(r_2, q_2)|r_1, q_1\rangle_a$ where $\mathbb{1}_2$ denotes the projector onto the two-QPs sector

$$\mathbb{1}_2 = \sum_{b,c} 4\pi\ell^2 \int \frac{d^2\boldsymbol{Q}\, d^2\boldsymbol{R}}{(2\pi)^2} \, |\boldsymbol{R}, \boldsymbol{Q}\rangle_{b,c} \, {}_{b,c}\langle \boldsymbol{R}, \boldsymbol{Q}| \,. \tag{138}$$

To compute $\mathbb{1}_2 \mathfrak{A}(r_2, q_2)|r_1, q_1\rangle_a$ we can therefore re-express the semiclassical states in term of (asymptotic) momentum eigenstates in terms of which

$$_{bc}\langle k_3, k_2|\mathcal{A}(0)|k_1\rangle_a = {}_{bc}\langle k_3, k_2|\mathcal{A}(0)|k_1\rangle_a^{(\text{reg})} + \frac{\delta_{a,c} f_b(\boldsymbol{k})}{k_1 - k_3} + \frac{\delta_{a,b} f_c(\boldsymbol{k})}{k_1 - k_2} \,, \tag{139}$$

where we separated regular contributions (first line) from kinematic poles (second line). Therefore

$$\begin{aligned}
\mathbb{1}_2 \, \mathfrak{A}(r_2, q_2)|r, q_1\rangle_a &= \sum_b \frac{F_b^*(q_2)}{1 + \delta_{a,b}} |r_1, r_2; q_1, q_2\rangle_{ab} \\
&\quad + \sum_{b,c} \int \frac{d^2Q\, d^2R}{(2\pi)^2} \, K_{abc}(\boldsymbol{r}, \boldsymbol{q}; \boldsymbol{R}, \boldsymbol{Q})|\boldsymbol{R}, \boldsymbol{Q}\rangle_{bc} \,.
\end{aligned} \tag{140}$$

Here the first and second terms on the right-hand-side arise respectively from the kinematic poles and the regular contribution to the 3-particle form factor. Crucially the kernel $K_{abc}(\boldsymbol{r}, \boldsymbol{q}; \boldsymbol{R}, \boldsymbol{Q})$ is negligible unless

$$|r_1 - r_2| \lesssim \ell \,, \qquad |R_1 - r_2| \lesssim \ell \,, \qquad |R_2 - r_2| \lesssim \ell \,. \tag{141}$$

Finally, note that, since the kernel $K_{abc}(\boldsymbol{r}, \boldsymbol{q}; \boldsymbol{R}, \boldsymbol{Q})$ is derived from the regular part of the form factor it will also be a regular object, as anticipated.

To summarize, $\mathfrak{A}(r, q)$ can only create or annihilate wave packets near $r$. In particular, when acting on a wave packet states where all wave packets' positions are far from $r$, $\mathfrak{A}(r, q)$ will create new wave packets in proximity of $r$ as described by Eq. (130), while the previously-present wave packets will remain undisturbed.

## 6.6 Scattering-connected contributions

We now proceed to discuss the long-time limit of the four-point function (6). We start by discussing a subset of processes contributing to $C_4$, which are scattering connected. We compute their amplitude and show that this gives us Eq. (24). We will then argue in subsection 6.7 that other contributions give rise to subleading corrections only.

To obtain the leading contributions we start by expressing $C_4$ as

$$C_4 = \int dr_1 \int dr_2 \int dx_3 \, e^{iq_3 x_3} \left\langle \Omega \left| \mathcal{A}(0, t_4)\mathcal{A}(x_3, t_3)\mathfrak{A}(r_2, q_2; t_2)\mathfrak{A}(r_1, q_1; t_1) \right| \Omega \right\rangle_C \,. \tag{142}$$

The processes we want to consider are the ones where the two operators $\mathfrak{A}$ on the right create two QPs in two spatially separated positions. These QPs will travel ballistically and, if their

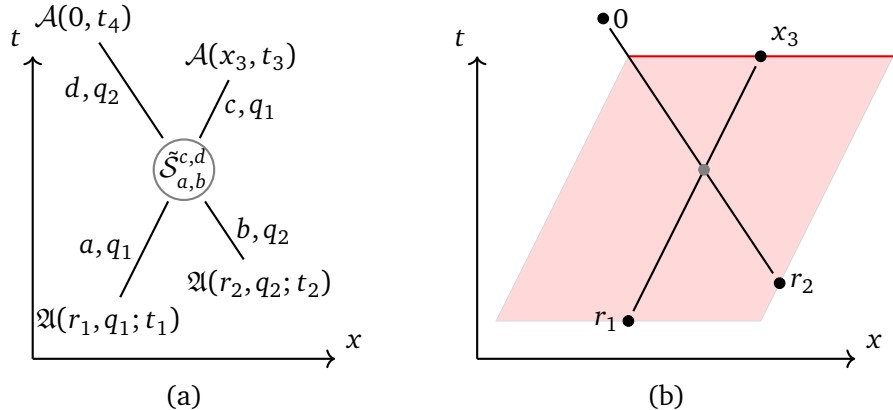

Figure 6: (a) Example of a scattering-connected process giving rise to the leading contribution when all time differences are large. $\mathfrak{A}(r_1, q_1; t_1)$ creates a QP wave packet around momentum $q_1$ and species $a$. Analogously, $\mathfrak{A}(r_2, q_2; t_2)$ creates a QP wave packet around momentum $q_2$ and species $b$. The quasiparticle scatter around position $r_S$ and time $t_S$ into a new pair of QP wave packets of momenta $q_1$ and $q_2$, and species $c$ and $d$. Ultimately these QPs are annihilated by the operators $\mathcal{A}(x_3, t_3)$ and $\mathcal{A}(0, t_4)$ respectively. (b) After integration over $r_1$ and $r_2$, the integral over $x_3$ gives the same contribution for any $x_3$ such that a scattering event takes place. For the example in the figure, this set is denoted by a red line.

trajectory cross, they scatter into a (possibly new) pair of QPs. Of these, one is annihilated by $\mathcal{A}(x_3, t_3)$ and the remaining QP is then annihilated by $\mathcal{A}(0, t_4)$. For concreteness we will assume that $\mathcal{A}(q_3, t_3)$ annihilates the QP with momentum $q_1$. (see Fig. 6a)

Before showing more formally how the long-time divergence arises, we discuss the intuition underlying it. We begin by noting that for the process to give a non-negligible contribution $r_2$ must lie in a region of size $\propto \ell$ centered at $-v_b(q_2)t_{42}$. Similarly, $r_1$ must be in a region centered at $x_3 - v_a(q_1)t_{31}$. Therefore we will be left with an integral over $x_3$ only. If $q_3 \approx -q_1$, all values of $x_3$ yield the same contribution as long as a scattering process takes place. Therefore the integral over $x_3$ will be proportional to the length of the set of $x_3$ for which a scattering process takes place (this set is denoted as a red segment in Fig. 6b). Given that the velocities of the WPs are fixed by $q_1$ and $q_2$, the length of this segment is proportional to $|t_{32}|$, yielding the anticipated long-time divergence.

In more detail, the amplitude of the process would be given by

$$
\begin{aligned}
C_{4,S} \simeq \sum_{a,b,c,d} & F_a^*(q_1)F_b^*(q_2)F_c(q_1)F_d(q_2)\mathcal{S}_{a,b}^{c,d}(q_1, q_2) \\
& \times e^{-i\epsilon_a(q_1)t_{21}} e^{-i(\epsilon_a(q_1)+\epsilon_b(q_2))t_{32}} e^{-i\epsilon_d(q_2)t_{43}} \\
& \times \int dx_3\, e^{i(q_3+q_1)x_3} \\
& \times \int dr_1 \int dr_2 W(0; r_2 + v_b(q_2)t_{42}, q_2) \\
& \times |W(x_4; r_1 + v_a(q_1)t_{31}, q_1)|\mathfrak{S}(r_1, r_2),
\end{aligned}
\tag{143}
$$

where we split the amplitude $W(x_3; r_S + v_c(k_3)t_{3S}, q_1)$ into its absolute value (last line) and its phase $e^{iq_1 x_3}$ (third line). We further introduced the function $\mathfrak{S}(r_1, r_2)$ (which is also implicitly a function of the various momenta and the QP species), defined as follows. $\mathfrak{S} = 1$ if $r_1$ and $r_2$ are such that the presence of a scattering process between time $t_2$ and $t_3$ described by

$\mathcal{S}_{a,b}^{c,d}(q_1, q_2)$ is compatible with the ballistic propagation of all QPs; otherwise $\mathfrak{S} = 0$.

To perform the integral over $r_1$ and $r_2$ it is convenient to change the integration variables to

$$r_3 = r_1 + v_a(q_1)t_{31}, \tag{144}$$

$$r_4 = r_2 + v_b(q_2)t_{42}, \tag{145}$$

in terms of which, the last two lines of Eq. (143) become

$$\int dr_3 \int dr_4 W(0; r_4, q_2) W(x_3; r_3, q_1) \mathfrak{S}'(r_3, r_4), \tag{146}$$

with $\mathfrak{S}'(r_3, r_4) := \mathfrak{S}(r_1, r_2)$. To make progress we note that the integrand is negligible everywhere *except* in the region where $r_4 = O(\ell)$ and $r_3 = x_3 + O(\ell)$. We also note that excluding the cases where the scattering takes place in proximity of $(r_2, t_2)$ or $(r_3, t_3)$ (in which our semiclassical treatment breaks down anyway) $\mathfrak{S}'(r_3 + O(\ell), r_4 + O(\ell)) = \mathfrak{S}'(r_3, r_4)$. In the integral we therefore replace $\mathfrak{S}'(r_3, r_4)$ with $\mathfrak{S}'(x_3, 0)$. Therefore, we approximate the integrals in Eq. (146) as $\mathfrak{S}'(x_3, 0)$. This leaves us to evaluate $\int dx_3 e^{i(q_1+q_3)x_3}\mathfrak{S}'(x_3, 0)$. We can see that $\mathfrak{S}' = 1$ if $x_3$ lies between the two points $x_3' = -v_b(q_2)t_{43}$ and $x_3'' = -v_b(q_2)t_{42} + v_a(q_1)t_{32}$, which define a segment of length $|v_a(q_1) - v_b(q_2)||t_{32}|$. If $q_3 \simeq -q_1$ the integrand is approximately constant over this segment, thus producing a signal proportional to its length, i.e. proportional to $|t_{32}|$, as anticipated.

More explicitly, we can perform the integration over $x_3$ to obtain that, if $q_3 \simeq -q_1$

$$\begin{aligned}
C_{4,S} \simeq \sum_{a,b,c,d} &F_a^*(q_1)F_b^*(q_2)F_c(q_1)F_d(q_2)\mathcal{S}_{a,b}^{c,d}(q_1, q_2) \\
&\times e^{-i\epsilon_a(q_1)t_{21}} e^{-i(\epsilon_a(q_1)+\epsilon_b(q_2))t_{32}} e^{-i(\epsilon_d(q_2))} \\
&\times \frac{|v_a(q_1) - v_b(q_2)|}{\alpha} \sin(\alpha|t_{32}|) + o(|t_{32}|),
\end{aligned} \tag{147}$$

with $\alpha = |v_a(q_1) - v_b(q_2)||q_3 + q_1|/2$. The $O(1)$ terms we neglected are subleading contribution in the $q_3 + q_1 \to 0$ limit. In this limit $\alpha \to 0$ and the last line of Eq. (147) reduces to $|v_a(q_1) - v_b(q_2)||t_{32}|$, as anticipated in Eq. (24). For finite, but small, $q_3 + q_1$, instead the signal grows linearly until $|t_{32}| \lesssim \alpha^{-1}$ when it reaches a maximum value of $\sim \alpha^{-1}$.

So far, we considered the case where $A(q_3, t_3)$ annihilates the rightmost QP after the scattering process. Another contribution to be taken into account is that where $A(q_3, t_3)$ annihilates the leftmost QP after the scattering. This will give a contribution identical to that in Eq. (147) but with $\mathcal{S}_{a,b}^{c,d}(q_1, q_2)$ replaced by $\mathcal{S}_{a,b}^{d,c}(q_1, q_2)$. The general answer will be given by the sum of these two contributions as in Eq. (24).

Finally, note that so far we did not specify the time ordering of $t_1$, $t_2$, $t_3$, and $t_4$. In fact all of the equations we wrote hold for any time ordering. This is thanks to the definition of $\mathcal{S}$ (123), which already takes into account whether the scattering process takes place during a forward or backward evolution.

Lastly, before proceeding we comment on what happens when $|v_a(q_1) - v_b(q_2)|$ vanishes, so that the QP worldlines are parallel and do not intersect. This case can in fact arise very naturally if $q_1 = q_2 = 0$ and $v_c(0) = 0$ for all QP species $c$ due to some symmetry, e.g. reflection symmetry. In this scenario, the leading term in Eq. (147) becomes zero. Furthermore, the whole assumption underlying the semiclassical calculation breaks down. In fact, we assumed that $r_2$, the position at which the second operator acts, is far from the position of the previously present QP. If all the quasiparticle velocities are the same, instead the only way to obtain a connected contribution is to have all operators act along the same ray. In this context the simple calculation presented here becomes unreliable. Instead, in this case, we expect that

the long-time four-point function grows like $\sqrt{|t_{32}|}$. Indeed this is the case for the integrable QFT for which the first few-QPs contributions to $C_4$ in Refs. [95, 96]. In Sec. 4.3 we will discuss these contributions in more detail and argue that they arise from a combination of WP spreading and scattering.

## 6.7 Other processes are subleading

We now proceed to argue that the contributions considered so far are the leading ones in the long-time limit $|t_{ij}| \to \infty$ for every $i \neq j$.

We will work under the assumption that in the long-time limit, the leading processes can be understood in terms of ballistically propagating WPs. As we already mentioned, we can separate these processes into scattering-connected (e.g. Fig. 1a) and fully-connected processes (e.g. Fig. 1b). The first one are processes where the graph consisting of operators $\mathcal{A}$ and QP trajectories is connected by one or more scattering vertices alone. In other words, scattering-connected processes are the ones that would reduce to product of two-point functions if the scattering matrix were trivial. Fully-connected processes, on the other hand are ones where the operators $\mathcal{A}$ and the QP trajectories form a connected graph by themselves. More formally, in the four-point function case, these are processes whose amplitude would involve matrix elements, that, upon using the factorization in Eq. (137) would involve non-trivial factors of the form $_{b_N}\langle r'_N, k'_N | \mathfrak{A}(x,q) | r_N, k_N \rangle_{a_N}$ in which both the bra and ket are not the ground state $|\Omega\rangle$. An example is the process depicted in Fig. 1(b), where $\mathfrak{A}(r_2, q_2; t_2)$ annihilates a previously present QP and creates two new QPs.

We begin by arguing that fully-connected processes are subleading in the long-time limit. The key idea is as follows. Imagine fixing the momenta of wavepackets, then the operator positions in fully-connected processes are fixed by the ballistic propagation of QPs, i.e. the integration over the operator positions is $O(1)$. Furthermore, as discussed in Subsec. 6.5, the matrix elements to locally annihilate and create QP wave packets are regular and cannot produce further divergences. Therefore the final integration over momenta (introduced to return to real space from the $k$-space of the preceding discussion) does not produce any extra long-time divergence and the overall contribution is $O(1)$.

While showing this in detail for the most general processes would be cumbersome, we exemplify the discussion in the case of the process in Fig. 1(b). To do so, we write $C_4$ as

$$\int dr_1, \int dr_2 \int dr_4 \langle \Omega | \mathfrak{A}(r_4, q_4; t_4) \mathcal{A}(0, t_3) \mathfrak{A}(r_2, q_2; t_2) \mathfrak{A}(r_1, q_1; t_1) | \Omega \rangle_C . \tag{148}$$

For a given set of positions $(r_1, r_2, r_4)$ we can (approximately) compute the amplitude of the process in terms of matrix elements of $\mathfrak{A}$ and $\mathcal{A}$ among semiclassical states. We are interested in processes where $\mathfrak{A}(r_1, q_1; t_1)$ and $\mathfrak{A}(r_4, q_4; t_4)$ create and annihilate a QP, while a two-QPs state is present between time $t_2$ and $t_3$. Therefore

$$\sum_{a,b} \int d^3 r \; F_b(-q_4) F_a^*(q_1) e^{-i\epsilon_b(-q_4)t_4 + i\epsilon_a(q_1)t_1}$$
$$\times \,_b\langle r_4 - v_b(-q_4)t_4 | \mathcal{A}(0, t_3) \mathbb{1}_2 \mathfrak{A}(r_2, q_2; t_2) | r_1 - v_a(q_1)t_1, q_1 \rangle_a . \tag{149}$$

Applying Eq. (140) to expand $\mathbb{1}_2 \mathfrak{A}(r_2, q_2; t_2) | r_1 - v_a(q_1)t_1, q_1 \rangle_a$ we can understand the structure of the dominant contributions at late times to $C_4$. Selecting the contribution in (140) arising from the kinematic poles and selecting the same contribution in the 3-particle form factor involving $\mathcal{A}(0, t_3)$ gives (147). Instead, selecting the part involving $K$ in either or both of the 3-particle form factors gives contributions that are *regular*, i.e. $\mathcal{O}(t_{32}^0)$.

Finally, in the limit $|t_{ij}| \to \infty$ for every $i \neq j$, the only scattering-connected processes that are compatible with the ballistic motion of wavepackets (for a generic set of momenta) are the

ones computed in the previous subsection, where each operator creates or destroys only one QP. Note that this last statement will cease to be true when we will allow two times to coincide in Secs. 7.1 and 7.2. In these cases, the ballistic motion of wavepackets will be compatible with new scattering-processes, where some operators create or annihilate multiple QPs. The contribution of these new processes will again be proportional to the overall timescale.

# 7  Wave-packet analysis for two equal times

We now relax the time-separation assumption and focus on the opposite limit where two times coincide.

## 7.1  Non-rephasing correlator

We begin by considering the non-rephasing correlator $C_{\text{NR}}$ defined in Eq. (21) (i.e. an instance of $C_4$ with $t_4 = t_2 = \tau_1$ and $q_3 = -q_2$). In this case, most of the arguments given in Sec. 6 remain valid. In particular, fully-connected processes are always $O(1)$, and, since scattering-connected processes can produce signals that grow linearly in time, they are the one dominating the long-time behaviour, or, equivalently, the strongest singularities in frequency space.

However, when two times coincide, there are scattering-connected processes where two operators can exchange many QP wavepackets, which are compatible with the ballistic kinematic of the wavepackets.

The processes we consider are like the one depicted in Fig. 2. Here, the second operator creates a shower of $n$ QPs, which propagate out ballistically. In doing so, these QPs can scatter with the QP created by the first operator. Eventually, the system needs to be evolved backward to time $t_2 = t_4$. If only forward-scattering processes —i.e. processes where the two QPs do not exchange any momentum, as we have throughout this manuscript— have taken place, the $n$ wave packets will refocus in the same region of size $\ell$, where they can be annihilated by a single operator. To further simplify the notation, we will also assume that upon scattering the QP index does not change, viz. the scattering matrix is diagonal: $\mathcal{S}^{cd}_{ab} \propto \delta_{ac}\delta_{bd}$. [The extension to the case of non-diagonal $\mathcal{S}$ is straightforward.]

Given that it is important to give a correct quantitative estimation of the amplitude of processes where a single operator creates many QPs, the treatment in Eq. 132 could be insufficient and we will switch to using the operator $A$ directly, for which a more precise treatment is possible (see Appendix B). The main results obtained in Appendix B are the following. When $A(q)$ acts on the vacuum, we generally have

$$U(t)A(q)|\Omega\rangle \simeq \sum_n \sum_a \frac{1}{\mathcal{N}_a} \int dx \int \frac{d^n \boldsymbol{k}}{(2\pi)^n} e^{ix\left(q - \sum_j k_j\right)} \mathcal{F}^*_{\boldsymbol{a}}(\boldsymbol{k}) e^{-i\sum_j \epsilon_{a_j}(k_j)t} |\boldsymbol{r};\boldsymbol{k}\rangle_{\boldsymbol{a}}, \quad (150)$$

with $r_j = x + v_{a_j}(k_j)t$. For integrable systems the relation above can be explicitly derived, and we can therefore relate $\mathcal{F}_{\boldsymbol{a}}(\boldsymbol{k})$ to the form factors in terms of asymptotic states $F_{\boldsymbol{a}}(\boldsymbol{k})$

$$\mathcal{F}_{\boldsymbol{a}}(\boldsymbol{k}) = (2\pi)^{n/2} \ell^n F_{\boldsymbol{a}}(\boldsymbol{k}). \quad (151)$$

Similarly to Eq. (132), Eq. (150) has a natural interpretation: The operator $A(q)$ can perturb the state in the neighbourhood of an arbitrary point $x$. As the state is evolved in time, it can be approximated as a superposition of wave packet states moving ballistically away from $x$.

For the purpose of computing $C_{\text{NR}}$, we express it as

$$C_{\text{NR}} = \underbrace{\langle\Omega| U^\dagger(\tau_1)A(-q_2)U^\dagger(\tau_2)}_{\text{bra}} \int dr_1 \underbrace{\mathcal{A}(0)U(\tau_2)A(q_2)U(\tau_1)\mathfrak{A}(r_1,q_1)|\Omega\rangle}_{\text{ket}}\Bigg|_C . \quad (152)$$

Note that since $t_{31} = \tau_1 + \tau_2$ is large, we must consider only cases where the first operator creates a single QP. We therefore consider the component of the state such that

$$\mathfrak{A}(r_1, q_1) |\Omega\rangle = \sum_a F_a^*(q_1) |r_1; q_1\rangle_a + \cdots, \tag{153}$$

with the ellipsis denoting other components of the state (e.g. more QPs states, or components that are beyond the QP description), which will give rise to subleading contributions in the long-$\tau$ limit and we will therefore neglect.

Time evolving this state and applying Eq. (150) we obtain the following expression

$$
\begin{aligned}
U(t)A(q_2)U(\tau_1)\mathfrak{A}(r_1, q_1)|\Omega\rangle \simeq \sum_n \sum_{(b,a)} \frac{1}{\mathcal{N}_b} \int dx_2 \int \frac{d^n k}{(2\pi)^n} e^{ix_2\left(q_2 - \sum_j k_j\right)} \mathcal{F}_b^*(k) F_a^*(q_1) \\
\times e^{-i\sum_j \epsilon_{b_j}(k_j)t} e^{-i\epsilon_a(q_1)(\tau_1 + t)} |(y(t), r_1'); (k, q_1)\rangle_{(b,a)},
\end{aligned}
\tag{154}
$$

with $y_j(t) = x_2 + v_{b_j}(k_j)t$ and $r_1' = r_1 + v_a(q_1)(\tau_1 + t)$. The rationale for this expression is as follows. At time $\tau_1$, the operator $A(q_2)$ will act in some arbitrary position $x_2$. If $x_2$ is far away from the position of the previous wave packet $r_1 + v_a(q_1)\tau_1$, then there will be an intermediate time $t + \tau_1$ —with $0 < t \ll \tau_2$—, where Eq. (150) is a good description of the state in the spatial segment $[x_2 - v_{MAX}t, x_2 + v_{MAX}t]$. If $r_1 + v_a(q_1)\tau_1$ is well-outside of this segment, then we expect a wave packet description of the form above to be valid. Note that, however, the above expression will fail to correctly capture the state when $r_1 + v_a(q_1)\tau_1$ is inside the interval. Nonetheless this will not constitute a problem for our analysis as, in the long-$\tau$ limit, the interval where the approximation breaks down remains constant, contributing to an $o(\tau_1, \tau_2)$ error in our calculation, which can be neglected w.r.t. the long-$\tau$ divergence we are interested in.

Evolving the state above up to time $\tau_1 + \tau_2$, we can then follow the discussion in Sec. 6.2: the various wave packets propagate ballistically and if a wave packet created by $A(q_2)$ collides with the wave packet created by $\mathfrak{A}(r_1, q_1)$, then a scattering event will take place. We start by considering the case where only forward scattering processes take place. In this case, at time $\tau_1 + \tau_2$, the position of the wave packet is determined uniquely by their group velocity, and the scattering events amount to multiplying the integrand in Eq. (154) by a factor $(1 + \mathfrak{S})$ where $(1 + \mathfrak{S})$ is given by the product the scattering amplitudes for each collision that takes place. Here $\mathfrak{S}$ implicitly depends on $r_1, x_2, k, q_1$, as well as $\tau_1$ and $\tau_2$, as all of these variables influence the kinematics of the wave packets and, therefore, which scattering events take place.

Finally, when acting with $\mathcal{A}(0)$ on the ket, we consider the case when this annihilates the QP created by $\mathfrak{A}(r_1, q_1)$. The ket side is then given by

$$
\begin{aligned}
|\text{ket}\rangle \simeq \sum_n \sum_{b,a} \frac{1}{\mathcal{N}_b} \int dx_2 \int \frac{d^n k}{(2\pi)^n} e^{ix_2\left(q_2 - \sum_j k_j\right)} \mathcal{F}_b^*(k) \\
\times e^{-i\sum_j \epsilon_{b_j}(k_j)t} |y(\tau_2); k\rangle_b \\
\times |F_a(q_1)|^2 e^{-i\epsilon_a(q_1)(\tau_1 + \tau_2)} \int dr_1 W(0; r_1 + v_a(q_1)(\tau_1 + \tau_2), q_1)(1 + \mathfrak{S}).
\end{aligned}
\tag{155}
$$

As we did in Subsec. 6.6, we can approximate $\mathfrak{S}$ with $\mathfrak{S}\big|_{r_1 = -v_a(q_1)(\tau_1 + \tau_2)}$ by committing an $o(\tau_1, \tau_2)$ error. In this way the integral $\int dr_1 W(0; r_1 + v_a(q_1)(\tau_1 + \tau_2), q_1)$ yields 1.

The bra side of $C_{\text{NR}}$ can be expanded using Eq. (150) directly, to obtain the following expression for $C_{\text{NR}}$

$$C_{\text{NR}} \simeq \sum_n \sum_{\boldsymbol{b},a} \frac{1}{\mathcal{N}_{\boldsymbol{b}}} \int dx_2 \int dx_2' \int \frac{d^n\boldsymbol{k}}{(2\pi)^n} \int \frac{d^n\boldsymbol{k'}}{(2\pi)^n} e^{ix_2\left(q_2-\sum_j k_j\right)} e^{ix_2'\left(q_2+\sum_j k_j'\right)} \mathcal{F}_{\boldsymbol{b}}^*(\boldsymbol{k}) \mathcal{F}_{\boldsymbol{b}}(\boldsymbol{k'})$$

$$\times e^{-i\sum_j \epsilon_{b_j}(k_j)\tau_2} e^{+i\sum_j \epsilon_{b_j}(k_j')\tau_2} {}_{\boldsymbol{b}}\langle \boldsymbol{y'}(\tau_2); \boldsymbol{k'} | \boldsymbol{y}(\tau_2); \boldsymbol{k}\rangle_{\boldsymbol{b}}$$

$$\times |F_a(q_1)|^2 e^{-i\epsilon_a(q_1)(\tau_1+\tau_2)} \mathfrak{S}, \tag{156}$$

with $y'(t) = x_2' - v_{b_j}(k_j')t$. Note that in this calculation we assumed that all scattering events to be forward-scattering, otherwise the overlap would have suppressed in the long-time limit.

Note that $\mathfrak{S}$ is independent of $\boldsymbol{k'}$ and $x_2'$. We can then perform the integral over $\boldsymbol{k'}$ and $x_2'$ as discussed in Appendix B to obtain

$$C_{\text{NR}} \simeq \sum_n \sum_{\tilde{\boldsymbol{b}},a} \frac{1}{\mathcal{N}_{\boldsymbol{b}}} \int \frac{d^n\boldsymbol{k}}{(2\pi)^n} \frac{|\mathcal{F}_{\boldsymbol{b}}(\boldsymbol{k})|^2}{(2\pi\ell^2)^n} (2\pi)\delta\left(q_2 - \sum_j k_j\right)$$

$$\times |F_a(q_1)|^2 e^{-i\epsilon_a(q_1)(\tau_1+\tau_2)} \int dx_2 \mathfrak{S}. \tag{157}$$

Finally, we can discuss the integral over $x_2$. For a fixed set of $\boldsymbol{k}$ and $\boldsymbol{a}$, $\mathfrak{S}(x_2)$ is piecewise constant. Upon a rescaling $\tau_2 \mapsto \lambda\tau_2$, the length of each piece is increased y a factor of $\lambda$, while the value of $\mathfrak{S}$ in each piece does not change. Therefore we can rewrite $C_{\text{NR}}$ in the scaling form

$$C_{\text{NR}} = \sum_a e^{-i\epsilon_a(q_1)(\tau_1+\tau_2)} \alpha_{\text{NR}}(a,q_1,q_2)\tau_2 + o(\tau_1,\tau_2), \tag{158}$$

$$\alpha_{\text{NR}}(a,q_1,q_2) = \sum_n \sum_{\boldsymbol{b}} \frac{1}{\mathcal{N}_{\boldsymbol{b}}} \int \frac{d^n\boldsymbol{k}}{(2\pi)^n} \frac{|\mathcal{F}_{\boldsymbol{b}}(\boldsymbol{k})|^2}{(2\pi\ell^2)^n} (2\pi)\delta\left(q_2 - \sum_j k_j\right) |F_a(q_1)|^2 \frac{\int dx_2 \mathfrak{S}}{\tau_2}. \tag{159}$$

The expression above does not rely on integrability, but only on the exsistence of a QP description for some of the low-energy excitations of the system. However, in the general case, $\mathcal{F}$ and the scattering matrices are hard to compute. In integrable systems, instead the form factors and the scattering matrices are known and $\alpha_{\text{NR}}$ can be explicitly computed. We exemplify such a calculation for the Ising model. First of all, we use Eq. (151) to replace the wave packet form factors $\mathcal{F}$ with the asymptotic-states form factors $F$ to obtain

$$C_{\text{NR}} = \alpha_{\text{NR}} e^{-i\epsilon(q_1)(\tau_1+\tau_2)}\tau_2 + o(\tau_1,\tau_2), \tag{160}$$

$$\alpha_{\text{NR}} = \sum_{n \text{ odd}}^{\infty} \alpha_{\text{NR}}^n, \tag{161}$$

$$\alpha_{\text{NR}}^n = |F(q_1)|^2 \frac{1}{n!} \int \frac{d^n\boldsymbol{k}}{(2\pi)^{n-1}} \delta\left(q_2 - \sum_j k_j\right) |F(\boldsymbol{k})|^2 \frac{\int dx_2 \mathfrak{S}(\boldsymbol{k},x_2)}{\tau_2}. \tag{162}$$

Here, $\mathfrak{S} = -2$ if an odd number of scattering processes take place, and $\mathfrak{S} = 0$ otherwise. The integral $\int dx_2 \mathfrak{S}$ can then be expressed as follows. Consider the set of velocities $\{v(q_1), v(k_1), \ldots, v(k_n)\}$, by permuting its elements we can define an ordered set of velocities $(\tilde{v}_1, \ldots, \tilde{v}_{n+1})$ such that $\tilde{v}_1 < \tilde{v}_2 < \cdots < \tilde{v}_{n+1}$. Then, it is easy to see that, when $r_1 + v(k_n)\tau_2 = \tilde{v}_j\tau_2$ for some $j$, $\mathfrak{S}$ switches value: it either jumps from 0 to $-2$ or vice-versa. Therefore, since $n$ is odd

$$\int dx_2 \mathfrak{S} = -2\tau_2 \sum_{j=1}^{n+1} (-1)^j \tilde{v}_j. \tag{163}$$

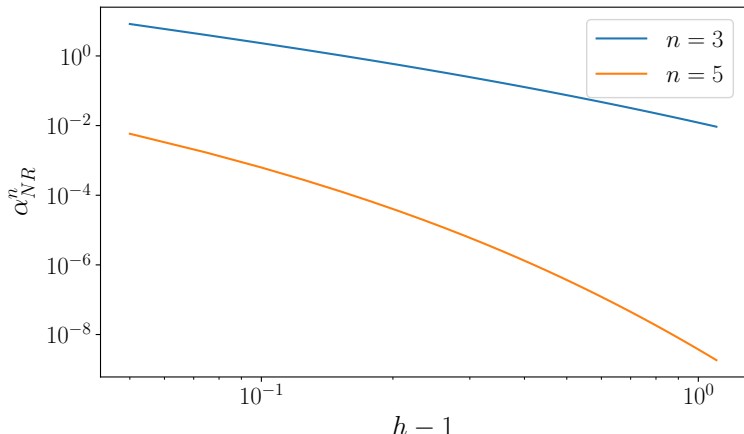

Figure 7: $\alpha_{\text{NR}}^n$ for the Ising model, as a function of the transverse field $h$. We expect the overall coefficient $\alpha_{\text{NR}} = \sum_{n \text{ odd}} \alpha_{\text{NR}}^n$ to be dominated by the $n = 3$ contribution for the values of $h$ reported in the figure.

We are then left with the task of performing the integral over $k_1, \ldots, k_n$ and sum over $n$.

For small $n$ we can numerically compute the integral over $k_1, \ldots, k_n$ by approximating it as a sum over a discrete set of momenta. For $q_1 = q_2 = 0$, we report the results for $n = 3, 5$ as a function of $h$ in Fig. 7. (Note that when $q_1 = q_2$, $\alpha_{\text{NR}}^1 = 0$) Given that the sum over $n$ must converge and that $\alpha_{\text{NR}}^5$ is already much smaller than $\alpha_{\text{NR}}^3$, we expect contributions at larger $n$ to give only negligible corrections.

## 7.2 Pump-probe correlator

The pump-probe correlator $C_{\text{PP}}$, defined in Eq. (22), can be analyzed in a similar fashion. In this case $t_4 = t_1$ and we must then consider scattering-connected processes where the first operator $A(q_1, 0)$ produces a shower of $n$ QPs, which are then annihilated by the fourth one $A(-q_1, 0)$, while $A(q_2, \tau_1)$ and $A(-q_2, \tau_1 + \tau_2)$ exchange a single QP (see Fig. 3).

The starting point is to re-express $C_{\text{PP}}$ as

$$
C_{\text{PP}} = \underbrace{\langle \Omega | A(-q_1) U^\dagger(\tau_1 + \tau_2)}_{\text{bra}} \int dr_2 \, \underbrace{\mathcal{A}(0) U(\tau_2) \mathfrak{A}(r_2, q_2) U(\tau_1) A(q_1) | \Omega \rangle}_{\text{ket}}_{C} \, . \tag{164}
$$

The computation of the leading term of $C_{\text{PP}}$ closely parallels the one of $C_{\text{NR}}$ in the previous section. For this reason, we simply quote the result

$$
\begin{aligned}
C_{\text{PP}} \simeq \sum_n \sum_{b,a} \frac{1}{\mathcal{N}_b} \int \frac{d^n k}{(2\pi)^n} \frac{|\mathcal{F}_b(k)|^2}{(2\pi \ell^2)^n} (2\pi) \delta \left( q_1 - \sum_j k_j \right) \\
\times |F_a(q_2)|^2 e^{-i\epsilon_a(q_2)\tau_2} \int dx_1 \mathfrak{S} \, .
\end{aligned} \tag{165}
$$

Here $x_1$ is the starting position of the shower of $n$ QPs created by $A(q_1)$. Instead, the position $r_1$, at which the single probe QP is created, is fixed to be within a distance $\ell$ from $-v_a(q_2)\tau_2$. Finally, $\mathfrak{S}$ accounts for the product of the phases accumulated from each forward-scattering process.

As for $C_{\text{NR}}$, the long-time divergence is determined by the behaviour of the integral $\int dx_1 \mathfrak{S}$. Looking at Fig. 3, it is clear that, for a fixed set of momenta $\boldsymbol{k}$, $\mathfrak{S}(x_1)$ is a piece-wise constant function. Furthermore, upon rescaling $\tau_{1,2} \mapsto \lambda \tau_{1,2}$ the length of each piece is proportional to $\lambda$. In order to make this scaling explicit, we can rewrite $C_{\text{PP}}$ as

$$C_{\text{PP}} = \sum_a (\tau_1 + \tau_2) e^{-i\epsilon_a(q_2)\tau_2} \mathcal{C}_{\text{PP}}\left(\frac{\tau_2}{\tau_1 + \tau_2}; a, q_1, q_2\right) + o(\tau_1, \tau_2), \tag{166}$$

where $\mathcal{C}_{\text{PP}}$ is given by

$$\mathcal{C}_{\text{PP}}\left(\frac{\tau_2}{\tau_1 + \tau_2}; a, q_1, q_2\right) = \sum_n \sum_b \frac{1}{\mathcal{N}_b} \int \frac{d^n \boldsymbol{k}}{(2\pi)^n} \frac{|\mathcal{F}_b(\boldsymbol{k})|^2}{(2\pi \ell^2)^n} (2\pi)\delta\left(q_1 - \sum_j k_j\right)$$
$$\times |F_a(q_2)|^2 \frac{\int dx_1 \mathfrak{S}\left(\frac{\tau_2}{\tau_1+\tau_2}; a, q_1, q_2\right)}{\tau_1 + \tau_2}. \tag{167}$$

Note that, from geometrical considerations it follows that

$$\mathcal{C}_{\text{PP}}\left(\frac{\tau_2}{\tau_1 + \tau_2} = 1; a, q_1, q_2\right) = \alpha_{\text{NR}}(a, q_1, q_2). \tag{168}$$

Next, we restrict ourselves to the simple case of the Ising model, where a single QP species exist and $\mathfrak{S} = (-1)^{\#(\text{scattering events})}$. In this case we can compute explicitly the scaling function $\mathcal{C}_{\text{PP}}$. To do so, we start by re-expressing $C_{\text{PP}}$ in terms of form factors $F$.

$$C_{\text{PP}} = (\tau_1 + \tau_2) e^{-i\epsilon(q_2)\tau_2} \mathcal{C}_{\text{PP}}\left(\frac{\tau_2}{\tau_1 + \tau_2}; q_1, q_2\right) + o(\tau_1, \tau_2), \tag{169}$$

$$\mathcal{C}_{\text{PP}}\left(\frac{\tau_2}{\tau_1 + \tau_2}; q_1, q_2\right) = \mathcal{C}_{\text{PP}}^n\left(\frac{\tau_2}{\tau_1 + \tau_2}; q_1, q_2\right), \tag{170}$$

$$\mathcal{C}_{\text{PP}}^n\left(\frac{\tau_2}{\tau_1 + \tau_2}; q_1, q_2\right) = \frac{1}{n!} \int \frac{d^n \boldsymbol{k}}{(2\pi)^n} |F(\boldsymbol{k})|^2 (2\pi)\delta\left(q_1 - \sum_j k_j\right) \tag{171}$$

$$\times |F_a(q_2)|^2 \frac{\int dx_1 \mathfrak{S}\left(\frac{\tau_2}{\tau_1+\tau_2}; q_1, q_2\right)}{\tau_1 + \tau_2}. \tag{172}$$

Finally, we can express the integral over $x_1$ in terms of the velocities $v(k_j)$. For this purpose we consider the set of positions

$$\{v(k_1)\tau_1, \ldots, v(k_n)\tau_1, v(k_1)(\tau_1 + \tau_2) - v(q_2)\tau_2, \ldots, v(k_n)(\tau_1 + \tau_2) - v(q_2)\tau_2\}, \tag{173}$$

and define the permuted set of coordinates $(\tilde{y}_1, \ldots, \tilde{y}_{2n})$ such that $\tilde{y}_1 < \tilde{y}_2 < \cdots < \tilde{y}_{2n}$. In terms of this, the integral over $r_2$ is given by

$$\int dr_2 \mathfrak{S} = -2 \sum_{j=1}^{2n} (-1)^j \tilde{y}_j. \tag{174}$$

Finally, as for $\alpha_{\text{NR}}$ we need to numerically perform the sum over momenta and $n$ to obtain $\mathcal{C}_{\text{PP}}$. For small enough $n$, we can compute a numerical approximation of $\mathcal{C}_{\text{PP}}^n$ by discretizing momentum space. $\mathcal{C}_{\text{PP}}^3$ and $\mathcal{C}_{\text{PP}}^5$ are reported in Fig. 8 as a function of $\tau_2/(\tau_1 + \tau_2)$ for $q_1 = q_2 = 0$ and a few values of transverse field $h$. For the values of the fields we analyzed, $\mathcal{C}_{\text{PP}}^5 \ll \mathcal{C}_{\text{PP}}^3$, suggesting that the sum over $n$ is dominated by $n = 3$. (As for $\alpha_{\text{NR}}$, the $n = 1$ contribution vanishes when $q_1 = q_2 = 0$.)

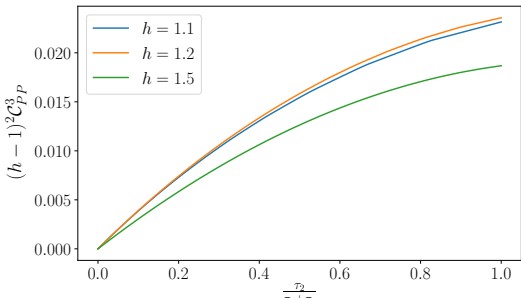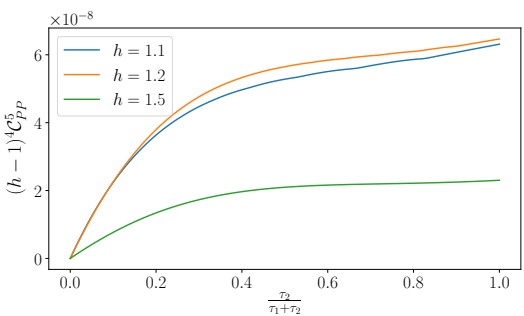

Figure 8: $\mathcal{C}_{\text{PP}}^n$ for the Ising model, as a function of $\tau_2/(\tau_1+\tau_2)$. We expect the overall function $\mathcal{C}_{\text{PP}} = \sum_{n\ \text{odd}} \mathcal{C}_{\text{PP}}^n$ to be dominated by the $n=3$ contribution for the values of $h$ reported in the figure.

## 8 Summary and conclusions

Nonlinear response holds the promise to become a useful tool in the experimental investigation of exotic strongly correlated materials. However, at present there is a shortage of examples where clear yet far-reaching insights can be inferred directly from the experimental data. Our work contributes to establishing such paradigms for the broad class of one-dimensional interacting many-particle quantum systems that feature stable quasiparticle excitations over the ground state. We have shown that third-order response functions in these systems exhibit linear-in-time divergences. As these signals become large at late times they eventually should dominate the nonlinear response functions and be readily observable in experiments.

The information encoded in this singular behaviour of the nonlinear response functions can be inferred from Eq. (24): when finite-momentum probes can be used, the detailed structure of the long-time divergence is determined by (i) data that can be extracted from linear response measurements, and (ii) the two quasiparticle scattering matrix. This suggests that it should be in principle possible to use nonlinear response as a way of experimentally measure the scattering matrix of collective excitations in condensed-matter systems.

Our work opens many interesting directions for further investigations. Perhaps the most important one is whether similar long-time divergences occur in higher dimensions. The nonperturbative arguments reported in Ref. [49] suggest that this is the case. It would be interesting to find a model where this can be explicitly verified through a direct calculation.

It would also be interesting to understand the fate of the linear-in-time divergences in the presence of disorder. In some settings this will yield localization of QPs even at arbitrarily weak disorder, therefore the linear-in-time growth described in this work would only persist as an intermediate-time effect, with the signal eventually saturating to a value set by the localization length. However, depending on the symmetry of the model, QPs could remain delocalized, but the wavepacket motion will be sub-ballistic. In this context, one might expect that long-time divergences will persist but possibly with a sub-linear growth in time.

## Acknowledgments

MF thanks Max McGinley, Sounak Biswas, Abhishodh Prakash, and Nick Bultinck for many insightful discussions and Max McGinley and Souank Biswas for collaboration on related projects.

**Funding information** This work was supported in parts by the EPSRC under grants EP/S020527/1 and EP/X030881/1 (FHLE, SAP). FHLE would like to thank the Institut Henri Poincaré (UAR 839 CNRS-Sorbonne Université) and the LabEx CARMIN (ANR-10-LABX-59-01) for their support.

# A Properties of wavepacket states

In this appendix we present some properties of the WP states that are used throughout the text. We first discuss the scattering of WPs and then present a formula for the identity resolution in terms of WP states.

## A.1 Scattering of wavepackets

In this appendix show how scattering of wave packets produces the time-evolution reported in Eq. (122). We begin by listing some identities that will be useful in the derivation. The two-particle scattering matrix is defined in terms of momentum-space scattering states by

$$|k_1, k_2\rangle_{a_1,a_2}^{(p)} = S_{a_1,a_2}^{a_1',a_2'}(k_1, k_2)|k_2, k_1\rangle_{a_2',a_1'}^{(p)}. \tag{A.1}$$

Given our assumption that our gapped single-particle states are kinematically protected, we have

$$U(t)|k_1, k_2\rangle_{a_1,a_2}^{(p)} = e^{-i\left(\epsilon_{a_1}(k_1) + \epsilon_{a_2}(k_2)\right)t}|k_1, k_2\rangle_{a_1,a_2}^{(p)}. \tag{A.2}$$

Assuming that $\ell^{-1}$ is much smaller than the momentum scale over which the two-particle $S$-matrix changes (i.e. we consider $\ell$ to be large compared to the lattice spacing/short distance cutoff) these imply the following relations for MWP states

$$|r_1, r_2; k_1, k_2\rangle_{a_1,a_2}^{(p)} \approx S_{a_1,a_2}^{a_1',a_2'}(k_1, k_2)|r_2, r_1; k_2, k_1\rangle_{a_2',a_1'}^{(p)}, \tag{A.3}$$

$$U(t)|r_1, r_2; k_1, k_2\rangle_{a_1,a_2}^{(p)} \approx e^{-iEt}|r_1', r_2'; k_1, k_2\rangle_{a_1,a_2}^{(p)}, \tag{A.4}$$

where we have defined

$$r_j' = r_j + v_{a_j}(k_j)t, \qquad E = \epsilon_{a_1}(k_1) + \epsilon_{a_2}(k_2). \tag{A.5}$$

Using (A.3) and (118) we then can approximately express MWP states in terms of $|r_1, r_2; k_1, k_2\rangle$ WP states as

$$
\begin{aligned}
|r_1, r_2; k_1, k_2\rangle_{a_1,a_2}^{(p)} \approx {} & \theta_H(r_{21})|r_1, r_2; k_1, k_2\rangle_{a_1,a_2} \\
& + \theta_H(r_{12}) S_{a_1,a_2}^{a_1',a_2'}(k_1, k_2)|r_1, r_2; k_1, k_2\rangle_{a_1',a_2'},
\end{aligned}
\tag{A.6}
$$

where we have defined $r_{ij} = r_i - r_j$. Using the definition (118) of WP states in terms of MWP states then gives

$$e^{+iEt}U(t)|r_1, r_2; k_1, k_2\rangle_{a_1,a_2} \approx \theta_H(r_{21})|r_1', r_2'; k_1, k_2\rangle_{a_1,a_2}^{(p)} + \theta_H(r_{12})|r_2', r_1'; k_2, k_1\rangle_{a_2,a_1}^{(p)}. \tag{A.7}$$

Finally we use Eq. (A.6) to arrive at the following approximate evolution equations for WP states

$$
\begin{aligned}
e^{+iEt}U(t)|r_1, r_2; k_1, k_2\rangle_{a_1,a_2} \approx {} & \theta_H(r_{21} r_{21}')|r_1', r_2'; k_1, k_2\rangle_{a_1,a_2} \\
& + \theta_H(r_{21})\theta_H(r_{12}') S_{a_1,a_2}^{a_1',a_2'}(k_1, k_2)|r_1', r_2'; k_1, k_2\rangle_{a_1',a_2'} \\
& + \theta_H(r_{12})\theta_H(r_{21}') S_{a_2,a_1}^{a_2',a_1'}(k_2, k_1)|r_1', r_2'; k_1, k_2\rangle_{a_1',a_2'}.
\end{aligned}
\tag{A.8}
$$

This result has a nice intuitive interpretation: the term in the first line corresponds to the situation where WPs do not collide, and consequently they do not acquire any additional phases. The second and third lines correspond to situations where the two WPs exchange their order, which involves a scattering process. We note that all approximate equalities presented here become asymptotically exact in the large-$\ell$ limit.

Finally we note that these formulas generalize to the $n$-QPs sector, subject to the qualifications stated in the main text. Withpout loss of generality we then can use the permutation invariance properties of the $|\boldsymbol{r}; \boldsymbol{k}\rangle_{\boldsymbol{a}}$ states to assume that $r_1 < r_2 < \cdots < r_n$. Then the state pick up only the $S$-matrix factors required to permute the QPs that have crossed during the time evolution.

## A.2  Approximate resolution of the identity

Here we show that the WP states can be used to write an approximate resolution of the identity, an observation that will be useful, e.g., in understanding the action of $\mathcal{A}$ on WP states. For simplicity, we limit our discussion to the 2-QP sector. We begin by observing that MWP states form an overcomplete set of states in the 2-QP sector

$$\frac{4\pi\ell^2}{2!} \sum_{\boldsymbol{a}} \int \frac{d^2\boldsymbol{k}\, d^2\boldsymbol{r}}{(2\pi)^2} |\boldsymbol{r}; \boldsymbol{k}\rangle_{\boldsymbol{a}}^{(p)\,(p)} \langle \boldsymbol{r}; \boldsymbol{k}| = \mathbb{1}_2 \,. \tag{A.9}$$

Eqn (A.9) can be established by acting on scattering states $|p_1, p_2\rangle_{a_1, a_2}$. To show that $\hat{R}$-ordered WP states approximately satisfy the same kind of relation, we expand

$$\frac{4\pi\ell^2}{2!} \sum_{a_1, a_2} \int \frac{d^2\boldsymbol{k}\, d^2\boldsymbol{r}}{(2\pi)^2} |r_1, r_2; k_1, k_2\rangle_{a_1, a_2\, a_1, a_2} \langle r_1, r_2; k_1, k_2|$$

$$= \frac{4\pi\ell^2}{2!} \sum_{a_1, a_2} \int \frac{d^2\boldsymbol{k}\, d^2\boldsymbol{r}}{(2\pi)^2} \Big[ \theta_H(r_{21}) |r_1, r_2; k_1, k_2\rangle_{a_1, a_2\, a_1, a_2}^{(p)\,(p)} \langle r_1, r_2; k_1, k_2|$$

$$+ \theta_H(r_{12}) |r_2, r_1; k_2, k_1\rangle_{a_2, a_1\, a_2, a_1}^{(p)\,(p)} \langle r_2, r_1; k_2, k_1| \Big]$$

$$\approx \frac{4\pi\ell^2}{2!} \sum_{a, b} \int \frac{d^2\boldsymbol{k}\, d^2\boldsymbol{r}}{(2\pi)^2} \Big[ \theta_H(r_{21}) |r_1, r_2; k_1, k_2\rangle_{a_1, a_2\, a_1, a_2}^{(p)\,(p)} \langle r_1, r_2; k_1, k_2|$$

$$+ \theta_H(r_{12}) S_{a_2, a_1}^{a_2', a_1'}(k_2, k_1) \Big[ S_{a_2, a_1}^{a_2'', a_1''}(k_2, k_1) \Big]^* |r_1, r_2; k_1, k_2\rangle_{a_1', a_2'\, a_1'', a_2''}^{(p)\,(p)} \langle r_1, r_2; k_1, k_2| \Big] \,,$$

where in the last step we used Eq. (A.3). Finally, by unitarity, we have that

$$\sum_{a_1, a_2} S_{a_2, a_1}^{a_2', a_1'}(k_2, k_1) \Big[ S_{a_2, a_1}^{a_2'', a_1''}(k_2, k_1) \Big]^* = \delta_{a_1', a_1''} \delta_{a_2', a_2''} \,. \tag{A.10}$$

Putting everything together we obtain

$$\frac{4\pi\ell^2}{2!} \sum_{a_1, a_2} \int \frac{d^2\boldsymbol{k}\, d^2\boldsymbol{r}}{(2\pi)^2} |\boldsymbol{r}; \boldsymbol{k}\rangle_{\boldsymbol{a}\, \boldsymbol{a}} \langle \boldsymbol{r}; \boldsymbol{k}| \approx \mathbb{1}_2 \,. \tag{A.11}$$

Here, as in the previous subsection, the approximation becomes asymptotically exact in the large-$\ell$ limit.

# B  Action of $A(q)$ on the vacuum

In this appendix we discuss the approximate relation

$$U(t)A(q)|\Omega\rangle \simeq \sum_n \sum_a \frac{1}{\mathcal{N}_a} \int dx \int \frac{d^n k}{(2\pi)^n} e^{ix\left(q-\sum_j k_j\right)} \mathcal{F}_a^*(k) e^{-i\sum_j \epsilon_{a_j}(k_j)t} |r;k\rangle_a , \qquad \text{(B.1)}$$

with $r$ given by $r_j = x + v(k_j)t$. Here $\mathcal{F}$ denotes the probability amplitude to produce a given wave packet state, and is generally challenging to compute from first principles.

In the following we summarize the argument reported in Ref. [49] for the general validity of Eq. (B.1) and show how, in integrable theories, where $A(q)$ can also be characterized in terms of asymptotic momentum states the relation above can be explicitly derived. Furthermore, if the asymptotic states form factors $F$ are known, $\mathcal{F}$ can be easily obtained from $F$. Finally, we will show how from Eq. (B.1) the standard result for the equal-time two-point function can be recovered.

The validity of the equation above can be argued as follows. The most general form for a wave packet state is

$$U(t)A(q)|\Omega\rangle = \sum_n \sum_a \frac{1}{\mathcal{N}_a} \int d^n x \int \frac{d^n k}{(2\pi)^{n-1}} \times \mathfrak{F}_a^*(k,x_j) e^{-i\sum_j \epsilon_{a_j}(k_j)t} |r;k\rangle , \qquad \text{(B.2)}$$

with $r_j = x_j + v(k_j)t$. Here the time-dependence is given in such a way that the evolution of the wave packet state is correct when $t$ is large enough and the wave packets are separated. Given that all QPs are created through a local operator on a state with a finite correlation length $\xi$, it follows that all positions $x_j$ are within the same region of size approximately $\xi$. Given that we chose $\ell \gg \xi$, we can neglect the differences among $x_j$ and approximate $x_j \simeq x$ for all $j$ and some position $x$. In this case, the function $\mathfrak{F}_n^*(k,x_j)$ can be replaced by another function $\mathfrak{F}_n^*(k,x)$ depending on $x$ alone. Finally, requiring that under translation by a length $y$ the state acquires a phase $e^{iqy}$ fully determines the $x$-dependence to be as in Eq. (B.1).

For an integrable system a convenient basis is that of the Bethe-ansatz eigenstates $|k\rangle_a$. In terms of these one can write

$$U(t)A(q)|\Omega\rangle = \sum_n \sum_a \frac{1}{\mathcal{N}_a} \int \frac{d^n p}{(2\pi)^n} (2\pi)\delta\left(q - \sum_j p_j\right) F_a^*(p) e^{-i\sum_j \epsilon_{a_j}(p_j)t} |p\rangle_a , \qquad \text{(B.3)}$$

where the form factors $F_a(k)$ can be determined through various techniques depending on the model in question. In this case Eq. (B.1) can be derived by re-expressing the Bethe ansatz states in terms of wave packet states. This also gives us a way of determining $\mathcal{F}_a(k)$ from $F_a(k)$.

The starting point is to act with the identity, which we resolve in terms of wave packets with width $\ell/\sqrt{2}$:

$$\mathbb{1} = \sum_n \sum_a \frac{1}{\mathcal{N}_a} \ell^n (2\pi)^{n/2} \int \frac{d^n r \, d^n k}{(2\pi)^n} |r;k\rangle_a^{\ell/\sqrt{2}} \quad {}^{\ell/\sqrt{2}}_a \langle r;k| . \qquad \text{(B.4)}$$

Rewriting the $\delta$-function in momentum space using the identity

$$2\pi\delta(q - \sum_j p_j) = \int dx \, e^{-ix(q - \sum_j p_j)} , \qquad \text{(B.5)}$$

we obtain

$$
U(t)A(q)|\Omega\rangle = \sum_n \sum_a \frac{1}{\mathcal{N}_a} \ell^n (2\pi)^{n/2} \int \frac{d^n\boldsymbol{p}\, d^n\boldsymbol{k}\, d^n\boldsymbol{r}}{(2\pi)^{2n}} \int dx\, F_a^*(\boldsymbol{p})
$$
$$
\times e^{ix(q-\sum_j p_j)} e^{-i\sum_j \epsilon_{a_j}(p_j)t} \left[ \prod_j \tilde{W}_{\ell/\sqrt{2}}^*(p_j; r_j, k_j) \right] |\boldsymbol{r};\boldsymbol{k}\rangle_a^{\ell/\sqrt{2}} .
$$
(B.6)

Next, we can integrate over $p_j$. Since the dominant contribution to the integral is due to the region where $|p_j - k_j| \lesssim \ell^{-1}$ for all $j$, we can approximate $F_a^*(\boldsymbol{p}) \simeq F_a^*(\boldsymbol{k})$. Furthermore, we expand to linear order the energy dependence on $p_j$: $\epsilon_{a_j}(p_j) \simeq \epsilon_{a_j}(k_j) + (p_j - k_j)v_{a_j}(k_j)$; the rationale being that higher-order corrections will give corrections like a diffusive broadening of the wave packet that we neglect here. The integration over $\boldsymbol{p}$ can then be performed to obtain

$$
U(t)A(q)|\Omega\rangle \simeq \sum_n \sum_a \frac{1}{\mathcal{N}_a} \pi^{n/2} \int \frac{d^n\boldsymbol{k}\, d^n\boldsymbol{r}}{(2\pi)^n} \int dx\, F_a^*(\boldsymbol{k}) e^{ix(q-\sum_j k_j)}
$$
$$
\times e^{-i\sum_j \epsilon_{a_j}(k_j)t} \left[ \prod_j e^{-\frac{(r-x-v_{a_j}(k_j)t)}{\ell^2}} \right] |\boldsymbol{r};\boldsymbol{k}\rangle_a^{\ell/\sqrt{2}} .
$$
(B.7)

Finally, we want to integrate over $\boldsymbol{r}$. For this purpose we can re-express

$$
|\boldsymbol{r};\boldsymbol{k}\rangle_a^{\ell/\sqrt{2}} = \int d^n\boldsymbol{x} \left[ \prod_{j=1}^n W_{\ell/\sqrt{2}}(x_j; r_j, k_j) \right] |\boldsymbol{x}\rangle_a .
$$
(B.8)

Integrating over $\boldsymbol{r}$, we can recognize that the integral can immediately be rewritten as a wave packet state of the form Eq. (130) with

$$
\mathcal{F}_a(\boldsymbol{k}) = (2\pi)^{n/2} \ell^n F_a(\boldsymbol{k}) .
$$
(B.9)

We now discuss how the wave packet description above can be used to approximate the equal-time two-point functions

$$
C_2(q,0) = \frac{1}{V} \langle \Omega | A(-q;0)A(q;0) | \Omega \rangle_C .
$$
(B.10)

We further show that in integrable systems the results obtained from the wave packet description agrees with the exact formula obtained through a form factor expansion

$$
C_2(q,0) = \sum_n \sum_a \frac{1}{\mathcal{N}_a} \int \frac{d^n\boldsymbol{k}}{(2\pi)^{n-1}} \delta\left(q - \sum_j k_j\right) |F_a(\boldsymbol{k})|^2 .
$$
(B.11)

While this discussion is not useful in itself, it will be used as a step in the calculation of four-point functions with two identical times, as discussed in Sec. 7.1 and 7.2.

To compute $C_2(q,0)$, we begin by rewriting it as

$$
C_2(q,0) = \frac{1}{V} \langle \Omega | A(-q;0)U^\dagger(t)U(t)A(q;0) | \Omega \rangle_C .
$$
(B.12)

We can then apply Eq. (B.1) to expand $U(t)A(q;0)|\Omega\rangle_C$ and its hermitian conjugate to obtain

$$
C_2(q,0) \simeq \frac{1}{V} \sum_n \sum_a \frac{1}{\mathcal{N}_a} \int \frac{d^n\boldsymbol{k}}{(2\pi)^n} \mathcal{F}_a^*(\boldsymbol{k}) \int \frac{d^n\boldsymbol{k}'}{(2\pi)^n} \mathcal{F}_a(\boldsymbol{k}')
$$
$$
\times \int dy\, e^{iy(q-\sum_j k_j)} e^{-it\sum_j \epsilon_{a_j}(k_j)} \int dy'\, e^{iy'(-q+\sum_j k_j')} e^{it\sum_j \epsilon_{a_j}(k_j')}
$$
$$
\times {}_a\langle \boldsymbol{r}';\boldsymbol{k}'|\boldsymbol{r};\boldsymbol{k}\rangle_a ,
$$
(B.13)

with $r_j = y + v_{a_j}(k_j)t$ and similarly $r'_j = y + v_{a_j}(k'_j)t$.

Now, we would like to perform the integration over $\boldsymbol{k}'$, $y$ and $y'$, in the attempt to recover Eq. (B.11). Assuming the time $t$ is sufficiently large, the wave packets will be well-separated and the overlap ${}_a\langle \boldsymbol{r}'; \boldsymbol{k}' | \boldsymbol{r}; \boldsymbol{k}\rangle_a$ factors into the product of terms ${}_{a_j}\langle r'_j; k'_j | r_j; k_j\rangle_{a_j}$. Note then that within the approximation we used throughout the manuscript

$$
\begin{aligned}
e^{-it\left(\epsilon_{a_j}(k_j) - \epsilon_{a_j}(k'_j)\right)} {}_{a_j}\langle r'_j; k'_j | r_j; k_j\rangle_{a_j} &\simeq {}_{a_j}\langle y'; k'_j | U^\dagger(t)U(t) | y; k_j\rangle_{a_j} \\
&= {}_{a_j}\langle y'; k'_j | y; k_j\rangle_{a_j}.
\end{aligned}
\tag{B.14}
$$

After this substitution, and after having approximate $F_a(\boldsymbol{k}') \simeq F_a(\boldsymbol{k})$, the integral over $\boldsymbol{k}'$ can be readily performed obtaining

$$
C_2(q,0) \simeq \frac{1}{V}\sum_n \sum_a \frac{1}{\mathcal{N}_a} \int \frac{d^n \boldsymbol{k}}{(2\pi)^n} \frac{|\mathcal{F}_a(\boldsymbol{k})|^2}{(2\pi\ell^2)^n} \int dy \int dy' \, e^{i(y-y')\left(q - \sum_j k_j\right)} e^{-\frac{n}{2\ell^2}(y'-y)^2}.
\tag{B.15}
$$

Note that the integrand depends only on $y - y'$, as could have been anticipated due to translational invariance. Therefore, we can set $y' = 0$ and remove the $V^{-1}$ prefactor. Furthermore, on general grounds, we expect the the integral over $\boldsymbol{k}$ to be negligible whenever $|y - y'| \gg \xi$. Given that we assumed $\ell \gg \xi$, the Gaussian envelope can be neglected to obtain

$$
C_2(q,0) \simeq \sum_n \sum_a \frac{1}{\mathcal{N}_a} \int \frac{d^n \boldsymbol{k}}{(2\pi)^n} \frac{|\mathcal{F}_a(\boldsymbol{k})|^2}{(2\pi\ell^2)^n} \int dy \, e^{iy\left(q - \sum_j k_j\right)},
\tag{B.16}
$$

where the integral over $y$ equals $2\pi\delta\left(q - \sum_j k_j\right)$.

Finally, if the system is integrable it is convenient to re-express $\mathcal{F}_a(\boldsymbol{k})$ in terms of $F_a(\boldsymbol{k})$ using Eq. (B.9), therefore re-obtaining Eq. (B.11).

# C Summation formulas for $\tilde{C}_4^{(1,2,1)}(q,t)$

## C.1 Transverse field Ising model

**Lemma 1:** Let $f(z)$ be a $2\pi$-periodic function that is analytic in a strip enclosing the interval $[-\pi, \pi]$ and $q \in \mathbb{R}$. Then we have

$$
\begin{aligned}
\frac{1}{L}\sum_{p\in\text{NS}} f(p)\frac{e^{-it\left(\epsilon(p)+\epsilon(Q-p)\right)}}{1-\cos(p-q)} &= \left[\left(\frac{L}{2} - t\left|\epsilon'(Q-q) - \epsilon'(q)\right|\right)f(q) + f'(q)\sigma\right]e^{-it\left(\epsilon(q)+\epsilon(Q-q)\right)} \\
&\quad - \sum_n \sigma_n \oint_{C_n} \frac{dz}{2\pi} \frac{f(z)e^{-it(\epsilon(z)+\epsilon(Q-z))}}{\left(1-\cos(z-q)\right)\left(e^{i\sigma_n z L} + 1\right)},
\end{aligned}
\tag{C.1}
$$

where $\sigma_n \in \{\pm 1\}$ and $C_n$ are counter-clockwise contours that enclose segments of $[-\pi, \pi]$ such that

$$
\left|\frac{e^{-it(\epsilon(z)+\epsilon(Q-z))}}{e^{i\sigma_n z L} + 1}\right| = o(t^0).
\tag{C.2}
$$

We note that (i) the number of contours required depends on $Q$; and (ii) the choice of contours ensures that the integrals give only sub-leading contributions, except when the linear in $t$ term on the r.h.s. of (C.1) vanishes. The latter occurs when $Q = 2q$. To understand the behaviour of the sum in that case we need to evaluate the leading correction due to the contour integrals. This arises from the contour that encircles $q$. To be specific we take $Q$ such that $\sigma_n = +1$.

Then the contour integral over $C_n$ can be simplified by noting that along the path in the lower half-plane the factor $e^{izL}$ becomes exponentially large in $L$, so that the integral vanishes in the thermodynamic limit. This leaves us with the contribution of the vertical parts of $C_n$ (which we ignore for now) and

$$I_C = - \int_{i\delta + a_{n+1}}^{i\delta + a_n} \frac{dz}{2\pi} \frac{f(z)e^{-it(\epsilon(z)+\epsilon(Q-z))}}{\left(1-\cos(z-q)\right)}. \tag{C.3}$$

For large $t$ this integral can be evaluated using a saddle-point approximation. The result is generally negligible, except in the case where $Q \approx 2q$ because then $\epsilon'(Q-q) - \epsilon'(q) \approx 0$ and the term linear in $t$ becomes very small. The location of the saddle point for $Q \approx 2q$ is

$$z_s = q - \frac{1}{2}\frac{\epsilon'(q)-\epsilon'(Q-q)}{\epsilon''(q)+\epsilon''(Q-q)} + \frac{\sigma}{2}\sqrt{\left(\frac{\epsilon'(q)-\epsilon'(Q-q)}{\epsilon''(q)+\epsilon''(Q-q)}\right)^2 + \frac{8i}{t[\epsilon''(q)+\epsilon''(Q-q)]}}, \tag{C.4}$$

and the saddle-point approximation then gives

$$I_C \approx \sqrt{\frac{2}{\pi}} \frac{f(z_s)e^{-it(\epsilon(z_s)+\epsilon(Q-z_s))}}{(z_s-q)^2\sqrt{it\left(\epsilon''(z_s)+\epsilon''(Q-z_s)\right)-2(z_s-q)^{-2}}}. \tag{C.5}$$

In the regime $t|\epsilon'(q)-\epsilon'(Q-q)| \ll 1$ the saddle point location becomes

$$z_s \approx q + \sigma\sqrt{\frac{i}{\epsilon''(q)t}}, \tag{C.6}$$

which leads to a square root increase in time.

**Lemma 2:** Proceeding similarly as above we can show that, for $k_1, k_2 \in \mathbb{R}$, as $L, t \to \infty$ and $k_1 - k_2 \to 0$

$$\frac{1}{L}\sum_{p\in\mathrm{NS}}\frac{e^{-it(\epsilon(p)+\epsilon(q-p))}f(p)}{\sin\frac{p-k_1}{2}\sin\frac{p-k_2}{2}} = -2i\,\mathrm{sign}(t)\,\mathrm{sign}(v(k_1)-v(q-k_1))f(k_1)$$

$$\times \frac{e^{-i(\epsilon(k_1)+\epsilon(q-k_1))t} - e^{-i(\epsilon(k_2)+\epsilon(q-k_2))t}}{k_1 - k_2} + o(|k_1-k_2|). \tag{C.7}$$

## C.2 IQFT with diagonal scattering

In this appendix we provide some details on how to carry out certain sums over solutions of the Bethe Ansatz equations in form factor expansions of four-point functions in IQFTs with a diagonal scattering matrix. More specifically, we are interested in summing over Bethe states with two particles $|\beta_1, b_1; \beta_2, b_2\rangle$ on a ring of length $R$, where the following momentum constraint is imposed

$$k_{b_1}(\beta_1) + k_{b_2}(\beta_2) = q = \frac{2\pi}{R}n, \quad n \in \mathbb{R}. \tag{C.8}$$

Solving this constraint fixes

$$\beta_2(\beta_1) = \mathrm{arcsinh}\left(\frac{q - k_{b_1}(\beta_1)}{m_{b_2}}\right). \tag{C.9}$$

The remaining Bethe equation for $\beta_1$ is

$$Rk_{b_1}(\beta_1) + \delta_{b_1, b_2}\big(\beta_1 - \beta_2(\beta_1)\big) = 2\pi I_1, \tag{C.10}$$

where we recall that $I_1$ are half-odd integers. It will be useful to rewrite the Bethe equations by changing variables to $z = k_{b_1}(\beta_1)$

$$Q(z) = Rz + \delta_{b_1, b_2}(\beta_1(z) - \beta_2(z)) = 2\pi I_1. \tag{C.11}$$

**Lemma 3:** Consider a momentum $k$ s.t. $e^{ikR} = 1$ and denote by $\tilde{\beta}_{1,2}$ the solutions of the equations

$$k_{b_1}(\tilde{\beta}_1) = k, \qquad k_{b_1}(\tilde{\beta}_1) + k_{b_2}(\tilde{\beta}_2) = q. \tag{C.12}$$

We denote with $\tilde{\beta}_1$ the rapidity for which $k_{b_1}(\tilde{\beta}_1) = k$ and consequently $\tilde{\beta}_2$ define by $k_{b_1}(\beta_1) + k_{b_2}(\beta_2) = q$. Then for large $t \ll R$ we have

$$\frac{1}{R}\sum_{\beta_1} \frac{e^{-it\left(\epsilon_{b_1}(\beta_1)+\epsilon_{b_2}(\beta_2)\right)}}{(k_{b_1}(\beta_1)-k)^2} = \frac{Re^{-it\left(\epsilon_{b_1}(\tilde{\beta}_1)+\epsilon_{b_2}(\tilde{\beta}_2)\right)}}{\left|1-S_{b_2,b_1}(\tilde{\beta}_2-\tilde{\beta}_1)\right|^2}$$
$$+ \frac{t|v(\tilde{\beta}_1)-v(\tilde{\beta}_2)|\mathcal{S}_{b_1,b_2}^{b_1,b_2}(k,q-k;k,q-k)}{\left|1-S_{b_2,b_1}(\tilde{\beta}_2-\tilde{\beta}_1)\right|^2} + \dots,$$

where the dots indicate terms that are subleading in $R$ or $t$ and $\mathcal{S}$ was defined in Eq. (123). In the case of an IQFT with diagonal scattering it takes the simpler form

$$\mathcal{S}_{a,b}^{a,b}(k_a(\alpha),k_b(\beta);k_a(\alpha),k_b(\beta)) = \begin{cases} S_{ab}(\alpha-\beta)-1, & \text{if } t(v(\alpha)-v(\beta)) > 0, \\ S_{ba}(\beta-\alpha)-1, & \text{else.} \end{cases} \tag{C.13}$$

Eqn (C.13) can be derived along similar lines as Lemma 1. We consider the contour integral

$$I(k,q,t) = \sum_{n=1}^{2} \sigma_n \oint_{C_n} \frac{dz}{2\pi} \frac{e^{-it\left(\bar{\epsilon}_{b_1}(z)+\bar{\epsilon}_{b_2}(q-z)\right)}}{(z-k)^2\left[1+e^{i\sigma_n Q(z)}\right]}, \tag{C.14}$$

where $C_{1,2}$ are rectangular contours encircling $(\infty,a)$ and $(a,\infty)$ respectively, where $a$ is fixed by the condition that the imaginary part of $\bar{\epsilon}_{b_1}(z) + \bar{\epsilon}_{b_2}(z-q)$ vanishes for $\text{Re}(z) = a$, $\sigma_n \in \{\pm 1\}$ are chosen such that for $z \in C_n$

$$\left| \frac{e^{-it\left(\bar{\epsilon}_{b_1}(z)+\bar{\epsilon}_{b_2}(q-z)\right)}}{1+e^{i\sigma_n Q(z)}} \right| = o(t^0), \tag{C.15}$$

and

$$\bar{\epsilon}_b(z) = \sqrt{m_b^2 + z^2}. \tag{C.16}$$

Using the residue theorem we see that $I(k,q,t)$ has two types of contributions: (i) the double pole at $z = k$ gives rise to the terms on the r.h.s. of (C.13); (ii) the poles of $1+e^{i\sigma Q(z)}$ occur at the solutions of the Bethe equations (C.11) and using that $Q'(k_{b_1}(\beta_1)) \approx R$ we recover minus the l.h.s. of (C.13). Finally, one can show that $I(k,q,t)$ is small compared to the r.h.s. of (C.13), which gives the desired result.

**Lemma 4:** In the same context of the previous lemma, we consider two momenta $k,k'$ s.t. $e^{ikR} = e^{ik'R} = 1$ and $|k-k'| \ll 1$. Then

$$\frac{1}{R}\sum_{\beta_1} \frac{e^{-it\left(\epsilon_{b_1}(\beta_1)+\epsilon_{b_2}(\beta_2)\right)}}{(k_a(\beta_1)-k)(k_a(\beta_1)-k')} = i\frac{\text{sign}\left(t(v(\tilde{\beta}_1)-v(\tilde{\beta}_2))\right)}{\left|1-S_{b_2,b_1}(\tilde{\beta}_2-\tilde{\beta}_1)\right|^2}\mathcal{S}_{b_1,b_2}^{b_1,b_2}(k,q-k;k,q-k)$$
$$\times \frac{e^{-it\left(\bar{\epsilon}_{b_1}(k)+\bar{\epsilon}_{b_2}(q-k)\right)} - e^{-it\left(\bar{\epsilon}_{b_1}(k')+\bar{\epsilon}_{b_2}(q-k')\right)}}{k-k'} + \dots \tag{C.17}$$

The justification is similar to that of Lemma 3.

# D   Computation of $H_{\mathrm{NR}}$

In this appendix, we determine the long-time limit of $H_{\mathrm{NR}}$ and ultimately establish Eq. (69). We start by rewriting $\widetilde{H}_{\mathrm{NR}}$ as

$$\widetilde{H}_{\mathrm{NR}}(t;\boldsymbol{\kappa}) = i^N \left[ \hat{\Sigma}_{\mathrm{NR}}(\kappa_N) * \hat{\Sigma}_{\mathrm{NR}}(\kappa_{N-1}) * \cdots * \hat{\Sigma}_{\mathrm{NR}}(\kappa_1) * \mathcal{L}(P_{\boldsymbol{\kappa}} - P_{\boldsymbol{p}}) \right], \tag{D.1}$$

where $N = n + m$ and $\hat{\Sigma}_{\mathrm{NR}}(\kappa)$ is an operator acting on functions $f(Q + p - \kappa)$ as

$$\hat{\Sigma}_{\mathrm{NR}}(\kappa) * f(Q + p - \kappa) = \frac{1}{L} \sum_{p \in \mathrm{NS}} \frac{1}{\sin\left(\frac{p-\kappa}{2}\right)} e^{-it(\epsilon(p)-\epsilon(\kappa))} f(Q + p - \kappa). \tag{D.2}$$

It is useful to define

$$B_{\mathrm{NR}}(Q^{(j)};\kappa,\sigma = \pm 1) = \frac{e^{-it\left(\epsilon(\kappa - Q^{(j)} - i\sigma\tilde{\gamma}) - \epsilon(\kappa)\right)}}{\sin\left(\frac{Q^{(j)} + i\tilde{\gamma}\sigma}{2}\right)}, \qquad Q^{(j)} = \sum_{i=j}^{N}(\kappa_i - p_i), \qquad \tilde{\gamma} = 2\mathrm{arcsinh}(\gamma). \tag{D.3}$$

In order to work out the action of the summation operators $\hat{\Sigma}_{\mathrm{NR}}$ the the large-$t$ limit we require the following

**Lemma 5:**   Let $f(z)$ be a $2\pi$-periodic function that is analytic in a strip enclosing the interval $[-\pi,\pi]$, $q,q' \in \mathrm{R}$, $Q \in \mathrm{NS}$. Then we have for $L \to \infty$

$$\frac{1}{L} \sum_{p \in \mathrm{NS}} f(p) \frac{e^{-it\epsilon(p)}}{\sin\left(\frac{p-q}{2}\right)} = -i\,\mathrm{sgn}\left(tv_q\right) f(q) e^{-it\epsilon(q)} + o(t^0),$$

$$\frac{1}{L} \sum_{p \in \mathrm{NS}} f(p) \frac{e^{-it\left(\epsilon(p)-\epsilon(q)+\epsilon(p+Q+i\eta)-\epsilon(q')\right)}}{\sin\left(\frac{p-q}{2}\right)\sin\left(\frac{p-q'+Q+i\eta}{2}\right)} = -i\,\mathrm{sgn}\left(t(v_q + v_{q+Q})\right) \frac{e^{-it\left(\epsilon(q+Q+i\eta)-\epsilon(q')\right)}}{\sin\left(\frac{q-q'+Q+i\eta}{2}\right)} f(q)$$

$$+ 2i\,\mathrm{sgn}\left(t(v_{q'-Q} + v_{q'})\right) \frac{e^{-it\left(\epsilon(q'-Q-i\eta)-\epsilon(q)\right)}}{\sin\left(\frac{q-q'+Q+i\eta}{2}\right)} f(q'-Q-i\eta)\delta_{\mathrm{sgn}\eta,\mathrm{sgn}(v_{q'-Q}+v_{q'})} + o(t^0), \tag{D.4}$$

where the $o(t^0)$ corrections are given in terms of a contour integral. Applying D.4 we obtain the following set of equations

$$\hat{\Sigma}_{\mathrm{NR}}(\kappa_j) * \mathcal{L}(Q^{(j)}) \approx -i\,\mathrm{sgn}(tv_{\kappa_j})\mathcal{L}(Q^{(j+1)}) - B_{\mathrm{NR}}(Q^{(j+1)};\kappa_j,\mathrm{sgn}(v_{\kappa_j})), \tag{D.5}$$

$$\hat{\Sigma}_{\mathrm{NR}}(\kappa_j) * B_{\mathrm{NR}}(Q^{(j)};\kappa,\sigma) \approx -i\,\mathrm{sign}\left(t(v_{\kappa_j} - v_\kappa)\right) B_{\mathrm{NR}}(Q^{(j+1)};\kappa,\sigma)$$

$$+ 2i\,\mathrm{sign}\left(t(v_{\kappa_j} - v_\kappa)\right)\delta_{\sigma,\mathrm{sign}\left(t(v_{\kappa_j} - v_\kappa)\right)} B_{\mathrm{NR}}(Q^{(j+1)};\kappa_j,\sigma). \tag{D.6}$$

Here we have replaced e.g. $\mathrm{sign}(\kappa_j - Q^{(j+1)}) \to \mathrm{sign}(\kappa_j)$, which can be justified under summations over $\kappa_j$ of the form

$$\lim_{L\to\infty} \sum_{\boldsymbol{\kappa}} H_{\mathrm{NR}}(t;\boldsymbol{\kappa}) f(\boldsymbol{\kappa}), \tag{D.7}$$

where $f$ are well-behaved functions. To explicitly compute $H_{\mathrm{NR}}$, we reorder the momenta such that $tv(\kappa_1) < \cdots < tv(\kappa_l) < 0 < tv(\kappa_N) < tv(\kappa_{N-1}) < \cdots < tv(\kappa_{l+1})$ (with $N$ even), which ensures that the second term on the r.h.s of Eq. (D.6) is always zero. After $N$ steps we arrive at

$$\widetilde{H}_{\mathrm{NR}}(t;\boldsymbol{\kappa}) = i^{N+l}(-i)^{N-l}\mathcal{L}(0) - i^N \sum_{j=1}^{l} i^{j-1}(-i)^{N-j} B_{\mathrm{NR}}(0;\kappa_j,-1)$$

$$- i^N \sum_{j=1}^{N-l}(-i)^{j-1}i^{N-j} B_{\mathrm{NR}}(0;\kappa_{l+j},+1). \tag{D.8}$$

Finally we define

$$\tilde{\kappa}_j = \begin{cases} \kappa_j, & \text{if } 1 \le j \le l, \\ \kappa_{N+l+1-j}, & \text{if } l+1 \le j \le N, \end{cases} \tag{D.9}$$

which allows us to rewrite $\tilde{H}_{\text{NR}}$ as

$$\tilde{H}_{\text{NR}}(t;\boldsymbol{\kappa}) = (-1)^l \mathcal{L}(0) + i \sum_{j=1}^{N} (-1)^j B_{\text{NR}}(0, \tilde{\kappa}_j, \text{sign}(l-j+1/2)) \tag{D.10}$$

$$\simeq \mathcal{L}(0) - 2t \sum_{j=1}^{N} (-1)^j v_{\tilde{\kappa}_j},$$

where in the last step we have expanded $B_{\text{NR}}(0, \tilde{\kappa}_j, \sigma)$ for $\tilde{\gamma} t \ll 1$.

# E Computation of $H_{\text{PP}}$

To compute $H_{\text{PP}}$ we take similar steps to the ones that we used to compute $H_{\text{NR}}$. We begin by defining a $\hat{\Sigma}_{\text{PP}}$ operator as

$$\hat{\Sigma}_{\text{PP}}(k) * f(p-k+Q, K-k+R; \dots) = \frac{1}{L^2} \sum_{\substack{p \in \text{NS} \\ K \in \text{R}}} \frac{e^{-it_{21}[\epsilon(K)-\epsilon(p)]}}{\sin\left(\frac{p-K}{2}\right)} \frac{e^{-it_{31}[\epsilon(p)-\epsilon(k)]}}{\sin\left(\frac{p-k}{2}\right)}$$

$$\times f(p-k+Q, K-k+R; \dots),$$

with the dots denoting possible additional arguments of $f$. Defining

$$A_{\text{PP}}(Q, R) = e^{-it_{32}\epsilon(p'_1-Q)} \mathcal{L}(R), \tag{E.1}$$

we can express $H_{\text{PP}}$ in the form

$$\tilde{H}_{\text{PP}}(\boldsymbol{t}; \boldsymbol{k}, p'_1) = e^{it_{32}\epsilon(p'_1)} \left[ \hat{\Sigma}_{\text{PP}}(k_n) * \cdots * \hat{\Sigma}_{\text{PP}}(k_1) * A_{\text{PP}}(Q^{(1)}, R^{(1)}) \right], \tag{E.2}$$

where we define

$$Q^{(i)} = \sum_{j=i}^{n} (p_j - k_j), \qquad R^{(i)} = \sum_{j=i}^{n} (K_j - k_j). \tag{E.3}$$

In order to obtain a closed set of recursion relations we introduce the two additional functions

$$B_{\text{PP}}(Q, R; k, \sigma) = i \frac{e^{-it_{31}[\epsilon(k-R-i\sigma\tilde{\gamma})-\epsilon(k)]-it_{32}\epsilon(p'_1-Q+R+i\sigma\tilde{\gamma})}}{\sin\left(\frac{R+i\sigma\tilde{\gamma}}{2}\right)}, \tag{E.4}$$

$$C_{\text{PP}}(Q, R; k, \sigma) = i \frac{e^{-it_{21}[\epsilon(k-R-i\sigma\tilde{\gamma})-\epsilon(k)]-it_{32}\epsilon(p'_1-Q)}}{\sin\left(\frac{R+i\sigma\tilde{\gamma}}{2}\right)}. \tag{E.5}$$

We then find

$$\hat{\Sigma}_{\text{PP}}(k_j) * A_{\text{PP}}(Q^{(j)}, R^{(j)})$$

$$\approx \text{sign}(t_{21}v(k_j)) \text{sign}\left(v(k_j)t_{31} - v(p'_1)t_{32}\right) A_{\text{PP}}(Q^{(j+1)}, R^{(j+1)})$$

$$+ \text{sign}(t_{21}k_j) \text{sign}(t_{31}v(k_j) - t_{32}v(p'_1)) B_{\text{PP}}(Q^{(j+1)}, R^{(j+1)}, k_j; \text{sign}\left(v(k_j)t_{31} - v(p'_1)t_{32}\right))$$

$$+ \text{sign}(t_{32}(v(k_j) - v(p'_1))) C_{\text{PP}}(Q^{(j+1)}, R^{(j+1)}; k_j, \text{sign}(t_{21}k_j)), \tag{E.6}$$

$$\hat{\Sigma}_{\mathrm{PP}}(k_j)B_{\mathrm{PP}}(Q^{(j)},R^{(j)};\kappa,\sigma)$$

$$\approx \mathrm{sign}(t_{31}(\nu(k_j)-\nu(\kappa)))\,\mathrm{sign}(t_{21}\nu(k_2)-t_{31}\nu(\kappa)+t_{32}\nu(p_1'))B_{\mathrm{PP}}(Q^{(j+1)},R^{(j+1)};\kappa,\sigma)$$

$$-2\sigma\,\mathrm{sign}(t_{32}(\nu(k_j)-\nu(p_1')))\delta_{\sigma,\mathrm{sign}(t_{21}\nu(k_j)-t_{31}\nu(\kappa)+t_{32}\nu(p_1'))}C_{\mathrm{PP}}(Q^{(j+1)},R^{(j+1)};k_j,\sigma)$$

$$+2\sigma\,\mathrm{sign}(t_{32}(\nu(k_j)-\nu(p_1')))B_{\mathrm{PP}}(Q^{(j+1)},R^{(j+1)};k_j,\sigma)\delta_{\sigma,\mathrm{sign}(t_{31}(\nu(k_j)-\nu(\kappa)))}\,, \tag{E.7}$$

$$\hat{\Sigma}_{\mathrm{PP}}(k_j)C_{\mathrm{PP}}(Q^{(j)},R^{(j)};\kappa,\sigma)$$

$$\approx \mathrm{sign}(t_{21}(\nu(k_j)-\nu(\kappa)))\,\mathrm{sign}(t_{31}\nu(k_2)-t_{21}\nu(\kappa)-t_{32}\nu(p_1'))C_{\mathrm{PP}}(Q^{(j+1)},R^{(j+1)};\kappa,\sigma)$$

$$-2\sigma\,\mathrm{sign}(t_{32}(\nu(k_j)-\nu(p_1')))\delta_{\sigma,\mathrm{sign}(t_{21}(\nu(k_j)-\nu(\kappa)))}C_{\mathrm{PP}}(Q^{(j+1)},R^{(j+1)};k_j,\sigma)$$

$$+2\sigma\,\mathrm{sign}(t_{32}(\nu(k_j)-\nu(p_1')))B_{\mathrm{PP}}(Q^{(j+1)},R^{(j+1)};k_j,\sigma)\delta_{\sigma,\mathrm{sign}(t_{31}\nu(k_j)-t_{21}\nu(\kappa)-t_{32}\nu(p_1'))}\,. \tag{E.8}$$

By repeatedly applying the recursion relations above, one could in principle express $\tilde{H}_{\mathrm{PP}}$ in terms of various sign functions (depending on various velocities), and $A$, $B$, and $C$ functions evaluated at $R = Q = 0$:

$$A_{\mathrm{PP}}(0,0) = e^{-it_{32}\epsilon(p_1')}\mathcal{L}(0)\,,$$

$$B_{\mathrm{PP}}(0,0;k,\sigma) = e^{-it_{32}\epsilon(p_1')}\big(\sigma\mathcal{L}(0)-2\big[t_{31}\nu(k)-t_{32}\nu(p_1')\big]\big)\,,$$

$$C_{\mathrm{PP}}(0,0;k,\sigma) = e^{-it_{32}\epsilon(p_1')}\big[\sigma\mathcal{L}(0)-2t_{21}\nu(k)\big]\,. \tag{E.9}$$

Here we have replaced e.g. $\mathrm{sgn}(k-Q^{(2)}) \to \mathrm{sgn}(k_j)$, which can be justified under summations over $\kappa_j$ of the form

$$\lim_{L\to\infty}\sum_{\kappa}H_{\mathrm{PP}}(t;\boldsymbol{k},p_1')f(\boldsymbol{k})\,, \tag{E.10}$$

where $f$ are well-behaved functions.

We now would like to establish the following properties of $\tilde{H}_{\mathrm{PP}}$:

(i) $\tilde{H}_{\mathrm{PP}}-\mathcal{L}(0)$ is finite in the $\gamma \to 0$ limit and the cancellation scheme we proposed is correct.

(ii) The remaining part can be bounded as $\mathcal{O}(|t_{32}|+|t_{31}|)$ for every $n$.

To establish (i), we can inspect the three recursion relations and, by analyzing all possible cases of the sign-functions, we can show the following. If the initial function on the LHS (either $A_{\mathrm{PP}}$, $B_{\mathrm{PP}}$, or $C_{\mathrm{PP}}$), once evaluated at $Q,R = 0$ gives a leading term of $\sigma\mathcal{L}(0)$ (where $\sigma = \pm 1$), then also the sum of the various terms in the RHS, once evaluated at $Q,R = 0$, will yield the same leading contribution of $\sigma\mathcal{L}(0)$. Since the starting point of the recursion is $A_{\mathrm{PP}}$, whose leading term is $\mathcal{L}(0)$, it follows by induction that the leading term of $\tilde{H}_{\mathrm{PP}}$ is again $\mathcal{L}(0)$. Thus the cancellation scheme we proposed works and $H_{\mathrm{PP}}$ is finite.

Once we know that (i) holds, (ii) follows in a straightforward manner. In fact, given the form of the recursion relation, we know that $H_{\mathrm{PP}}$ will be a linear combination of objects like $\big[t_{31}\nu(k_j)-t_{32}\nu(p_1')\big]$ and $t_{21}\nu(k_j)$ with finite integer coefficients. Since the maximum group velocity is finite, all the terms are bounded by $4\nu_{\max}(|t_{32}|+|t_{21}|)$. Therefore, $H_{\mathrm{PP}}(\boldsymbol{t};\boldsymbol{k},P)$ is bounded by some $\boldsymbol{k}$- and $P$-independent constant times $|t_{32}|+|t_{21}|$.

Finally, we focus on the case where $t_{32} = \tau_2 > 0$, $t_{21} = \tau_1 > 0$. We conjecture that by repeated application of the recursion relation one obtains the following formula for $\tilde{H}_{\mathrm{PP}}$

$$\tilde{H}_{\mathrm{PP}}(\tau_1,\tau_2;\boldsymbol{k},P) = \mathcal{L}(0)-2\sum_{j=1}^{2n}(-1)^j\tilde{y}_j\,, \tag{E.11}$$

where the $\tilde{y}_j$ are drawn from the set

$$Y = \{\nu(k_1)\tau_1,\ldots,\nu(k_n)\tau_1,\nu(k_1)(\tau_1+\tau_2)-\nu(P)\tau_2,\ldots,\nu(k_n)(\tau_1+\tau_2)-\nu(P)\tau_2\}\,, \tag{E.12}$$

in increasing order, viz. $\tilde{y}_1 < \tilde{y}_2 < \cdots < \tilde{y}_{2n}$. If the conjecture were true, this would imply that the semiclassical picture presented in Sec. 7.2 is indeed correct. We are currently unable to prove this conjecture for general $n$, however, we can verify it explicitly for $n = 1, 2, 3$.

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
