# Peer review of "Long-time divergences in the nonlinear response of gapped one-dimensional many-particle systems"

_SciPost Physics, doi:SciPost Phys. 19, 086 (2025)_

## Round 2 · Referee Report · Anonymous (Referee 1) · 2025-6-20

Strengths

  • very interesting quasiparticle description
  • timely and important question

Weaknesses

  • not fully clear what are the classes of model displaying the divergence

Report

The paper is very clear and the calculations are all precise and sound. I honestly do not have any other recommendation than to just publish the paper as it stands.
The only slight modification I would ask is to make more clear what are the types of systems the authors expect these divergences to take place. Clearly quasi particles play an important role, but it would be good to give some explicit example.

Recommendation

Publish (surpasses expectations and criteria for this Journal; among top 10%)

---

## Round 2 · Referee Report · Anonymous (Referee 2) · 2025-7-30

Strengths

1 - precise statement for the result 2 - result entails promising experimental developments for strongly interacting systems 3 - sound derivations

Weaknesses

1 - confuse structure 2 - unclear connection with known literature

Report

The manuscript fully meets the criteria.

I found the work sound and the statement sharp. In general, my main concern regards the structure adopted by the authors, together with the relevance of the wave packet analysis. The work introduces certain linearly divergent terms in higher-order response functions for gapped models with stable quasi-particles, at least in a large enough momentum range. Here, the wave packet approach should play the crucial role of extending the models where the result can be applied, namely by including also non-integrable systems. I found though the analytic derivations by the means of wave packets less transparent and intuitive than the ones for integrable models; moreover, it was also though to link computations in the wave packet framework with those for integrable field theories. For sake of clarity, I would have put the extension by the means of wave packets and semiclassics after all the discussion regarding pump-probe responses in integrable systems, or in a dedicated appendix.

I have some questions regarding different points contained in the manuscript:

  1. In eq. (3), how is the term $\langle A(q)\rangle_0$ defined?
  2. "Then the action of the pump potential can be viewed as a quantum quench"; for perturbative quenches (i.e. where $\mu$ is very small) perturbation theory is predicting only in the regime $1/m\ll t \ll \mu^{-1/(2-\Delta)}$ where $\Delta$ is the scaling dimension of the perturbing field (in this case the pump potential). How should I reconcile this result with your statement and your prediction of long-time divergences?
  3. Across eq. (10) the assumption of splitting the two exponentials between the pump and the probe parts is made. As far as I understand, this is valid only for lowest orders in $\mu$ and $\mu'$. How does this fit with the aim of the paper of investigating higher orders?
  4. Is there a typo in eq. (60), in the second denominator of the second line $\frac{1}{k_1-k_1}$?
  5. In eq. (69) you integrate over $r_3$ but there is no variable with that name in the integrand. Is it $r_4$?

Recommendation

Publish (surpasses expectations and criteria for this Journal; among top 10%)

---

## Round 3 · Author Response

We thank both referees for their positive assessments of our manuscript. In the following we reply to their comments in detail and indicate the changes we have made.
Referee 1:
We thank the referee for their comments. A discussion of the class of systems we expect our theory to apply to was given in section 2.3 of our manuscript. We believe that one-dimensional models with stable, gapped, coherent quasiparticle excitations will generically exhibit the kind of nonlinear response we identify. This class includes gapped integrable models like the transverse-field Ising chain in the paramagnetic phase or the sine-Gordon QFT in the attractive regime as well as non-integrable models where stable excitations exist as a result of kinematic protection. Here examples are Haldane-gap spin chains or the spin-1/2 XXZ chain in a staggered magnetic field.
Referee 2:
We thank the referee for their comments. We have followed their advice to restructure our manuscript. We now start with a discussion of nonlinear response in integrable models, and follow this by sections presenting the wave-packet analysis (which applies more widely).
Our responses to the specific points the referee raised is as follows:
-
We thank the referee for pointing out that we had not defined $\langle A(q)\rangle_0$. The latter denotes the expectation values in absence of the perturbation. We have added a sentence in our manuscript explaining this.
-
The nonlinear response formalism is based on an expansion in the strength of the perturbation, and the orders in the expansion gives rise to the different nonlinear response functions. As we show, the latter exhibit divergences. However, it is clear on physical grounds that these divergences must be cut off in the actual response of the system. The way this comes about is that higher response functions have stronger divergences, and summing these to all orders gives a finite result. This is akin to summing infrared divergences in perturbation theory.
-
The pump and probe processes are separated in time. For simplicity we consider an instantaneous pump process at $t=0$, which is then followed by a probe process at later times. Given this separation in time on physical grounds there is no issue with considering higher orders in the parameters $\mu$ and $\mu'$.
-
We thank the referee for spotting this typo, which we have corrected.
-
We thank the referee for spotting this typo, which we have corrected.

---

## Editorial Decision

published